# Flexible circuit mechanisms for context-dependent song sequencing

Frederic A. Roemschied[1,2], Diego A. Pacheco[1,3], Max J. Aragon[1], Elise C. Ireland[1], Xinping Li[1], Kyle Thieringer[1], Rich Pang[1] & Mala Murthy[1✉]

Sequenced behaviours, including locomotion, reaching and vocalization, are patterned differently in different contexts, enabling animals to adjust to their environments. How contextual information shapes neural activity to flexibly alter the patterning of actions is not fully understood. Previous work has indicated that this could be achieved via parallel motor circuits, with differing sensitivities to context[1,2]. Here we demonstrate that a single pathway operates in two regimes dependent on recent sensory history. We leverage the *Drosophila* song production system[3] to investigate the role of several neuron types[4–7] in song patterning near versus far from the female fly. Male flies sing 'simple' trains of only one mode far from the female fly but complex song sequences comprising alternations between modes when near her. We find that ventral nerve cord (VNC) circuits are shaped by mutual inhibition and rebound excitability[8] between nodes driving the two song modes. Brief sensory input to a direct brain-to-VNC excitatory pathway drives simple song far from the female, whereas prolonged input enables complex song production via simultaneous recruitment of functional disinhibition of VNC circuitry. Thus, female proximity unlocks motor circuit dynamics in the correct context. We construct a compact circuit model to demonstrate that the identified mechanisms suffice to replicate natural song dynamics. These results highlight how canonical circuit motifs[8,9] can be combined to enable circuit flexibility required for dynamic communication.

During courtship, *Drosophila* males chase and sing to females[10,11] (Extended Data Fig. 1a); song is generated via wing vibration and composed into bouts of two primary modes termed 'pulse' and 'sine' (Fig. 1a). Male song patterning, timing and intensity are known to be modulated by feedback cues stemming from the female[3,12]. Here we investigate how song production neurons in the brain and VNC[4–7,13,14] are functionally organized to generate different song patterns in different contexts. We utilize a combination of broad-range optogenetic activation in freely behaving animals, automated behavioural quantification, neural recordings and manipulations, and circuit modelling.

## Context alters sequencing of male song

Each song bout consists of either simple trains of a single mode (pulse or sine only) or complex trains of rapid alternations between song modes (Fig. 1a,b), and males continually switch between singing simple and complex songs throughout courtship (Extended Data Fig. 1b). Previous work has demonstrated that males produce pulse song, the louder mode, at larger distances to the female, and sine song once closer[3,12,13,15], and has suggested that alternation between modes involved dedicated descending pathways for pulse and sine song that mutually inhibit each other to control song output[16]. We collected a large dataset of

courtship interactions, combining high-resolution video and audio[3,17] (Extended Data Fig. 1d). When examining song bout composition, we found that at close proximity to the female (less than 4 mm), males sing longer, complex bouts composed of alternations between pulse and sine elements, but beyond 4 mm, they sing shorter pulse-only bouts (Fig. 1c–f); these two contexts occur throughout courtship and also correspond to differences in male forward velocity (Extended Data Fig. 1e–g). Although song bout composition is a smooth function of distance, we term these two contexts 'near' and 'far' throughout the study, for simplicity. Song bout complexity may be desirable to the female, as the majority of bouts immediately preceding copulation are complex (Extended Data Fig. 1h,i).

Song at all distances is biased to bouts with leading pulse song ('p' for pulse-only bout or 'ps...' for complex bout starting with pulse; Fig. 1d), suggesting that the song pathway is organized to drive activity in pulse-generating neurons initially, in both contexts. The production of complex sequences might then arise via reciprocal interactions between pulse-producing and sine-producing neurons, but only in the near context. Finally, as the change in song complexity near the female is coupled with longer song bouts (Fig. 1f,h), inhibition to the song pathway (to suppress song when no female is present, or to keep song bouts short when far from a female) may be lifted when the male is near the female. Below we test these hypotheses.

[1]Princeton Neuroscience Institute, Princeton University, Princeton, NJ, USA. [2]Present address: European Neuroscience Institute, Göttingen, Germany. [3]Present address: Harvard Medical School, Boston, MA, USA. ✉e-mail: mmurthy@princeton.edu

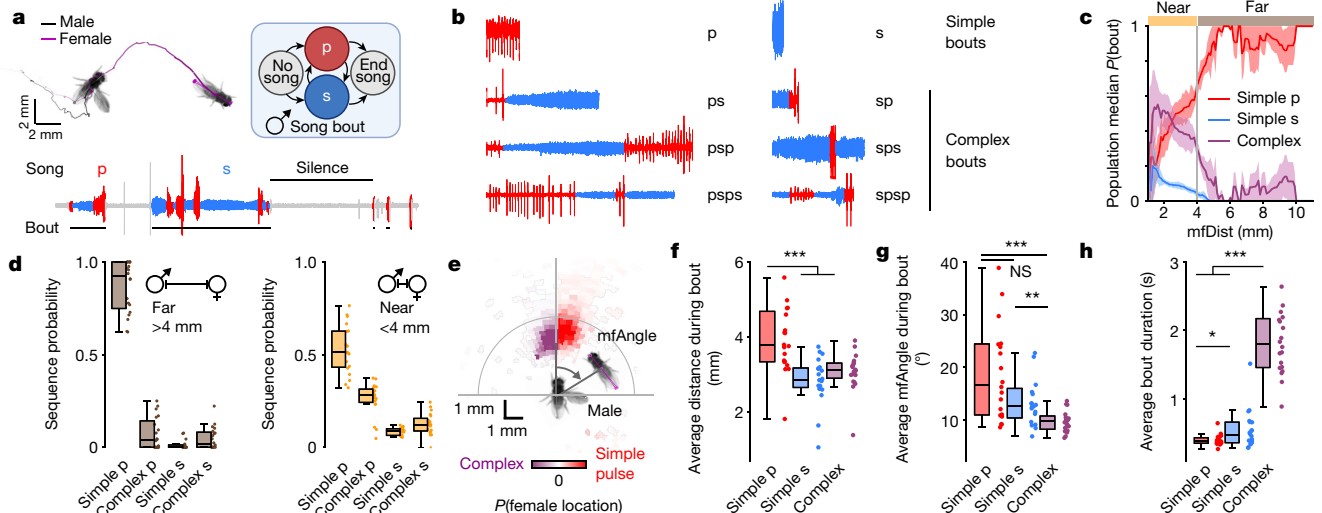

**Fig. 1 | Context-dependent differences in song sequencing in
*D. melanogaster*. a**, *Drosophila* male courtship song is structured into bouts
comprising two main modes: 'pulse' (p) and 'sine' (s). We focus on song bout
patterning, although the duration, amplitude and spectral modulation of pulse
and sine trains constitute other sources of song variability[11,12] (Extended Data
Fig. 1c). **b**, Song bouts consist of either simple pulse or sine trains, or complex
sequences involving continuous alternations between modes. **c**, Population-
averaged probability (median ± median absolute deviation from the median) of
wild-type males singing simple pulse, simple sine or complex bouts at a given
mfDist. The grey vertical line indicates the distance threshold of 4 mm used to
define far and near song bouts. **d**, The distribution of song sequence types
differs far versus near the female. Complex p are complex bouts starting in
pulse mode. Complex s are complex bouts starting in sine mode. Both far from
and near the female, simple pulse bouts constituted the majority of all bouts
(more than 95% and around 55%, respectively), followed by complex 'ps…' bouts

near the female (around 30%). Simple sine bouts constituted the minority of
bouts at all distances. **e**, *P*(female location) during the production of simple
pulse (red, right half) versus complex bouts (purple, left half) in male-centric
coordinates (male at origin), averaged across recordings. mfAngle is the angle
of the female thorax relative to the body axis of the male. Complex bouts are
more likely to be produced when females are close and in front of the male.
**f,g**, Average mfDist (**f**) and mfAngle (**g**) during simple and complex song bouts.
**h**, Average duration of simple and complex song bouts. For **c–h**, *n* = 20 wild-type
males (biological replicates) courting wild-type females (see Supplementary
Table 2 for genotypes). For **d**, **f–h**, central mark indicates the median; the
bottom and top edges of the box indicate the 25th and 75th percentiles,
respectively. Whiskers extend to 1.5 times the interquartile range away from
the box edges. For **f–h**, Wilcoxon rank-sum test for equal medians. *$P < 0.05$,
**$P < 0.01$, ***$P < 0.001$, NS, not significant.

## VNC rebound circuits enable complex song

We expressed csChrimson[18] in two types of song-producing neurons,
either pIP10 brain-to-VNC descending neurons[4,14] (one neuron per
hemisphere; Fig. 2a) or TN1 VNC neurons[5,19] (a population of roughly
30 neurons in the wing neuropil of the VNC divided into 5 subtypes
(TN1A–E); Fig. 2a), and analysed song produced following bilateral
activation. Even though *Drosophila* males sing via unilateral wing vibra-
tion, both the extended wing and the closed wing receive similar motor
activity during song production, indicating that song patterning is
independent of wing choice[20].

By utilizing an optogenetic stimulation protocol that spanned mul-
tiple orders of magnitude in both irradiance and duty cycle (Fig. 2a,b),
we explored how varying activity in these two cell types affected song
production. Consistent with previous findings[4,5,14], activation of either
pIP10 or TN1 neurons in solitary males drove stimulus-locked pulse
or sine song, respectively (Fig. 2c,h). However, in a fraction of males,
strong optogenetic stimuli drove pIP10 neurons to produce 'rebound'
sine song following the offset of pulse song (Fig. 2b,c and Extended Data
Fig. 2a; consistent with the observation in ref. 13). Strong stimulation
of TN1 neurons drove reliable sine song with some intermittent pulse
song (Fig. 2h and Extended Data Fig. 2b), as expected given that the TN1
population (see Methods) comprises some pulse-driving neurons[5,21].

The restriction of rebound sine following activation of pIP10 to high
optogenetic activation levels suggests that the activity dynamics that
generate complex bouts are under inhibition in solitary males, pos-
sibly due to a lack of male arousal. Consistent with this hypothesis,
optogenetic activation of either TN1 or pIP10 in males paired with
females reliably drove long bouts of complex song across a broader

range of stimulus parameters versus in solitary males (Fig. 2d,f,g,i,k–l
and Extended Data Fig. 2c,d) and this complex song was driven pre-
dominantly when near the female (Extended Data Fig. 2e,f), suggesting
that female sensory cues unlock the ability of pIP10 or TN1 neurons to
produce complex song; rebound song is not produced in the presence
of males (Extended Data Fig. 2g,h).

Mutual inhibition between pulse-producing and sine-producing
neurons (red and blue nodes in Fig. 2m), combined with cell-intrinsic
rebound excitability[8], could account for complex bout generation in the
functionally disinhibited circuit; activity of the pulse node (depicted as
a single node, but comprising multiple pulse-driving cell types) would
drive pulse song production and inhibit sine production, whereas termi-
nation of activity in the pulse nodes would stop pulse song production
and release inhibition of the sine node (again, probably comprising
multiple sine-driving cell types), leading to post-inhibitory rebound
activity and production of rebound sine song. In this simplified model
(in which pIP10 provides input primarily to the neurons of the pulse
node), the pulse and sine nodes consist of two units: one that provides
excitation (to drive motor output) and another that provides inhibi-
tion (to suppress the other song mode). As activation of either pIP10
or TN1 neurons in decapitated male flies still resulted in rebound sine
or pulse song, respectively, comparable with that produced in intact
males (Fig. 2e,f,j,k and Extended Data Fig. 2i,j), the rebound circuit
must be fully contained within the VNC.

## Neural signatures of the rebound circuit

We next activated pIP10 neurons while recording from Dou-
blesex (Dsx) neurons in the VNC (Fig. 3a,b; see Methods). VNC

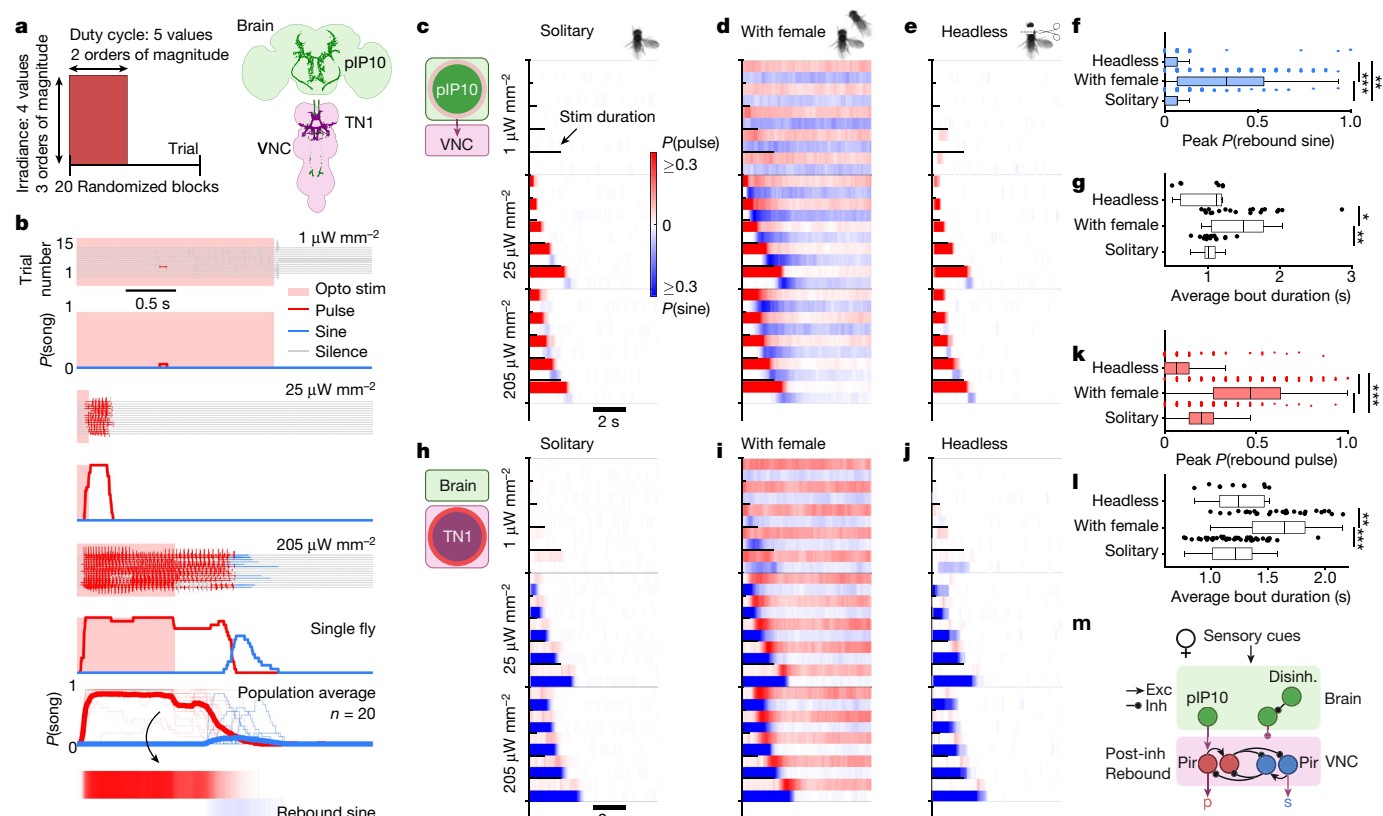

**Fig. 2 | Reciprocal interactions between pulse-producing and sine-producing neurons in the presence of a female. a**, Broad-range optogenetic stimulation of song neurons pIP10 and TN1 (see Methods). One block is 15 trials for 8 s each. Neuron schematic in **a** was adapted from ref. 5, Elsevier, and ref. 14, Elsevier, under a Creative Commons licence CC BY 4.0. **b**, Song production per trial and time-resolved song probabilities across trials following optogenetic activation of pIP10 neurons in a solitary male. Responses are shown for 3 out of 20 randomized stimulus blocks. Pulse and sine probability for the third example stimulus block, averaged across $n = 20$ recordings (bottom). Rebound sine is the production of sine song immediately following pulse song production. Opto stim, optogenetic stimulation. **c**–**e**,**h**–**j**, Average song probabilities (**b**) for all stimulus blocks (distinct stimuli per pair of rows); activation of pIP10 (**c**–**e**) or TN1 (**h**–**j**) in solitary males (**c**,**h**), males paired with a wild-type female (**d**,**i**) and decapitated solitary males (**e**,**j**). **f**, Rebound sine probability following activation of pIP10 neurons (highest irradiance level only) in solitary, female-paired or headless males. Female presence promotes complex bout (pulse followed by rebound sine) generation following pIP10 activation. **g**,**l**, Average duration of song bouts generated via activation of pIP10 (**g**) or TN1 (**l**) neurons in solitary, female-paired or headless males. Female presence promotes longer song bouts following activation of either neuron type. **k**, Rebound pulse probability following activation of TN1 neurons (highest irradiance level only) in solitary, female-paired or headless solitary males. Female presence promotes complex bout generation (sine followed by rebound pulse) following TN1 activation. **m**, Simplified circuit model of song pathway; female cues 'unlock' complex bout generation via modulation of post-inhibitory (post-inh) rebound excitability (exc) in pulse-driving and sine-driving neurons of the ventral nerve cord (VNC). Disinh., disinhibition. Pir, post-inhibitory rebound. For **c**–**g**, $n = 20$ (solitary), 20 (with female) and 10 (headless) males (biological replicates). For **h**–**l**, $n = 23$ (solitary), 28 (with female) and 10 (headless) males (biological replicates). For **f**,**g**,**k**,**l**, Wilcoxon rank-sum test for equal medians. $*P < 0.05$, $**P < 0.01$ and $***P < 0.001$. Central mark indicates the median; the bottom and top edges of the box indicate the 25th and 75th percentiles, respectively. Whiskers extend to 1.5 times the interquartile range away from the box edges.

Dsx⁺ neurons include both TN1 neurons[5] and dPR1 neurons[4], and all are excitatory[22]. Although TN1A neurons drive sine song, other subtypes of Dsx⁺ TN1 neurons probably contribute to pulse song production[5,21]. We therefore expected to observe neural activity among the TN1 population both correlated and anti-correlated with pIP10 activation (and therefore implicated in pulse or rebound sine song production, respectively); importantly, these subsets should be distinct from each other across repeated optogenetic stimulation.

We observed a broad range of temporal response patterns within the TN1 population following pIP10 activation. The activity of nearly half of the recorded TN1 neurons was positively correlated to the pIP10 stimulus, whereas a smaller fraction showed activity perfectly anti-correlated to the stimulus (Fig. 3b), with alternating peaks in activity persisting beyond the stimulation (Fig. 3c–e and Extended Data Fig. 3a,b). We found these anti-correlated pairs on both sides of the VNC (as expected, to drive pulse–sine rebound

activity in both wings). We had expected to detect only a small number of sine-producing neurons, given that the fraction of stimulus presentations with rebound sine rarely exceeded 30% in solitary or headless males (Fig. 2c,e,f). Imaging from either pIP10 axons or Dsx⁺ dPR1 neurons showed tight correlation to the optogenetic stimulus (not shown). We hypothesize the existence of intermediary inhibitory neurons that are responsible for coupling in the rebound circuit (Fig. 2m).

To confirm the proposed role of rebound excitability in driving sine and complex song, we recorded song of homozygous mutant males lacking the rebound-facilitating hyperpolarization-activated cation current $I_h$[23]. These mutant males were able to sing, but sang mostly simple pulse bouts, independent of distance to the female (Extended Data Fig. 3c–f). Reducing expression of either $I_h$ or *Rdl* (GABA-A receptor, required for post-inhibitory rebound) in TN1 neurons also reduced song complexity (Extended Data Fig. 3g–i).

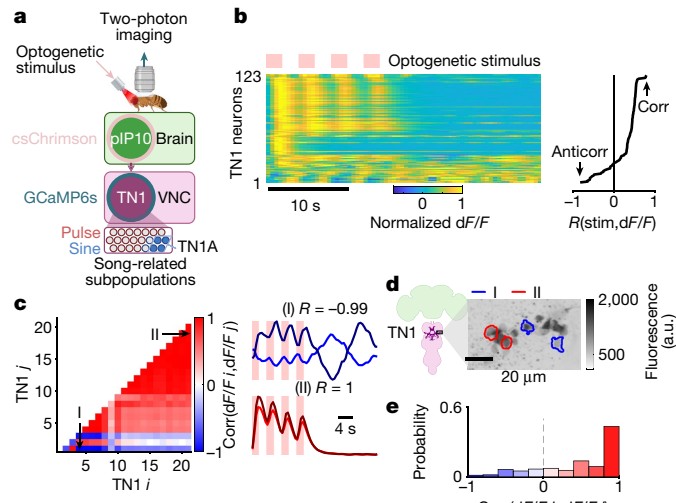

**Fig. 3 | Investigation of rebound dynamics among Dsx⁺ TN1 neurons of the VNC. a**, Two-photon calcium imaging from VNC Dsx⁺ TN1 neurons combined with optogenetic activation of pIP10 descending neurons (see Methods for details; see Supplementary Table 2 for genotypes). The numbers of pulse-related and sine-related subpopulations are according to refs. 5,21. Schematic in **a** was created using BioRender (https://biorender.com). **b**, TN1 neurons show diverse calcium response dynamics following pIP10 activation (123 TN1 neurons across 3 biological replicate flies). Each row shows the normalized calcium response (d$F/F$) of a single soma, averaged over seven trials (and each trial contained four stimulus presentations), and responses are sorted by their correlation (corr) with the optogenetic stimulus. The optogenetic stimulus pattern was chosen to produce pulse song followed by rebound sine (see Fig. 2c) in solitary males. **c**, Pairwise correlation of trial-averaged activity between any two TN1 neurons from one hemisphere in one male (to avoid any cross-hemisphere effects due to, for example, wing choice), following pIP10 activation. Examples of strong anti-correlation and correlation are shown on the right (I/II; $P < 1 \times 10^{-50}$), where light-red boxes indicate stimulus intervals. **d**, Time-averaged fluorescence of the calcium indicator GCaMP6s expressed in Dsx⁺ TN1 neurons. The red and blue regions of interest correspond to anti-correlated or correlated pairs shown in **c**. a.u., arbitrary units. Schematic in **d** was adapted from ref. 5, Elsevier. **e**, Distribution of calcium response correlation coefficients (computed per hemisphere and per male) across TN1 recordings in $n = 3$ males. Colours are as in **c**. Owing to the near-perfect anti-correlation observed for some neuron pairs, we assume that the intermediary neurons of the rebound circuit (Fig. 2m) are mutually inhibitory, in addition to providing inhibition to song-generating neurons. For **c**,**e**, '$i$' and '$j$' denote indices to pairs of recorded TN1 neurons.

## Female sensory cues enable complex song

To determine what brain mechanisms drive the rebound circuit in the VNC, we explored the role of P1a[6,24], a subset of pC1 neurons[25] and pC2 (refs. 7,22,26) cell types, previously implicated in song production.

P1a neurons are driven by taste cues collected during tapping[27]; these neurons in turn can drive a persistent arousal state[28]. Because P1a neurons have been suggested to be upstream of pIP10 neurons[4], we hypothesized that activating P1a neurons in solitary males would mimic our results with pIP10 activation in the presence of a female (Fig. 2d). By contrast, we found that activation of P1a neurons in solitary males produced persistent and variable song (Fig. 4a and Extended Data Fig. 5a) along with suppression of wing extension during the optogenetic stimulus (Extended Data Fig. 5b). Although activation of additional pC1 neurons[13] or longer P1a activation[28] can drive stimulus-locked song, our data indicate that P1a activity alone is probably insufficient for temporally precise initiation of complex song.

pC2 neurons in males[29] consist of two subtypes (pC2l and pC2m; Extended Data Fig. 4a,b) and detect both visual and auditory cues[7,26]; in the female FlyWire connectome[30,31], pC2l neurons receive direct inputs from both visual (lobula columnar neurons) and auditory projection

neurons. Activation of pC2 neurons in solitary males drove pulse song followed by rebound sine, similar to pIP10 activation in the presence of a female (Fig. 4b and Extended Data Fig. 5c). We also observed persistent and variable song in the period outside of optogenetic activation, as well as a near-linear relationship between the duration of the optogenetic stimulus and the amount of rebound sine song (Fig. 4c), suggesting that pC2 neural activity controls the transition from simple to complex bout generation. Pulse song is also composed of two main types (Pfast and Pslow)[13], and the duration of pC2 activity determined the selection of pulse type: brief activity mainly drove Pfast (like activation of pIP10 in solitary males), whereas more sustained activity increased the relative amount of Pslow (like activation of pIP10 near a female; Extended Data Fig. 5d). These results support the conclusion that pC2 neurons serve as a main determinant of song composition.

Together, these results suggest that pC2 neurons directly drive pulse song production via pIP10, but simultaneously drive P1a neurons to generate persistent song and functionally disinhibit the rebound circuit in the VNC to enable complex song bouts (Fig. 4p). In line with this hypothesis, we found that simultaneous activation of pIP10 and P1a neurons in solitary males produced highly reliable and long complex bouts, well beyond the levels observed for activation of the individual neuron types, including pIP10 activation in males near a female (Fig. 4d–f and Extended Data Fig. 5e).

We next focused on whether pC2 neurons are required for singing. A previous study[7] has reported increased amounts of song in males with blocked synaptic transmission in pC2 (via expression of tetanus toxin light chain (TNT))[32], so we recreated the pC2 > TNT flies and re-ran the silencing experiment in our new behavioural rig (Extended Data Fig. 1d). By contrast, we found that silencing pC2 chemical synapses led to an overall reduction in song (Fig. 4g), an increase in the relative amount of simple pulse bouts and a reduction in song complexity (Fig. 4h,i). These new results support a model with pC2 at the top of the song circuit hierarchy; a direct connection from pC2 to pIP10 has been confirmed via expansion microscopy[33].

We propose that P1a neurons mediate functional disinhibition (rather than direct excitation) of the VNC rebound circuit. Although similar mechanisms, the former is computationally favourable, as disinhibitory gating preserves the dynamic range for processing of sensory information in target neurons, reduces spurious responses[9] and is more consistent with our observation that P1a activity does not directly drive song bouts (Fig. 4a). If P1a neurons disinhibit the rebound circuit, then a separate source of excitation is needed to drive song sequences, now identified as pC2 neurons that mediate parallel drive to both pIP10 and P1a neurons (Fig. 4p).

Although male brain connectome data are not yet publicly available, we analysed the female FlyWire connectome[30,31] for GABAergic disinhibitory motifs downstream of pC1 neurons (P1a neurons are a subset of male pC1 neurons; pC1 neurons also exist in females). We found that disinhibition is a common motif downstream of all subtypes of pC1 neurons in females (Extended Data Fig. 5f,g). We activated P1a neurons while imaging from all GABAergic neurons in male flies (Extended Data Fig. 5h), and found regions of interest corresponding to neurons with either activity immediately following P1a activation (we term these 'F1 follower neurons') or inhibited by F1 follower activity ('F2 follower neurons'; Extended Data Fig. 5i–k). F2 followers were dispersed, suggesting the existence of multiple disinhibitory circuits (Extended Data Fig. 5f,g).

We investigated the contribution of both visual[3,12] and chemosensory[27] cues in driving complex song. We found that male tap rate (see Methods; Fig. 4j and Extended Data Fig. 5l) is higher during complex song bouts versus either before these bouts or during or before simple bouts (Fig. 4k,l; also true for wild-type song, Extended Data Fig. 5m), suggesting that acute (tap-triggered) activation of P1a during an ongoing bout, rather than P1a-mediated arousal on longer timescales, promotes complex bout generation. Consistent with this, priming the

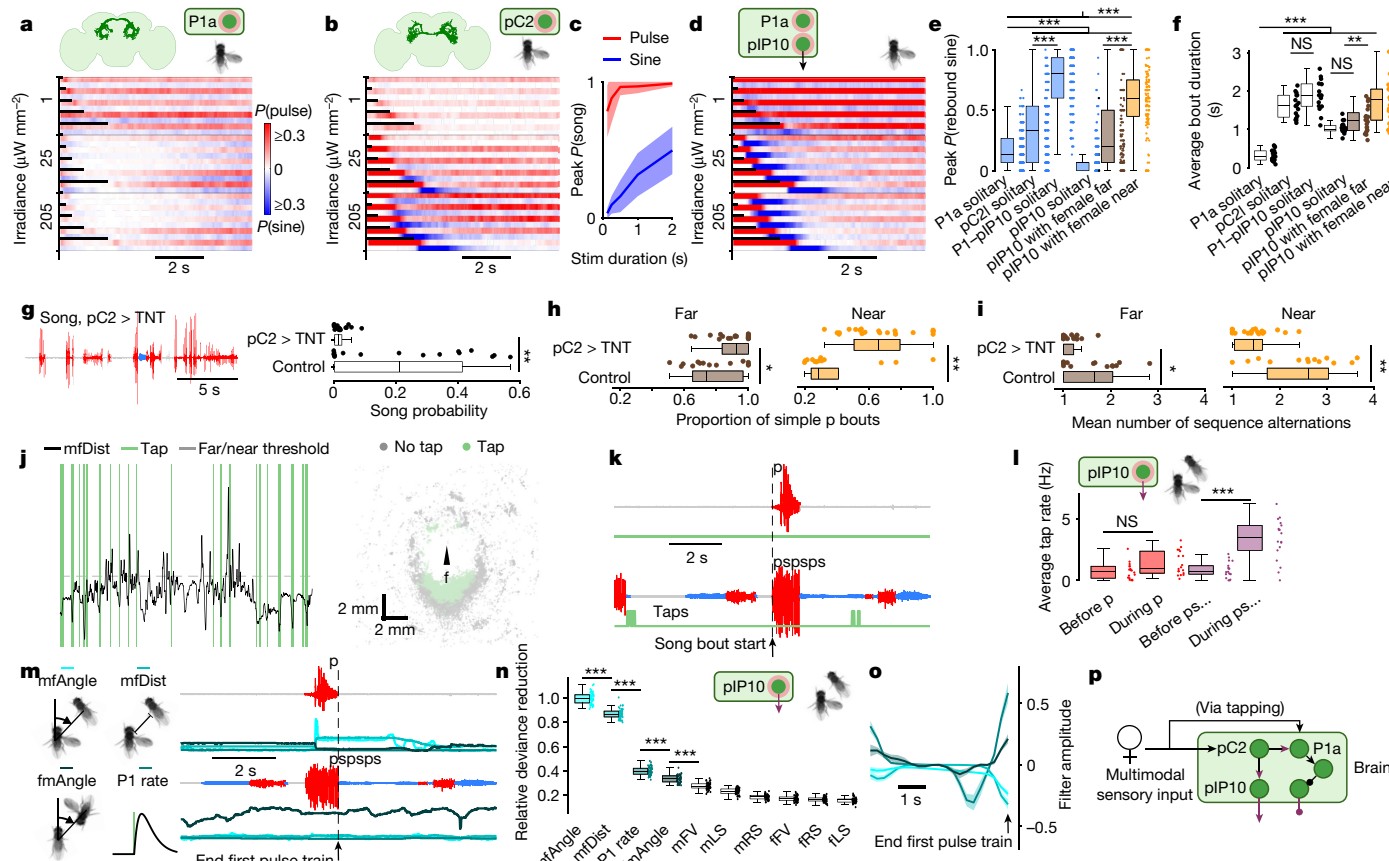

**Fig. 4 | Acute female sensory cues promote complex song bout generation.**
**a,b,d**, Pulse and sine song probabilities following optogenetic activation of P1a
(**a**), pC2 (**b**), or both pIP10 and P1a neurons (**d**), in solitary male flies (*n* = 17, 16
and 16 biological replicates; genotypes are available in Supplementary Table 2).
Schematic in **a** was adapted from ref. 24, eLife Sciences, under a Creative
Commons licence CC CY 4.0. Schematic in **b** was adapted from ref. 7, Elsevier.
**c**, Peak song probability per optogenetic stimulus duration for pC2 neurons
(25 µW mm⁻²). **e,f**, Peak rebound sine probability (**e**) and average bout duration
(**f**) for optogenetic activation (25 and 205 µW mm⁻²) of pC2, pIP10 or P1a
neurons in solitary males or males paired with a wild-type female (data shown in
**a,b,d**; Fig. 2c,d). **g–i**, Song amount (**g**), proportion of simple pulse bouts (**h**) and
song complexity (mean number of pulse–sine or sine–pulse alternations) (**i**) in
pC2 > TNT males paired with wild-type females. **j**, Automated tap detection
(green; see Methods) and mfDist (black) with a 4-mm threshold for far or near
context (grey horizontal dashed line) from an example recording (left). Male
locations during tap and no tap events, in female (f)-centric coordinates
(recording is the same as on the left) (right). **k**, Examples of simple (top) and
complex (bottom) pulse bouts along with detected taps (green). **l**, Average tap
rate (see Methods) before and during simple and complex pulse-leading bouts,

driven by activation of pIP10 in males paired with a wild-type female (Fig. 2d;
*n* = 18 biological replicates). **m**, To fit generalized linear models (GLMs)
predicting pulse bout type (simple versus complex), we used movement
features or P1 rate (shades of cyan; see Methods) over the 5 s preceding the end
of the first pulse train. fmAngle, female–male angle; mfAngle, male–female
angle. **n**, GLM relative deviance reduction for features predicting bout type
(**m**). Input features are ranked by their predictive power (*n* = 51 model fits on
random subsets of data from *n* = 18 biological replicates; see Methods). fFV,
female forward velocity; fLS, female lateral speed; fRS, female rotational
speed; mFV, male forward velocity; mLS, male lateral speed; mRS, male
rotational speed. **o**, GLM filters for the four most predictive features in **n**.
**p**, Updated model of the brain circuitry involved in male song sequencing.
In **e,f,h,i,l,n**, Wilcoxon rank-sum test for equal medians. **P* < 0.05, ***P* < 0.01 and
****P* < 0.001. For **g**, ***P* < 0.01, two-sample Kolmogorov–Smirnoff test for equal
distributions. For **g–i**, *n* = 21 and *n* = 17 biological replicates for experimental
and control groups. For **g,k,m**, red and blue indicate pulse and sine song,
respectively. For **e–i,l,n**, central mark indicates the median; the bottom and top
edges of the box indicate the 25th and 75th percentiles, respectively. Whiskers
extend to 1.5 times the interquartile range away from the box edges.

male (and driving P1a) via exposure to a female (Extended Data Fig. 5o)
only weakly enhanced the complexity of optogenetically driven song
compared with solitary males not subject to priming (Extended Data
Fig. 5p–u). These results corroborate that P1a neurons have a modula-
tory effect on behaviour at short timescales[34].

Using generalized linear modelling[3] (Fig. 4m–o; see Methods), we
found that reductions in the angle of the female's body relative to the
body axis of the male (mfAngle; see Fig. 1e), in addition to male–
female distance (mfDist), within the 1 s leading up to the end of the first
pulse train in a song bout, were the most predictive of whether a pulse
bout ended and remained simple, or continued to become a complex
bout (Fig. 4m–o). Compared with mfAngle, an estimate of P1a activity
derived from the tap detection data (see Methods) had only roughly

40% predictive power, suggesting that tapping and the resulting activity of P1a neurons alone do not fully predict bout complexity. These
results imply that combined sensory modalities contribute to song
bout complexity: probably visual activity (encoding female distance
and angle) relayed through pC2 and tap rate relayed through P1a both
contribute to driving complex song sequences (Fig. 4p). pC2 neurons
can also be driven by auditory activity in the presence of another male[7].
For wild-type song, these results hold, but in addition the male's own
speed (his forward velocity) is predictive of bout complexity (Extended
Data Fig. 5n), consistent with previous work showing that speed influences song choice, even in blind males[3]. Indeed, in the absence of a
female, persistent and variable song driven by P1a activation (Fig. 4a)
is preceded by an increase or decrease in self motion, respectively

(Extended Data Fig. 7i). Together, these results support a model (Fig. 4p) in which different sensory cues (for example, vision or taste) and parallel pathways contribute to the choice of simple versus complex bouts during male–female courtship.

## A circuit model of song patterning

Our behavioural and neural imaging results suggest how naturalistic song statistics arise from the specific functional architecture of the male song circuit. First, we found evidence for a core rebound circuit in the VNC with mutual inhibition between pulse-producing and sine-producing neurons and rebound dynamics in the inhibited nodes (Fig. 2m and Extended Data Fig. 3d–f,h,i). Second, we found evidence for a direct pulse pathway from the brain to the VNC that integrates sensory signals from the female (Fig. 4p). Third, our results suggest a disinhibitory brain pathway onto both nodes of the core circuit, which is driven by sensory input of different modalities (for example, taste and vision via P1a and pC2, respectively). Two mechanisms to drive P1a and downstream circuitry could facilitate continuous complex song production during different aspects of courtship (Extended Data Fig. 1a). To test whether these few computational features are sufficient to explain naturalistic song statistics, we implemented them in a spiking neural circuit model (see Methods; Fig. 5a), comprising only four nodes (termed 'pC2', 'inh' for inhibitory, 'p' for pulse and 's' for sine). Sensory input to the pC2 node was modelled as naturalistic mfDist (see Methods and Supplementary Table 3 for details). This simple model was sufficient to recapitulate naturalistic song bout statistics both far from and near the female (Fig. 5b–e; compare Fig. 1d,f).

Removing individual computational features in the model (see Methods) resulted in overall worse fits to the data than the full model (Fig. 5f), especially when removing disinhibition or rebound excitability of the sine node. Fit performance for a model lacking rebound pulse but capable of rebound sine was similar to that of the full model, highlighting the relative importance of rebound excitability of the sine node (compared with the pulse node) as a computational feature of the song circuit. This is consistent with our conclusion that the pulse production pathway is driven directly via sensory input to pC2 and subsequently to pIP10 (Fig. 4p), but that the sine node does not require direct drive. Indirect drive of the sine node explains the small amount of simple sine song observed in both experiments and simulations (Figs. 1d and 5c), as disinhibition-mediated rebound activity can occasionally drive the sine neuron first (Extended Data Fig. 6a), depending on the internal (membrane voltage) states of the sine and pulse neurons. One possible advantage of the proposed song circuit design based on dominant or leading input to one node of a core rebound circuit is simplicity of control, as theoretically, this architecture allows for switching between simple pulse song and arbitrarily complex pulse–sine sequences, by solely adjusting the level and timing of pIP10 activity. To test this, we used closed-loop optogenetic activation of pIP10 during courtship, triggered on the real-time detection of sine song (see Methods; Extended Data Fig. 7d,e), and found that such activation increased both bout complexity and duration (Extended Data Fig. 7f), uncovering that in *Drosophila*, patterned activity of a single descending neuron (acting on a disinhibited VNC circuit due to female presence) suffices to generate highly complex song outputs.

Experimental data were best described when our circuit model comprised a disinhibitory motif, not a quasi-equivalent excitatory motif, as this failed to produce song bouts with leading sine song (Fig. 5f and Extended Data Fig. 7a–c). In principle, context-dependent (dis-)inhibition could also be achieved via combinations of descending neuromodulatory or peptidergic systems, and ionotropic systems, although such modulation would need to be on timescales of milliseconds to seconds. In addition, in the biological circuit, other factors such as spike-frequency adaptation (present but not explicitly modelled here; Fig. 5b) could have a role. In line with this hypothesis, we performed

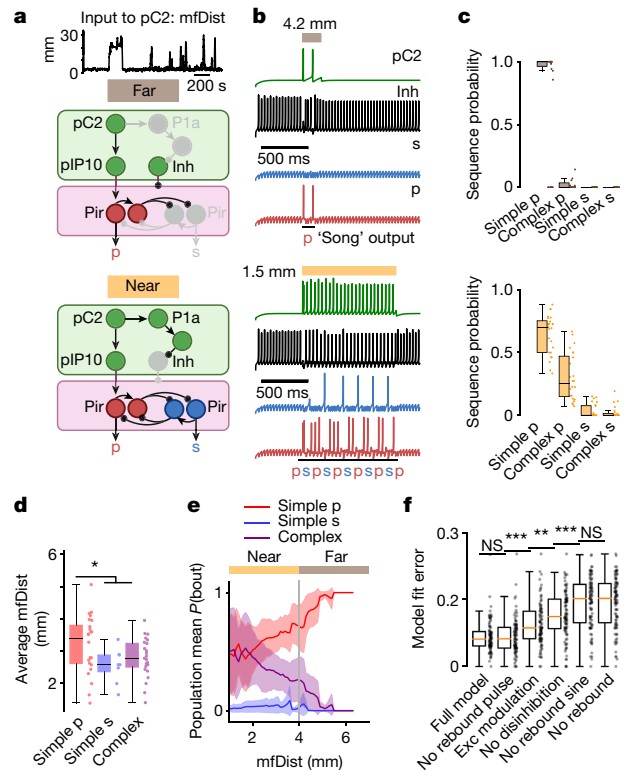

**Fig. 5 | Neural circuit model of context-dependent song patterning. a**, Circuit model for male song patterning far from and near a female. mfDist (top), the only input to the model, enters the circuit via the pC2 node (see Methods), which drives the pulse pathway. Strong input (near the female) additionally disinhibits the VNC rebound circuit, enabling complex song production (alternating activity of the pulse and sine nodes). Grey indicates nodes becoming inactive at far or near conditions. Here, bout termination mainly relies on increases in mfDist (Extended Data Fig. 6a,b), consistent with ref. 3. **b**, Spiking neuronal network of four nodes (pC2, inh, p and s) representing the key computational features of the circuit in **a**, disinhibition, rebound excitability and mutual inhibition, fit to wild-type courtship data (see Methods). Model simulations with brief and weak (top) or long and strong (bottom) input to pC2 (corresponding to mfDist = 4.2 and 1.5 mm) result in either simple ('p') or complex ('psp...') song outputs. **c**, Song statistics for genetic algorithm fits of the model in **b** to song data at far (top) or near (bottom) distance (see Methods; experimental distributions shown in Extended Data Fig. 7k). The model reproduces bout statistics of courting wild-type flies (see Fig. 1d). **d,e**, Average mfDist (**d**) or population-averaged probability (mean ± mean absolute deviation from the mean) at a given mfDist (**e**) of simulated simple pulse, simple sine or complex bouts (models as in **c**) matches observations in courting wild-type flies (see Fig. 1c,f). Vertical grey line in **e** separates near and far contexts. **f**, Fit error (genetic algorithm objective function) for the full model versus models with individual computational features knocked out (see Methods), or disinhibition replaced with an excitatory motif ('exc modulation'; see Methods; Extended Data Fig. 7a–c). For **c–f**, *n* = 24 (**c–e**) and *n* = 93 (**f**) genetic algorithm model fits to song (400 and 200 s each for **c–f**) randomly chosen from *n* = 20 wild-type recordings (biological replicates). For **d,e**, Wilcoxon rank-sum test for equal medians. \**P* < 0.05, \*\**P* < 0.01 and \*\*\**P* < 0.001. For **c,d,f**, central mark indicates the median; the bottom and top edges of the box indicate the 25th and 75th percentiles, respectively. Whiskers extend to 1.5 times the interquartile range away from the box edges.

in vivo patch-clamp recordings of pIP10 and found clear signs of spike-frequency adaptation (Extended Data Fig. 8a–c).

Our circuit model predicts that blocking descending inputs to the core pulse node should strongly reduce the amount of bouts with leading pulse song. To test this prediction, we re-examined published data[13]

with expression of TNT[32] or inward-rectifying potassium channels (Kir2.1)[35] in pIP10 neurons. As both TNT and Kir2.1 prevent chemical synaptic transmission, as expected, the amount of simple pulse-only bouts during courtship with a female was significantly reduced compared with genetic controls (Extended Data Fig. 7g). However, males expressing TNT (but not Kir2.1) in pIP10 produced more sine-leading bouts than controls (Extended Data Fig. 7h), suggesting a potential role for electrical synapses (which remain intact in TNT flies) in mediating sine song generation. Electrical synapses between pIP10 and the inhibitory interneurons of the pulse-rebound circuit (Extended Data Fig. 7j) might help to generate the near-perfect anti-correlation between subsets of TN1 neurons that we observed (Fig. 3c–e and Extended Data Fig. 3a,b).

## Discussion

The ability to alter the sequencing of actions to match the current environmental context is observed across animals and behaviours, including for social interactions[36–38]. Here we provide insights into the underlying mechanisms by focusing on song production in two contexts in *Drosophila melanogaster*: near versus far from a female. Using quantitative behaviour, modelling, broad-range optogenetics, circuit manipulations and neural recordings, we found that simple song (of primarily the pulse mode) is driven by low-level or brief activation of pC2 brain neurons, which drive a pair of pIP10 brain-to-VNC descending neurons. To generate complex bouts, stronger, longer-duration pC2 neuron activity simultaneously drives pIP10 and recruits P1a neurons to functionally disinhibit core circuitry in the VNC, allowing pIP10 descending signals to produce rapid alternations of pulse and sine song. Song alternations are facilitated by combination of mutual inhibition and rebound excitability in pulse-driving and sine-driving neurons of the VNC, allowing for sine song production without the need for excitatory drive. Here, the sensory context, encoded ultimately by acute P1a neural activity, determines which song repertoire (simple pulse or complex) is accessible to descending commands, effectively implementing context dependence via two operational modes of a single circuit[39].

Context dependence of acoustic communication is known in other species, including songbirds[40] and primates[41]; the circuit mechanisms that we have uncovered here may therefore serve as a useful template in investigating those systems at the cellular level. The presence of the female has opposing effects on song variability in flies and birds, species in which females prefer either variable[42] or stereotyped[43] song, respectively. In flies, we showed that female proximity relieves the core song circuit from inhibition to promote song variability (rapid pulse–sine alternations of varying length), whereas in birds, female presence suppresses song variability via direct inhibition of basal ganglia neurons[44].

Context dependence has also been reported for escape responses in noctuid moths, crickets and flies; in the moth, two distinct wing motor patterns (directed turning away from low-intensity ultrasound and power dive to escape high-intensity ultrasound) arise from continuous changes in sensory cues[45], similar to our finding of context-dependent changes in song output. In crickets and flies, context dependence of escape behaviours is achieved via gating of a single ascending interneuron by the flight motor pattern generator[46], or via state-dependent gating of descending neuron activity[47], similar to our proposed role of P1a brain neurons in mediating context-dependent song patterning via functional disinhibition of the VNC circuit.

Relating our results with previous work on song production and patterning in *Drosophila*, we show that first, previous work has suggested that pIP10 neurons drive only the pulse mode of song[4,14]; however, those studies did not explore the broad range of optogenetic activation parameters used here, highlighting the value of varying neural activity levels during behaviour to uncover circuit dynamics.

Second, although our computational model of the song circuit can recapitulate song dynamics using only mfDist as contextual information, previous work has demonstrated that the male's own locomotor speed is also highly predictive of song patterning; however, although we do not yet know where self-motion information enters the song pathway, our model predicts that it should be integrated at the level of pIP10 or downstream, pushing the song pathway towards pulse song production, without engaging the disinhibition arm of the pathway (via P1a neurons) that would lead to sine song production.

Third, previous work uncovered that there are two distinct types of pulse song termed Pfast and Pslow, and that the choice of pulse type depends on distance to the female[13]: males produce Pfast (the louder mode of song) at further distances, switching to Pslow (the softer pulse type) when close. Our data indicate that the relative amount of Pfast and Pslow is ultimately controlled by the activity of brain pC2 neurons (Extended Data Fig. 5d). How VNC neurons[4] coordinate the production of the two pulse types remains to be elucidated, but they must ultimately act via the ps1 motor neuron[5], which has been shown to be required for males to switch from Pslow to Pfast when far from females[13].

Fourth, our study also provides a mechanistic explanation for a previous discovery of two hidden internal states in the male brain underlying song production, termed 'close' and 'chasing'[48]. Our work suggests that the P1a disinhibition arm of the pathway underlies the difference in these two states; in the close state, in which sine song dominates and males are close to females, the P1a disinhibition circuit is engaged and sensory-driven pIP10 activity drives pulse–sine complex bouts. In the chasing state, in which males are farther from females and moving faster, the P1a disinhibition circuit is not engaged and pIP10 activity drives primarily pulse-only simple bouts. This interpretation explains the observation that males continually toggle between close and chasing states throughout courtship, that close-state durations are longer than chasing-state durations, and why activation of pIP10 neurons in the presence of a female paradoxically both drove pulse song and pushed males into a state (close) that promoted sine song production[48].

Last, our work also adds to the range of roles of the P1a neural cluster in modulating social behaviour at different timescales[24,26,28,34,49]. Although previous work emphasized the role of P1a in gating and sustaining male courtship behaviour by controlling a minutes-long arousal state, here we identified an acute role for P1a in shaping behaviour, similar to ref. 34. We showed that recent activation (timescales of milliseconds to seconds) of P1a neurons unlocks the potential for males to produce complex song (whereas separately, P1a neurons promote persistent singing). This may explain why males continually tap females throughout courtship: not only to maintain arousal but also to gate the production of long (complex) song bouts preferred by the female[42].

Our computational model of the song circuit reveals that few key features (mutual inhibition, rebound excitability and disinhibition) are sufficient, in combination with excitatory drive from fluctuating contextual cues, to recapitulate natural song dynamics (Fig. 5). These same features have been shown to contribute to motor pattern generation in both invertebrates and vertebrates[50–53], although they are combined in new ways within the male song circuit. Such a minimalist circuit design both offers a simple control mechanism for reacting to rapid changes in sensory context, and requires only few developmental changes to either derive this circuit from a unisex template[16] or alter the circuit to generate new song types in other species[14]. Yet, we do not rule out the existence of redundant or additional pathways, including descending connections to sine-driving neurons in the VNC. Although emerging connectomes for the male brain and VNC[19] will reveal additional neurons and circuit elements that shape male song patterning (for example, uncovering the circuits that mediate functional disinhibition downstream of P1a excitatory neurons or the detailed connectivity between VNC neurons downstream of pIP10), our study highlights how hypotheses about circuit function can be tested via quantitative analysis and modelling of natural, context-dependent behaviour.

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

## Methods

### Fly strains and rearing
See Supplementary Tables 1 and 2.

### Behavioural apparatus
Behavioural experiments were performed in two custom-made circular chambers (modified from ref. 13) within black acrylic enclosures. Ambient light was provided through an LED pad inside each enclosure (3.5″ × 6″ white, Metaphase Technologies). For each chamber, video was recorded at 60 fps (FLIR Blackfly S Mono 1.3 MP USB3 Vision ON Semi PYTHON 1300, BFS-U3-13Y3M-C, with TechSpec 25 mm C Series VIS-NIR fixed focal length lens) using the Motif recording system and API (loopbio GmbH), run via Python 2.7, and using infrared illumination of around $22\,\mu W\,mm^{-2}$ (Advanced Illumination High Performance Bright Field Ring Light, 6.0″ O.D., wash down, IR LEDs, iC2, flying leads) and an infrared bandpass filter to block the red light used for optogenetics (Thorlabs premium bandpass filter; diameter 25 mm, central wavelength = 850 nm, full width at half maximum = 10 nm). Sound was recorded at 10 kHz from 16 particle velocity microphones (Knowles NR-23158-000) tiling the floor of each chamber. Microphones were hand-painted with IR absorbing dye to limit reflection artefacts in recorded videos (Epolin Spectre 160). Temperature was monitored inside each chamber using an analogue thermosensor (Adafruit TMP36).

### Optogenetics
Flies were kept for 3–5 days on regular fly food or food supplemented with all-*trans* retinal (ATR) at 1 ml ATR solution (100 mM in 95% ethanol) per 100 ml of food. ATR-fed flies were reared in the dark. CsChrimson was activated at $1$–$205\,\mu W\,mm^{-2}$, using 627-nm LEDs (Luxeon Star).

### Behavioural assays
For all behavioural experiments, virgin males and virgin females were used 3–5 days after eclosion. Experiments were started within 120 min of the incubator lights turning on. Males and females were single and group housed, respectively. Flies were gently loaded into the behavioural chamber before an experiment, using a custom-made aspirator. Females were placed first for paired experiments. Chamber lids were painted with Sigmacote (SL2, Sigma-Aldrich) to prevent flies from walking on the ceiling, and kept under a fume hood to dry for at least 50 min before an experiment. Videos were manually scored for copulation. Data beyond copulation were excluded from analysis, unless statistical biases required exclusion of the entire recording.

**Free courtship.** Free courtship recordings were performed for 30 min, as previously described[3].

**Optogenetic neural activation.** A fixed stimulus frequency of 1/8 Hz was used for optogenetic neural activation. Stimulus irradiance could take four distinct values (0, 1, 25 and $205\,\mu W\,mm^{-2}$), spanning three orders of magnitude, and stimulus duty cycle could take five distinct values (1/64, 1/32, 1/16, 1/8, and 2/8), and both irradiance and duty cycle were combined in a full factorial design, resulting in 16 distinct blocks (pooling blocks with zero irradiance) that were presented in pseudo-randomized order for 120 s each.

### Offline song segmentation
For subsequent offline analysis, song was segmented as previously described[11,13], using a modified sine detection parameter to account for different acoustics in the setup used here (Params.pval = $1 \times 10^{-7}$). For a given recording, the output of the song segmentation algorithm included information about the start and end of each bout and each sine train, as well as the centre of each detected pulse, and a snippet of noise not including song. To reduce the risk of contaminating bout statistics with artificially split bouts due to low amplitude of sine song (the softer song mode), we excluded all bouts containing sine song with amplitude below a chosen signal-to-noise (SNR) threshold. Specifically, we estimated the noise amplitude using the noise segment that is automatically detected and returned by the song segmentation software (thus not containing song), by first reducing the 16-dimensional (for 16 microphones) noise segment to a one-dimensional vector by storing the noise value of the loudest microphone at each time point, and then defining noise amplitude as the 99th percentile of the absolute value of the one-dimensional noise vector. Sine amplitude was calculated similarly, such that the SNR for a given sine bout was the ratio of the sine amplitude and the noise amplitude. We excluded bouts containing sine song with an SNR below 1.3 from further analysis. Furthermore, the song segmenter occasionally split individual sine trains, due to intermittent noise. Uncorrected, this could, for example, split a 'psp' bout into one 'ps' and one 'sp' bout very close in time. This allowed us to use a simple temporal threshold to merge such bouts if the inter-bout interval was below 0.5 s. The segmentation software is freely available at https://github.com/murthylab/MurthyLab_FlySongSegmenter.

### Tracking
Male and female poses (locations of head, thorax, and left and right wing tip) were automatically estimated and tracked, and manually proofread for all videos using SLEAP[17] (sleap.ai).

### Song behaviour analysis
**Song probabilities.** For experiments with open-loop optogenetic neural activation, the probability for a male to sing pulse or sine song at any point in time during a trial of a given stimulus block was computed as the fraction of trials containing pulse or sine song. For analyses separating song probabilities into far and near contexts, the average mfDist within a trial was thresholded to assign the trial to one of the two contexts. Song probabilities for each context were then calculated using only those trials assigned to that context.

**Song sequences.** Song segmentation provided information about the start and end of each bout, and all pulse and sine events within a bout, allowing to assign each bout a label describing the sequence of contained pulse and sine trains ('p' for a bout containing only pulse song, 'spspspsp' for a bout starting with sine song followed by several alternations between pulse and sine). For statistics, we reduced the amount of different bout types by abbreviating all bouts with one or more song alternations as 'ps…' or 'sp…' and referred to these as 'complex p' or 'complex s'. 'Acute' and 'persistent' bouts were defined as bouts starting during a stimulus or after stimulus offset, respectively. Rebound song was defined as song that started after stimulus offset, in a bout that started during a stimulus (for example, if the initial pulse train in a ps bout starts during a stimulus, but the following sine train starts after stimulus offset, that is considered rebound sine).

**Tap detector model.** The tap detector model was constructed using a convolutional neural network. The convolutional neural network consisted of two two-dimensional convolutional layers followed by two fully connected layers. The two convolutional layers had 32 output and 64 output channels, respectively, a kernel size of 5 and a stride of 1. The outputs of each convolutional layer were passed through a rectified linear unit nonlinearity and a two-dimensional max pooling layer with a kernel size of two and stride of two. The first fully connected layer had 53,824 input and 32 output features followed by a rectified linear unit nonlinearity, and the second fully connected layer had 32 input and 2 output features corresponding to scores for a tap or non-tap. The model was trained using the AdamW algorithm for 100 epochs with a batch size of 16 and a learning rate of 0.0001. The model was constructed and trained using the PyTorch library[54].

To train the convolutional neural network, video frames (size 128 × 128) of courting flies centred on the male were manually labelled as a tap or non-tap event using a custom graphical user interface. Ten videos were used for creating the tap dataset, with 12,606 manual annotations total. Of these annotated frames, 70% were used for training and 30% were held out for model validation. Receiver-operating characteristic analysis was performed on held out data to determine the relationship between model recall (true-positive rate) and fallout (false-positive rate) as a function of tap detection threshold.

**Tap rate analysis.** Tap rate was quantified as the number of taps within a song bout, divided by the duration of the bout (to compare with time before a bout, we used the number of taps within an equally sized window preceding the bout, divided by bout duration).

**Tap-based model of P1a neural activity.** We convolved the binary output of the tap detection network (tap = 1/no tap = 0, using a threshold on tap probability of $P(\text{tap}) \geq 0.9$) with the known calcium fluorescence of P1 neurons in response to a single tap of the female abdomen (tap-triggered average[27]) to get an estimate of P1a neural activity in freely courting males on a moment-to-moment basis. We deconvolved the estimated calcium fluorescence signal with a kernel of the GCaMP6s calcium response (time constant of 2.6 s)[55] to obtain an estimate of P1a rate, which we used for further analysis.

**Bout-triggered analysis of tap rate.** For a given recording, the binary tap detector output at video resolution was first upsampled to audio resolution, using the camera trigger signal for synchronization. For each song bout with leading pulse song (simple p or complex ps…), the number of detected taps occurring during the bout, $n_{\text{during}}$, was counted, and this was divided by the duration of the bout, $B$, to produce tap rate during the bout, $R_{\text{during}} = n_{\text{during}}/B$. As a control, the tap rate before the bout was computed as the number of taps occurring in an equally sized time window $B$ immediately preceding the bout, $R_{\text{before}} = n_{\text{before}}/B$. Tap rates were averaged (using the mean) per animal across simple and complex bouts and used for further analysis.

**Generalized linear model analysis.** To estimate the relative predictive power of different sensory features on the choice of bout (here, complex versus simple p), we used the generalized linear modelling framework with a sparse before penalize non-predictive history weights, as previously described[3,56]. In brief, ten sensory features (male and female forward velocity (mFV and fFV), lateral speed (mLS and fLS), rotational speed (mRS and fRS), the angle of the male (female) thorax relative to the female (male) body axis (fmAngle and mfAngle), the distance between the male and female thorax (mfDist), and the instantaneous rate of P1a neurons estimated from detected taps (P1 rate)) were first smoothed using a moving average filter with a width of 20 video frames (0.33 s). Then, 21 uniformly distributed samples were extracted from the smoothed features within the 5 s of history leading up to the end of the first pulse train of each bout with the leading pulse song (for simple pulse bouts, this corresponded to the end of the bout). Extracted features were z-scored per feature, to account for different feature dimensions and scales. Inputs to the generalized linear model (GLM) were the transformed features and a corresponding binary vector indicating whether a given feature history corresponded to a simple or complex pulse bout, and outputs were estimated filters for each feature (providing information on which dynamics in the feature, within the history window, were most predictive for bout type) and the relative deviance reduction (a measure of model performance). To estimate fit robustness, we repeated GLM fitting 51 times, each time using 70% of the input data (sampled randomly without replacement). For each feature, the mean across fits and the mean absolute deviation from the mean across fits were calculated and used for display.

## Two-photon functional imaging

We imaged the activity of Dsx+ cells in the VNC following pIP10 optogenetic activation using a custom-built two-photon laser scanning microscope[57,58]. Virgin male flies (5–8 days old) were mounted and dissected as previously described[59], with minor differences. In brief, we positioned the fly ventral head and thorax side facing up to the underside of the dissection chamber, exposing both the ventral side of the central brain and the ventral side of the VNC. From the head, we removed the proboscis, surrounding cuticle, air sacks, tracheas, and additional fat or soft tissue. From the VNC, we removed thoracic tissue ventral to the VNC (for example, legs and cuticle), exposing the first and second segments of the VNC. Perfusion saline was continuously delivered to the meniscus between the objective and the dissection chamber throughout the experiment. We imaged Dsx+ TN1 cells (one hemisphere at a time), located in the ventral side of the second segment of the VNC. Specifically, although we used flies that express the calcium indicator GCaMP6s in all Dsx+ neurons, we only imaged the prothoracic and mesothoracic neuromeres, and the accessory mesothoracic neuropil of the VNC. Together, these regions house the Pr1–3, Pr4, Ms1–3 and TN1 cluster of neurons[60], whose somas have distinct and identifiable locations. We manually segmented somas from these regions that, based on their anatomical location, were unambiguously identified as TN1 neurons. TN1 can be distinguished from dPR1 (which belongs to the Pr1–3 cluster) based on the position of the somas in the anteroposterior axis. Similarly, TN1 can be readily distinguished from its neighbouring clusters (Pr4 and Ms1–3) based on its more lateral and ventral location relative to the accessory mesothoracic neuropil, as well as the smaller size of its somas. Our manual segmentation was based on these criteria rather than on neural responses. We recorded 3–4 subvolumes of approximately $70 \times 70 \times 20\ \mu m^3$ at a speed of 1 Hz ($0.3 \times 0.3 \times 2\ \mu m^3$ to $0.4 \times 0.4 \times 2\ \mu m^3$ voxel size), covering the full ventral-to-dorsal extent of the TN1 cluster (~70 μm). Volumetric data were collected using ScanImage 2017 and processed using FlyCalMan[58] (https://github.com/murthylab/FlyCalMan) via Matlab 2018b. In brief, volumetric time series of the GCaMP6s signal was motion corrected in the $xyz$ axes using the NoRMCorre algorithm[61], and temporally resampled to correct for different slice timing across planes of the same volume and to align timestamps of volumes relative to the start of the optogenetic stimulation (linear interpolation). Subvolumes consecutively recorded along the $z$ axis were stitched along the $z$ axis using NoRMCorre. Dsx+ TN1 somas were segmented by using the constrained non-negative matrix factorization algorithm to obtain temporal traces and spatial footprints of each soma as implemented in CaImAn[58,62] (the initial number and $xyz$ location of all TN1 somas were manually pre-defined). For pIP10 activation, we used an optogenetic protocol that combined long stimuli driving strong pulse and weaker rebound sine when activating pIP10 in solitary, freely behaving males (Fig. 2c,e). Specifically, we used a stimulus of 2 s ON (at 13 μW mm$^{-2}$ irradiance) and 2 s OFF repeated four times to maximize the magnitude of evoked GCaMP responses. Imaging started 10 s before stimulus onset, where baseline activity was measured, and lasted 10 s after stimulus offset.

## Neural circuit model of song bout statistics

Network simulations were performed using the Brian2 package[63] with Python3. Individual neurons were defined as variants of the Izhikevich model[64] with known spiking properties (such as rebound or tonic spiking) that matched experimental predictions. In brief, the neuronal membrane potential $v$ was modelled via three ordinary differential equations:

$$\frac{\mathrm{d}v}{\mathrm{d}t} = 0.04v^2 + 5v + 140 - u + I + \frac{g_e + g_i}{\tau_{\text{syn}}} \tag{1}$$

$$\frac{\mathrm{d}u}{\mathrm{d}t} = a(bv - u) \tag{2}$$

$$\frac{\mathrm{d}g_{e,i}}{\mathrm{d}t} = -\frac{g_{e,i}}{\tau_{e,i}}, \tag{3}$$

with the membrane recovery variable $u$, the timescale $a$ and the sensitivity $b$ to subthreshold fluctuations of the membrane potential of the recovery variable, and the input current $I$. $g_e$ and $g_i$ are excitatory and inhibitory conductances, and $\tau_{syn}$ is the synaptic time constant. Whenever the membrane potential reached 30 mV, this was considered an action potential and the membrane variables were reset via

$$v = c, \qquad u = u + \mathrm{d}. \tag{4}$$

The full song circuit model comprised four Izhikevich neurons, termed p (pulse), s (sine), pC2 and inh. Parameters $a$, $b$, $c$ and $d$ were chosen to enable post-inhibitory rebound dynamics for the pulse and sine node, and tonic spiking for the pC2 and inh nodes (Supplementary Table 3). Inhibitory connections were defined mutually between pulse and sine, from inh to both pulse and sine, and from pC2 to inh. A single excitatory connection was defined from pC2 to pulse. Together, pC2 provided excitatory input to the pulse node and functional inhibition to the inh node, mimicking the direct pulse pathway from pC2 via pIP10 to the VNC, and the proposed disinhibitory pathway from pC2 via P1a (activated for short mfDist and strong input to pC2; Fig. 5a), respectively. For each spike in a presynaptic neuron, the synaptic conductance $g_{e,i}$ was incremented by $w_{e,i}$. $w_{e,i}$ were free parameters that were fit during genetic algorithm optimization. The remaining free parameters were the amount of tonic input current into the inh node ($I_{tonic}$, regulating the amount of tonic inhibition onto the core pulse–sine circuit, mimicking the male's default, unaroused, state), and a multiplicative factor $I_e$ that controlled the gain of the sensory input current into pC2. The sensory input current into pC2 was the mfDist during a given recording of wild-type courtship, subjected to nonlinear (NL) transformation via

$$I_{pC2} = I_e \cdot \mathrm{NL}(\mathrm{mfDist}), \tag{5}$$

$$\mathrm{NL}(\mathrm{mfDist}) = \frac{\alpha}{1 + \exp(-\beta \cdot (x_0 - \mathrm{mfDist}))}, \tag{6}$$

to facilitate strong/weak input current to pC2 at short/large distance. Numerical simulations of the network were performed using Euler integration, and spike times of each node were recorded for further analysis. Specifically, 'song sequences' of the model were defined based on the activity of the pulse and sine node, such that a coherent spike train of one node that was at least 300 ms separated from the next spike of the other node was considered a simple bout, whereas alternating activity of the two nodes within 300 ms was considered a complex bout. This simplifying assumption allowed us to fit the model to experimental song statistics, using genetic algorithm optimization (see below). We did not explicitly model a mechanism to control bout duration, and we expect that additional features such as recurrent excitation in the pulse and sine nodes are required to sustain pulse or sine trains. All model parameters are specified in Supplementary Table 4.

**Genetic algorithm optimization.** The distribution of model bout types in response to a given naturalistic stimulus was directly comparable with the actual distribution of male song bouts corresponding to the sensory stimulus, which we exploited to fit the four free parameters of the model (a scalar gain factor for the input to the pC2 node, the strength of a constant input current to the inh node, and one global weight each for all excitatory and inhibitory connections) to the

experimental data. Specifically, we used genetic algorithm optimization (the geneticalgorithm package in Python, https://pypi.org/project/geneticalgorithm/) to minimize the root-mean-squared difference between the experimental and simulated bout distribution (using six bout types, 'p', 'ps', 'psp…', 's', 'sp' and 'sps…', to provide more information to the algorithm than when using the four categories ultimately used for analysis; this led to slightly better model fits), as well as the absolute difference between the number of experimental and simulated bouts ($\Delta n_{bout}$), via the objective function root-mean-squared difference + $0.1 \cdot \Delta n_{bout}$ (see Supplementary Table 4 for optimization parameters and ranges). The relative scaling of the two objectives was chosen to prioritize reproducing the bout distribution over the number of bouts. All genetic algorithm parameters are specified in Supplementary Table 4. Four hundred-second pieces of song data, randomly chosen from all 20 wild-type recordings with at least 10% of song bouts produced far from the female (mfDist > 4 mm), were used as input to the genetic algorithm.

**Knockout simulations.** To test the relevance of different computational features of the circuit model, we compared genetic algorithm fit performance for the full model (here using 200-s song snippets, randomly chosen from all wild-type recordings) to fit performance for versions of the model with individual computational features 'knocked out' or replaced. Specifically, although in the full model both the p and the s nodes were rebound excitable (by choosing the appropriate values for parameters $a$, $b$, $c$ and $d$ (see Supplementary Table 3), rebound excitability was knocked out in the pulse (no rebound pulse), sine (no rebound sine) or both nodes (no rebound) by adjusting parameters $a$, $b$, $c$ and $d$ (to turn these nodes from 'rebound spiking' into 'tonic spiking'; see Supplementary Table 3). Disinhibition was knocked out by removing the inhibitory synapses of the inh node onto the pulse and sine nodes. To compare fits to experimental data for the default model comprising disinhibition and a model comprising excitatory modulation of the pulse and sine nodes, we replaced the inhibitory weights onto and from the inh node with excitatory weights, forming an excitatory node ('exc') for which we removed the tonic input that was present for the inh node in the disinhibitory model.

### Irradiance measurements
Irradiance levels reported for optogenetic neural activation in freely behaving flies were measured (using a Thorlabs PM100D power meter) at the centre of the experimental chamber, with the chamber lid in place. Two identical experimental setups were used for behavioural experiments, and irradiance levels were calibrated to have uniform voltage-to-irradiance conversion across setups.

Irradiance reported for optogenetic stimuli during two-photon calcium imaging was measured (also using a Thorlabs PM100D power meter) at approximately the level of the preparation (after the objective).

### Statistics
Statistical analyses were performed either in Matlab 2019a or Python 3.7. The two-sided Wilcoxon rank-sum test (Mann–Whitney $U$-test) for equal medians was used for statistical group comparisons unless noted otherwise. Error bars indicate mean ± mean absolute deviation from the mean unless otherwise specified. Sample sizes were not predetermined but are similar to those reported in previous publications[13,34]. Experimenters were not blinded to the conditions of the experiments during data collection and analysis. Experimental groups were defined based on genotype, and data acquisition was randomized with respect to different genotypes. All attempts at replication were successful. For box plots, the central mark indicates the median, the bottom and top edges of the box indicate the 25th and 75th percentiles, respectively. Whiskers extend to 1.5 times the interquartile range away from the box edges.

## Reporting summary

Further information on research design is available in the Nature Portfolio Reporting Summary linked to this article.

## Data availability

Data are available on request from the corresponding author. Source data are provided with this paper.

## Code availability

The circuit model simulation code is available at github.com/murthylab.

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

**Acknowledgements** We thank J. Goldberg, A. Falkner, J. Pillow, I. Witten, J. Clemens, S. Ahmed and B. Mimica for comments on the manuscript; G. Guan for technical assistance; and M. Choi for assistance with real-time song segmentation and development of closed-loop optogenetic protocols. Schematics in Fig. 3a and Extended Data Fig. 5h were created using BioRender (https://biorender.com). We acknowledge funding from the German Research Foundation (DFG Forschungsstipendium RO 5787/1-1 and RO 5787/2-1) to F.A.R., the Sloan-Swartz Foundation to R.P., the Howard Hughes Medical Institute, via a Faculty Scholar Award to M.M., the NIH BRAIN Initiative via NS104899 to M.M., and an NIH NINDS R35 Research Program Award to M.M.

**Author contributions** F.A.R., E.C.I., K.T., D.A.P., X.L. and M.J.A. collected the data. F.A.R., R.P. and M.J.A. developed the model and code. F.A.R. analysed the data and generated the figures. F.A.R. and M.M. wrote the manuscript. F.A.R. and M.M. conceptualized the study.

**Competing interests** The authors declare no competing interests.

**Additional information**
**Correspondence and requests for materials** should be addressed to Mala Murthy.

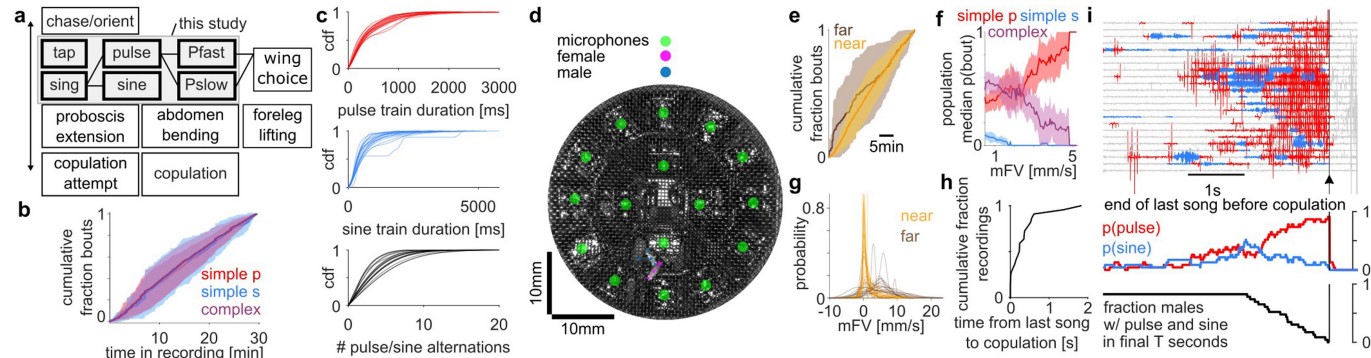

**Extended Data Fig. 1 | Context-dependence of song sequencing in *Drosophila melanogaster* males (supplement to Fig. 1). a**, Male behaviors during courtship (modified from[65], including aspects from[13,66]), with those focused in the present study highlighted in the grey box. **b**, Cumulative fraction of simple pulse (red), simple sine (blue), or complex (purple) bouts over time in recording for $n$ = 20 wild-type male-female pairs (biological replicates). **c**, Distribution of pulse train duration, sine train duration, and the number of pulse-sine alternations in male song of $n$ = 20 wild-type male-female pairs (biological replicates). **d** Chamber for behavioral experiments. Male courtship song was recorded using 16 microphones (green) tiling the chamber floor. Female (magenta) and male (blue) fly pose and tracks were estimated using SLEAP[17]. **e** Cumulative fraction of far (brown) and near (yellow) bouts over time in recording for $n$ = 20 wild-type male-female pairs (biological replicates). **f**, Population-averaged probability to sing simple pulse, simple sine, or complex bouts relative to male forward velocity (mFV). Color code as in (**a**). **g**, Distribution of mFV near (yellow) and far (brown) from the female. **h**, The majority (91%) of final bouts (the last song bout prior to copulation) occur within 0.6 seconds preceding copulation. **i**, The majority (59%) of bouts immediately preceding copulation are complex ($n$ = 23 wild-type pairs with copulation within a 20 minute recording). Song bouts are aligned to bout end. Time-resolved probability of pulse (red) and sine song (blue) (shown below song traces) rises prior to copulation. Black curve at the bottom shows the fraction of males that sing both pulse and sine song in the time prior to copulation. 80% of males sang both song modes within the final 1.5 seconds of song before copulation, suggesting complex bouts facilitate mating. **b,e**, mean ± mean absolute deviation from the mean. **b,c,f,g**, $n$ = 20 recordings of male-female pairs (biological replicates).

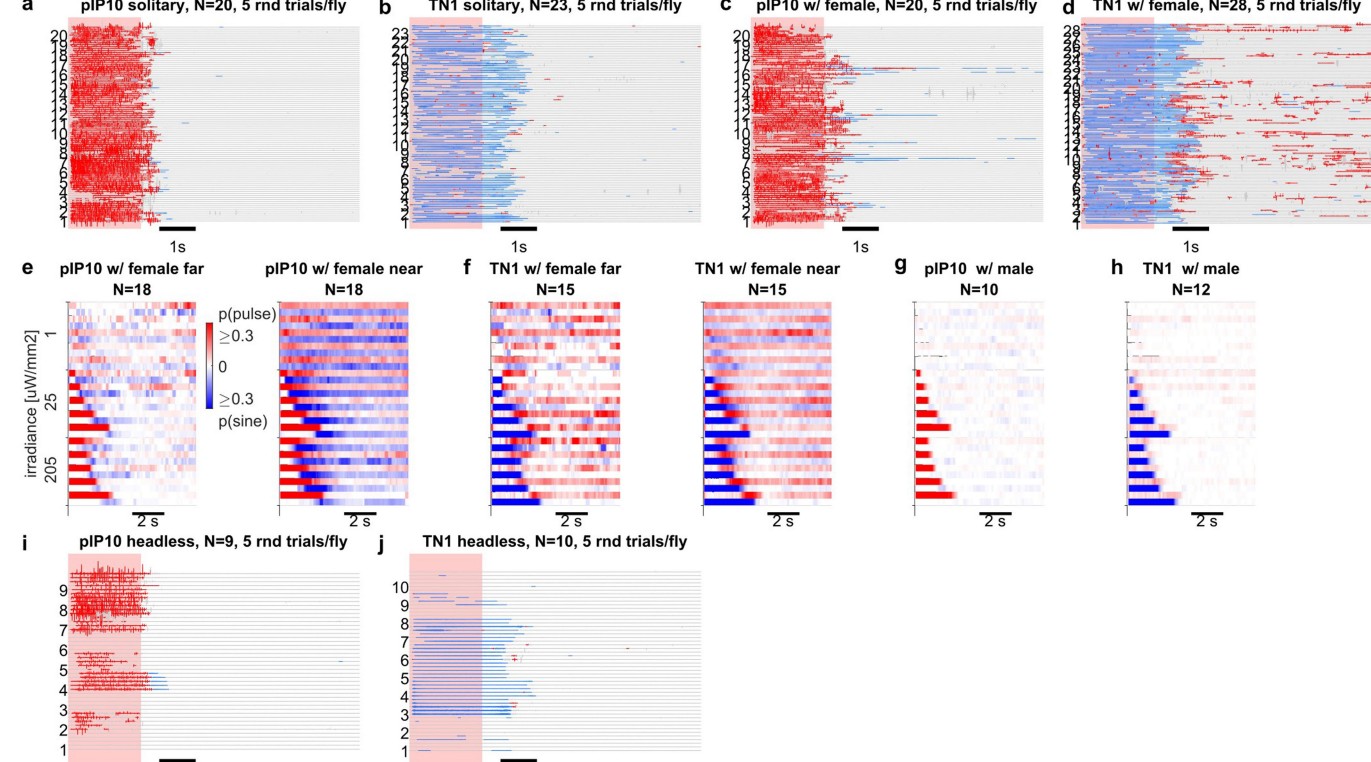

**Extended Data Fig. 2 | Reciprocal interactions between pulse- and sine-producing neurons (supplement to Fig. 2). a**, Example raw song responses drawn from $n$ = 20 solitary pIP10 > CsChrimson males (biological replicates) with a single type of optogenetic stimulus (205uW/mm2 on for 2s per 8s trial). For every recording, five out of 15 trials were randomly chosen for display. Numbers on y-axis indicate recording. Color code: red - pulse song, blue - sine song, grey - silence, pink - optogenetic stimulus. **b**, Example raw song responses drawn from $n$ = 23 solitary TN1 > CsChrimson males (biological replicates) to the same stimulus type shown in **a**). **c**, Example raw song responses frawn from $n$ = 20 pIP10 > CsChrimson males (biological replicates), paired with a wild-type female, to the same stimulus type shown in **a**). **d**, Example raw song responses drawn from $n$ = 28 TN1 > CsChrimson males (biological replicates), paired with a wild-type female, to the same stimulus type shown in **a**). **e**, Population-averaged

song responses of $n$ = 18 pIP10 > CsChrimson males (biological replicates) paired with a wild-type female as shown in Fig. 2d, but split into instances during which male and female were far or near (as quantified in Fig. 4e,f). **f**, Population-averaged song responses of $n$ = 15 TN1 > CsChrimson males (biological replicates) paired with a wild-type female as shown in Fig. 2i, but split into instances during which male and female were far or near (as quantified in Fig. 4e,f). **g**, Population-averaged song responses of $n$ = 10 pIP10 > CsChrimson males (biological replicates) paired with a wild-type male. **h**, Population-averaged song responses of $n$ = 12 TN1 > CsChrimson males (biological replicates) paired with a wild-type male. **i**, Example raw song responses drawn from $n$ = 9 solitary headless pIP10 > CsChrimson males (biological replicates) to the same stimulus type shown in **a**). **j**, Example raw song responses drawn from $n$ = 10 solitary headless TN1 > CsChrimson males (biological replicates) to the same stimulus type shown in **a**).

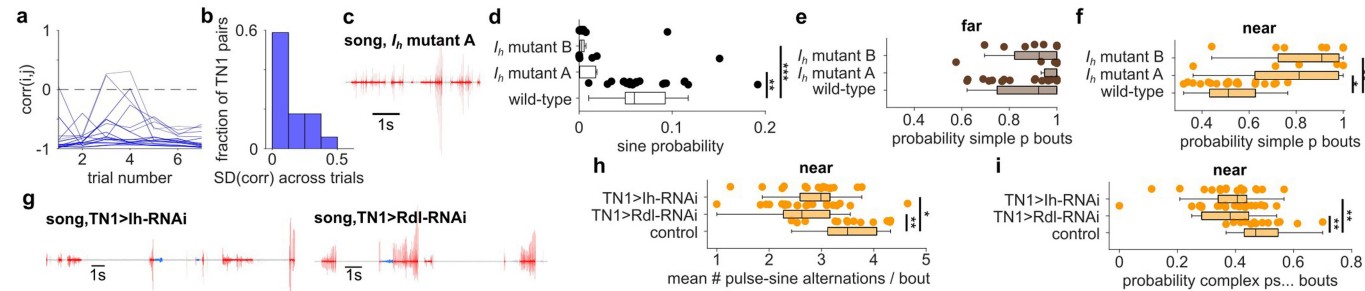

**Extended Data Fig. 3 | Post-inhibitory rebound dynamics in the VNC (supplement to Fig. 3). a**, Anti-correlation between calcium responses of TN1 neuron pairs persists across trials. While Fig. 3c shows the correlation between trial-averaged calcium responses of TN1 neuron pairs in one fly, here we show the correlation between TN1 pairs for individual trials (7 trails, each trial consisting of four optogenetic stimulus presentations and a pause, as shown in Fig. 3b), only for pairs with trial-averaged anticorrelation coefficient below −0.8 ($n = 17$). **b**, Standard deviation (SD) across trials of the correlation coefficients shown in **a**). The majority of anti-correlated TN1 pairs are consistent across trials. **c**, Song of a mutant male systematically lacking $I_h$, courting a wild-type female. **d**, Overall sine probability (fraction of time spent singing sine song in a 30-minute recording) for two different strains of $I_h$ mutants (mutant A, $I_h^{03055}$, and mutant B, $I_h^{01485}$; see Supplementary Table 2) and wild-type males. **e**, Proportion of simple pulse bouts in song of $I_h$ mutants and wild-type males, produced far from (>4mm) a wild-type female. **f**, Proportion of simple pulse bouts in song of $I_h$ mutants and wild-type males, produced near (<4mm) a wild-type female. **g**, Song of males with TN1-specific downregulation of $I_h$ or $Rdl$

(GABA-A receptors). **h**, Mean number of pulse-sine alternations in song of males with TN1-specific downregulation of $I_h$ or $Rdl$ (GABA-A receptors), and genetic controls (see Supplementary Table 2), produced near (<4mm) a wild-type female. **i**, Proportion of complex bouts with leading pulse mode, in song of males with TN1-specific downregulation of $I_h$ or $Rdl$ (GABA-A receptors), and genetic controls (see Supplementary Table 2), produced near (<4mm) a wild-type female. **h-i**, $n = 17$ for TN1 > $I_h$, $n = 20$ for TN1 > $Rdl$, $n = 15$ for genetic controls (all biological replicates). The effect of $I_h$ reduction was modest, possibly because neurons other than TN1 contribute to sine song production, because rebound excitability in TN1 neurons arises from a degenerate set of ion channels that are robust to small perturbations (via knockdown) of $I_h$[67], or because a reduction of $I_h$ channels maintains some rebound excitability through increased channel conductance at stronger hyperpolarization[68]. **d-f**, $n = 7$ for mutant A, $I_h^{03055}$, $n = 9$ for mutant B, $I_h^{01485}$, and $n = 20$ for wild-type males (biological replicates). **d-f,h,i** Wilcoxon rank-sum test for equal medians; *P < 0.05, **P < 0.01, ***P < 0.001; NS, not significant.

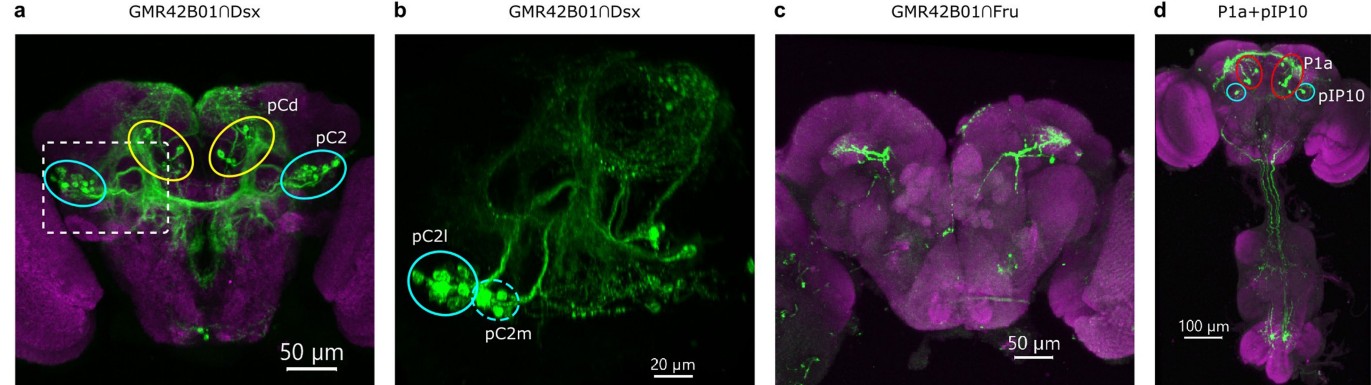

**a** GMR42B01∩Dsx  **b** GMR42B01∩Dsx  **c** GMR42B01∩Fru  **d** P1a+pIP10

**Extended Data Fig. 4 | Neurons targeted in genetic driver lines (supplement to Fig. 4). a**, Male brain expressing CsChrimson.mVenus via GMR42B01 ∩ Dsx (green). Neuropil is labeled with nc82 (magenta). The intersection labels pC2 neurons (circled in blue), as well as 6 pCd-like neurons (circled in yellow). These pCd-like neurons do not express Fru (see panel **c**), and therefore constitute a different subset of neurons than the pCd neurons contributing to persistent male arousal downstream of P1a neurons in[28]. Broad-range optogenetic activation in males using the genetic driver for pCd neurons from[28] produces no song (data not shown). **b**, Zoom of the boxed area in **a**), showing that the pC2 population labeled in the intersection consists of both pC2l neurons (solid circle) and pC2m neurons (dashed circle). **c**, Male brain expressing CsChrimson.mVenus via GMR42B01 ∩ Fru (green). Neuropil is labeled with nc82 (magenta). **d**, Male brain and VNC expressing CsChrimson.mVenus (green) via both P1a and pIP10 drivers. Neuropil is labeled with nc82 (magenta). Cell bodies of P1a are circled in red. Cell bodies of pIP10 are circled in cyan (see Supplementary Table 2 for full genotypes).

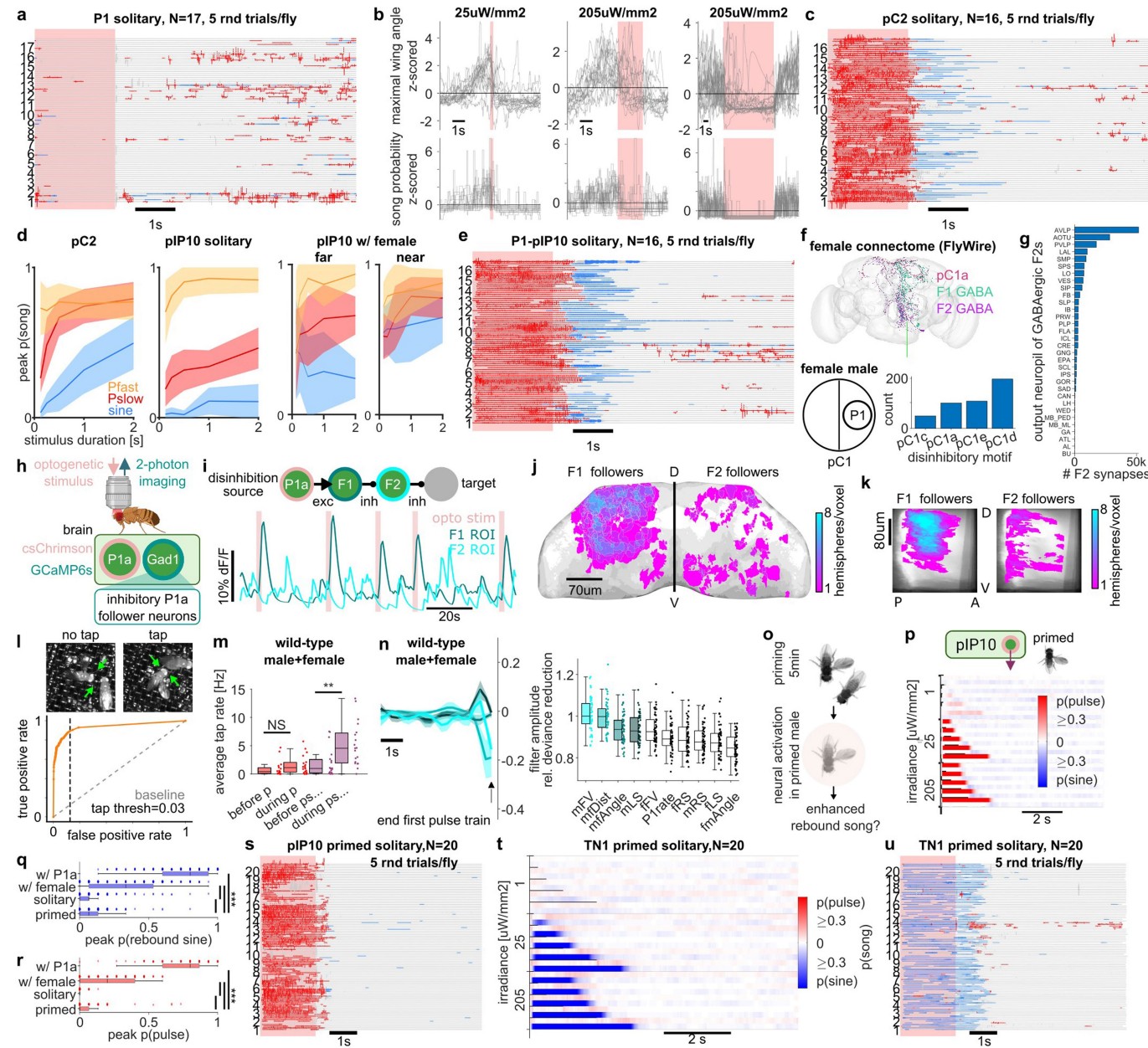

**Extended Data Fig. 5** | See next page for caption.

**Extended Data Fig. 5 | Sensory feedback, disinhibition, and P1a neuron priming in the generation of complex song bouts (supplement to Fig. 4).**
**a**, Example raw song responses drawn from $n = 17$ solitary P1a > CsChrimson males (biological replicates) to a single type of optogenetic stimulus (205uW/mm2 on for 2s per 8s trial). For every recording, five out of 15 trials were randomly chosen for display. Numbers on y-axis indicate recording. Color code: red - pulse song, blue - sine song, grey - silence, pink - stimulus. **b**, Z-scored maximal wing angle (top) and probability to sing (bottom) of solitary males around optogenetic activation of P1a for three different stimuli (25 and 205 uW/mm2 for 250 ms and 2s, respectively, during 8s trials in $n = 17$ biological replicates, and 205 uW/mm2 for 10s during 100s trials in $n = 20$ biological replicates). Each line in the top row corresponds to the mean across trials. **c**, Example raw song responses drawn from $n = 16$ solitary pC2 > CsChrimson males (biological replicates) to the same stimulus type shown in **a**). **d**, Peak probability of two types of pulse song termed Pfast and Pslow (orange and red[13]) and sine song (blue) as a function of stimulus duration for intermediate-irradiance activation (25uW/mm2) of pC2 or pIP10 in solitary males, or pIP10 in males far or near from a wild-type female ($n = 16/20/20$ biological replicates). **e**, Example raw song responses drawn from $n = 16$ solitary P1a-pIP10 > CsChrimson males (biological replicates) to the same stimulus type shown in **a**). **f**, Left: P1 neurons constitute a male-specific subset of pC1 neurons[25,57]. Top right: disinhibitory circuit motif (an inhibitory 'F1' follower neuron inhibiting another 'F2' follower neuron) postsynaptic to an excitatory (cholinergic) neuron of the pC1a subset, identified in public female connectome data, using FlyWire[69,70]. Bottom right: Number of GABAergic disinhibitory motifs postsynaptic to neurons of the pC1 subtypes a-d, detected in the female connectome. **g**, Output neuropils of F2 follower neurons for all disinhibitory motifs in (**f**), sorted by the number of output synapses. The majority of output synapses target the anterior ventrolateral protocerebrum (AVLP), the anterior optic tubercle (AOTU), and the posterior ventrolateral protocerebrum (PVLP). **h**, Two-photon calcium imaging from GABAergic (Gad1+) brain neurons combined with optogenetic activation of P1a brain neurons (see Supplementary Methods for details; see Supplementary Table 2 for genotypes). Schematic in **h** was created using BioRender (https://biorender.com). **i** Example Gad1 calcium responses for two regions of interest (ROIs) showing activity locked to stimulation ('opto stim') of P1a ('F1 ROI') or suppressed activity during F1 activity ('F2 ROI'), as expected for neurons forming a disinhibitory motif postsynaptic to P1a (schematic at top). **j**, Anatomical distribution along the dorsal-ventral (D-V) axis of ($n = 262$) F1 and ($n = 75$) F2 follower ROIs (see **i**) recorded in two hemispheres, but collapsed to the left/right hemisphere respectively for visualization. **k**, Anatomical distribution of the F1 and F2

follower ROIs shown in **j**, across a sagittal slice of the brain. **l**, Tap-detector model performance. (Top) Example of non-tap (left) and tap (right) events. Green arrows indicate the position of male foreleg tarsi. (Bottom) Receiver operator characteristic (ROC) curve for model after 100 epochs of training (orange points). Each point corresponds to a different tap probability threshold. Area under the ROC curve (AUC) is used as an evaluation metric - an ideal model would have an AUC of 1. Performance of a null model (gray diagonal line) is included for comparison. **m**, Average tap rate before and during simple and complex pulse bouts, for $n = 20$ wild-type male-female pairs (biological replicates; analog to Fig. 4l). **n**, A generalized linear model (GLM) to predict complex vs. simple pulse bout production based on the history of sensory features prior to the end of the first pulse train in each (ps… complex or p simple) bout in $n = 51$ random samples from $n = 20$ biological replicate recordings of wild-type male-female pairs (analog to Fig. 4n,o). Sensory features are ranked by their predictive power, and GLM filters are shown for the four most predictive features. **o**, To test for effects of persistent male arousal on optogenetically driven song, males were primed (allowed to court a virgin wild-type female) for 5 minutes preceding optogenetic activation. **p**, Song probabilities for optogenetic activation of pIP10 neurons in solitary males that were primed. $n = 19$ biological replicates. **q**, Comparison of peak rebound sine probability for optogenetic activation at intermediate and strong irradiance (25 and 205uW/mm2) of pIP10 in primed, solitary, female-paired, or P1a-coactivated males. **r**, Comparison of peak pulse probability for optogenetic activation at lowest irradiance (1uW/mm2) of pIP10 in groups identical to those in (**q**). **s**, Example raw song responses drawn from $n = 20$ solitary pIP10 > CsChrimson males (biological replicates) to the same stimulus type shown in **a**). Males were primed (allowed to court a virgin wild-type female, to induce male courtship state) for five minutes prior to the start of the optogenetic stimulus protocol. **t**, Population-averaged song responses of $n = 20$ primed solitary TN1 > CsChrimson males (biological replicates). **u**, Raw song responses of $n = 20$ primed solitary TN1 > CsChrimson males (biological replicates) to the same stimulus type shown in **a**). **j**,**k**, $n = 4$ biological replicate animals. **p**,**r**, Simple pulse song was induced in a fraction of primed males even for the weakest levels of activation, in contrast to males subject to identical stimulation without priming, suggesting that male arousal modulates the excitability of pIP10 neurons at the timescale of minutes but without promoting complex song (compare Fig. 2c). **q**,**r**, $n = 19/20/20/16$ biological replicates for activation of pIP10 in primed, solitary, female-paired, or P1a-coactivated males. **m**,**q**,**r**, Wilcoxon rank-sum test for equal medians; *P < 0.05, **P < 0.01, ***P < 0.001, 1; NS, not significant.

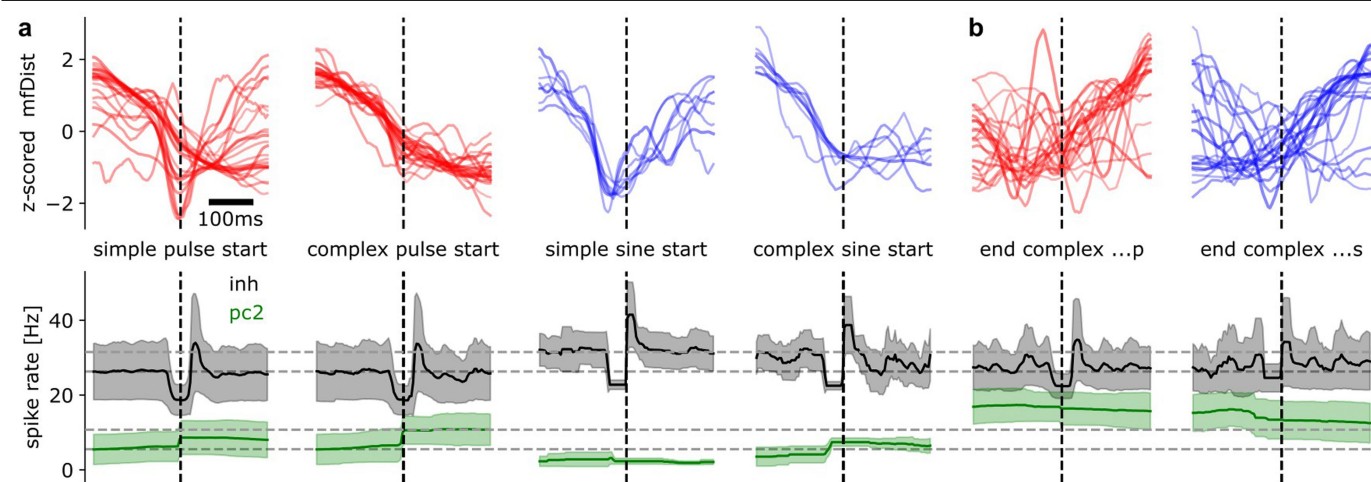

**Extended Data Fig. 6 | Neural activity dynamics driving simple and complex bouts in the song circuit model (supplement to Fig. 5). a**, (top) Z-scored male-female distance (mfDist) from wild-type courtship data (which served as input to the model) triggered around the time of simple (p,s) or complex (ps...,sp...) bout start in simulations of the song circuit model. Each line is the z-scored mfDist averaged across bouts for one simulation (lines were smoothed for visualization, using a uniform filter of 44.4 ms length). Every simulation uses song randomly chosen from all wild-type recordings (such that the chosen song contained a minimum of 10% of bouts at mfDist ≥ 4 mm, and the fit error / objective function value was below 0.1). For all bout types, mfDist decreases around the time of bout start. (Bottom) Instantaneous spike rate of the 'pC2' (green) and 'inh' (black) nodes of the circuit model around the time of bout onset. Distinctly timed release from inh-mediated inhibition in combination with distinct levels of pC2-mediated excitation drives different bout types. **b**, Dynamics of z-scored mfDist (top) and instantaneous spike rate of the pC2 and inh nodes in the circuit model at the time of bout termination, for complex bouts ending in pulse (left) or sine mode (right). In both cases, bout termination is accompanied by increases in mfDist and a resulting reduction in pC2-mediated excitation of the pulse and sine node. **a-b**, $n = 24$ model fits to song (400 seconds each) randomly chosen from $n = 20$ wild-type recordings (biological replicates).

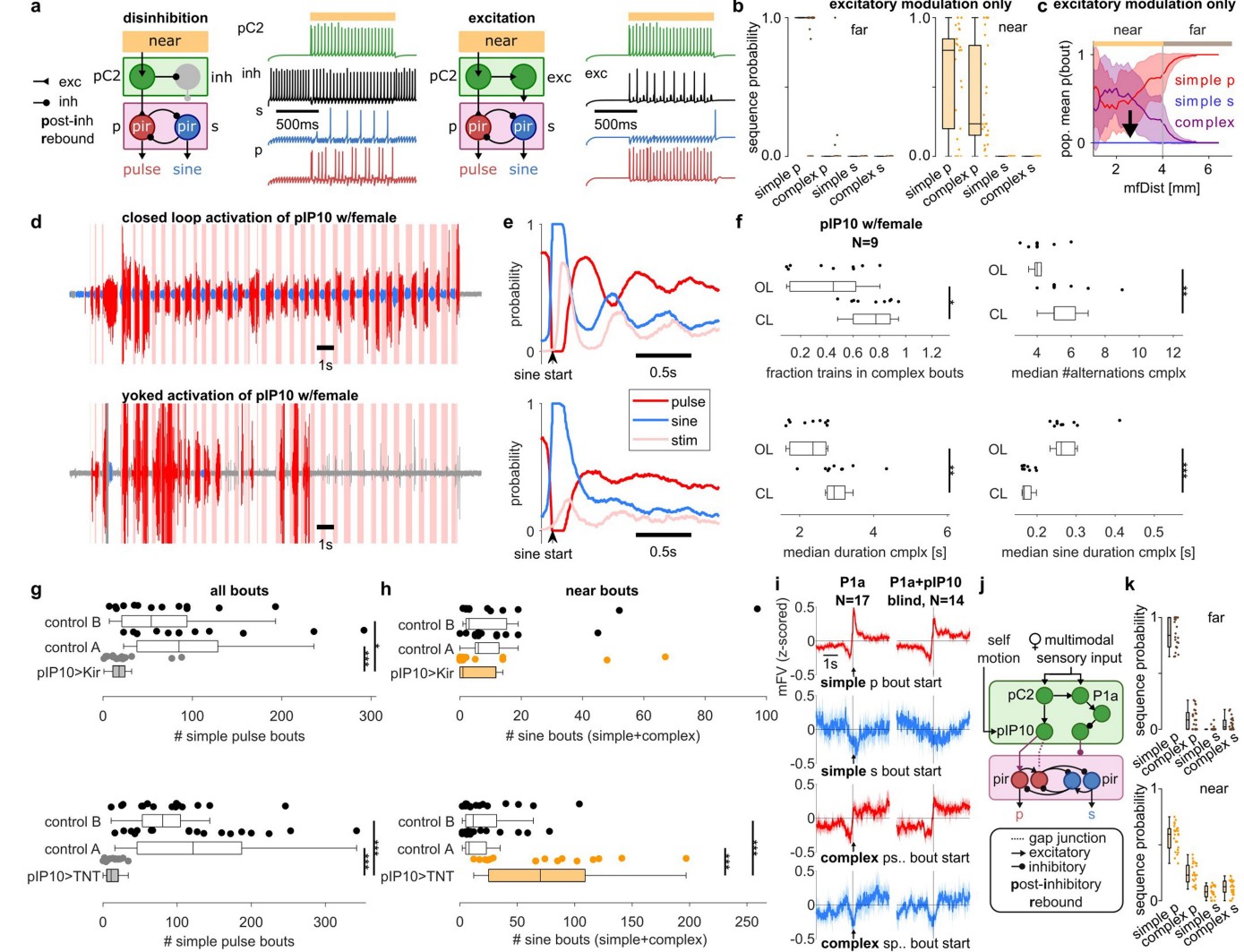

**Extended Data Fig. 7** | See next page for caption.

**Extended Data Fig. 7 | Testing and expanding the neural circuit model of context-dependent song patterning (supplement to Fig. 5). a**, Song circuit model with default disinhibitory modulation of the pulse/sine rebound circuit (left) and quasi-equivalent excitatory modulation (right), with simulated responses of the respective four nodes to a 'near' input. **b**, 'Song' statistics of the model with excitatory modulation (a, compare Fig. 5f). In contrast to the default model with disinhibition, the excitatory model exclusively produces pulse bouts (simple and complex). **c**, Population-averaged probability of simulated simple pulse (red), simple sine (blue), or complex (purple) bouts at a given male-female distance (mfDist) in the model with excitatory modulation (a) matches the relationship between distance and song types observed in courting wild-type flies for simple pulse and complex bouts, but not for simple sine bouts (compare with Fig. 1c). **d**, Triggering pIP10 activation on sine song in males courting a wild-type female strongly increases bout duration and complexity compared to controls with yoked activation (that is, identical stimulus statistics as in the closed loop condition but uncorrelated to the control male's song). Song shown from an example recording. **e**, Song and stimulus probability around the onset of male sine song, for closed loop (top) and yoked (bottom) activation of pIP10 during the recording shown in **a**). **f**, Population level comparison of four song features between closed-loop (CL) and yoked (OL) activation of pIP10 ($n = 9$ biological replicates): the fraction of trains belonging to complex bouts, the median number of sine-pulse or pulse-sine alternations in complex bouts, the median duration of complex bouts, and the median sine train duration within complex bouts. To show that all effects extend beyond generation of a single rebound sine, only 'psp...' bouts were considered for these analyses. Wilcoxon rank-sum test for equal medians; *P < 0.05; **P < 0.01; ***P < 0.001. **g**, Amount of simple pulse song bouts produced during courtship of a female, in recordings of males with tonically hyperpolarized pIP10 neurons (top) and males with blocked chemical synapses in pIP10 neurons (bottom; via expression of inward-rectifying potassium channels in pIP10 neurons, VT040556 > kir, and via expression of tetanus toxin light chain / TNT in pIP10 neurons, VT040556 > TNT; filled box plots),

compared to two genetic controls (blank box plots). See Supplementary Table 2 for genotypes. Under both manipulations, the amount of simple pulse bouts was strongly reduced in male song. **h**, Amount of simple and complex sine song bouts produced near a female, in recordings of males with tonically hyperpolarized pIP10 neurons (top) and males with blocked chemical synapses in pIP10 neurons (bottom; same manipulations as in **g**), compared to two genetic controls (blank box plots). The amount of bouts with leading sine was increased in males with blocked chemical synapses in pIP10, but unaffected in males with tonically hyperpolarized pIP10 compared to controls. **i** Z-scored male forward velocity (mFV) around the start of simple pulse (p), simple sine (s), or complex bouts with leading pulse (ps..) or sine (sp..), for solitary males with optogenetic activation of P1a and intact vision ($n = 17$ biological replicates; same as Fig. 4a), or blind males with simultaneous activation of P1a and pIP10 ($n = 16$ biological replicates; same as Fig. 4d). Only song bouts outside the stimulus interval (persistent song) are included here. At onset, bouts with leading pulse or sine show increases and decreases in mFV. **j**, Circuit model to explain the findings in **d**,**e**: chemical synapses from pIP10 onto the VNC pulse node explain the reduction in simple pulse bouts with kir and TNT expression in pIP10. Gap junctions (electrical synapses) between pIP10 and the inhibitory interneuron node of the pulse pathway facilitate simple and complex sine bouts with blocked chemical synapses in pIP10, by transforming pIP10 activity through the electric synapses into inhibition onto sine driving neurons, leading to rebound sine bouts after termination of pIP10 activity. **k**, Song bout statistics at far and near distances, for $n = 24\,200$ second segments randomly drawn from wild-type courtship data ($n = 20$ biological replicates) that were used to fit the model shown in Fig. 5a–c. **b**,**c**, $n = 93$ genetic algorithm fits to experimental song data randomly chosen from $n = 20$ biological replicates. **f-h**, Wilcoxon rank-sum test for equal medians; *P < 0.05, **P < 0.01, ***P < 0.001. **g**,**h**, $n = 15/13/13$ biological replicates for VT040556 > kir and the two genetic controls, $n = 16/18/16$ biological replicates for VT040556 > TNT and the two genetic controls.

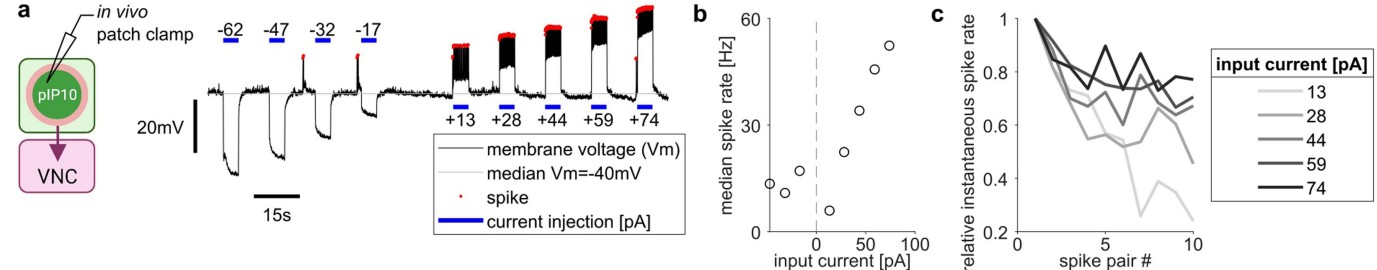

**Extended Data Fig. 8 | Spike-frequency adaptation in pIP10 neurons (supplement to Fig. 5). a**, In vivo patch-clamp electrophysiology of descending neuron pIP10. Action potentials (spikes) were observed both during injection of positive current and following injection of negative current (post-inhibitory rebound spikes). **b**, For each current stimulus amplitude, instantaneous spike rates were defined as the inverse of each inter-spike interval for all successive pairs of spikes observed within the trial. Shown is the median (per stimulus) instantaneous spike rate following negative or during positive current injection. **c**, Instantaneous spike rate of the first ten spike pairs in each trial, normalized by the spike rate of the first pair. Normalized spike rate decreases for successive spikes, indicative of spike-frequency adaptation.

# Reporting Summary

## Statistics

For all statistical analyses, confirm that the following items are present in the figure legend, table legend, main text, or Methods section.

| n/a | Confirmed | |
|---|---|---|
| ☐ | ☒ | The exact sample size (*n*) for each experimental group/condition, given as a discrete number and unit of measurement |
| ☐ | ☒ | A statement on whether measurements were taken from distinct samples or whether the same sample was measured repeatedly |
| ☐ | ☒ | The statistical test(s) used AND whether they are one- or two-sided<br>*Only common tests should be described solely by name; describe more complex techniques in the Methods section.* |
| ☐ | ☒ | A description of all covariates tested |
| ☐ | ☒ | A description of any assumptions or corrections, such as tests of normality and adjustment for multiple comparisons |
| ☐ | ☒ | A full description of the statistical parameters including central tendency (e.g. means) or other basic estimates (e.g. regression coefficient) AND variation (e.g. standard deviation) or associated estimates of uncertainty (e.g. confidence intervals) |
| ☐ | ☒ | For null hypothesis testing, the test statistic (e.g. *F*, *t*, *r*) with confidence intervals, effect sizes, degrees of freedom and *P* value noted<br>*Give P values as exact values whenever suitable.* |
| ☒ | ☐ | For Bayesian analysis, information on the choice of priors and Markov chain Monte Carlo settings |
| ☒ | ☐ | For hierarchical and complex designs, identification of the appropriate level for tests and full reporting of outcomes |
| ☒ | ☐ | Estimates of effect sizes (e.g. Cohen's *d*, Pearson's *r*), indicating how they were calculated |

*Our web collection on statistics for biologists contains articles on many of the points above.*

## Software and code

Policy information about availability of computer code

| Data collection | Sound and video recordings from freely behaving flies were obtained using custom scripts, run via python 2.7.<br>Functional imaging data were collected using ScanImage 2017 software and custom scripts run via Matlab 2018b (https://github.com/murthylab/FlyCaImAn). Electrophysiology data were collected using wavesurfer version 0.982 (https://wavesurfer.janelia.org/), run via Matlab 2018b |
|---|---|
| Data analysis | All analyses of behavior, functional imaging, and electrophysiology data were performed using custom scripts in Matlab R2019a. Circuit model simulations and analyses were performed using custom scripts in python 3.7.<br>Code will be available at https://github.com/murthylab/ |

For manuscripts utilizing custom algorithms or software that are central to the research but not yet described in published literature, software must be made available to editors and reviewers. We strongly encourage code deposition in a community repository (e.g. GitHub). See the Nature Portfolio guidelines for submitting code & software for further information.

## Data

Policy information about availability of data

All manuscripts must include a data availability statement. This statement should provide the following information, where applicable:
- Accession codes, unique identifiers, or web links for publicly available datasets
- A description of any restrictions on data availability
- For clinical datasets or third party data, please ensure that the statement adheres to our policy

Data are available upon request from the corresponding author. Source data are provided with this paper.

## Research involving human participants, their data, or biological material

Policy information about studies with human participants or human data. See also policy information about sex, gender (identity/presentation), and sexual orientation and race, ethnicity and racism.

| | |
|---|---|
| Reporting on sex and gender | *Use the terms sex (biological attribute) and gender (shaped by social and cultural circumstances) carefully in order to avoid confusing both terms. Indicate if findings apply to only one sex or gender; describe whether sex and gender were considered in study design; whether sex and/or gender was determined based on self-reporting or assigned and methods used. Provide in the source data disaggregated sex and gender data, where this information has been collected, and if consent has been obtained for sharing of individual-level data; provide overall numbers in this Reporting Summary. Please state if this information has not been collected. Report sex- and gender-based analyses where performed, justify reasons for lack of sex- and gender-based analysis.* |
| Reporting on race, ethnicity, or other socially relevant groupings | *Please specify the socially constructed or socially relevant categorization variable(s) used in your manuscript and explain why they were used. Please note that such variables should not be used as proxies for other socially constructed/relevant variables (for example, race or ethnicity should not be used as a proxy for socioeconomic status). Provide clear definitions of the relevant terms used, how they were provided (by the participants/respondents, the researchers, or third parties), and the method(s) used to classify people into the different categories (e.g. self-report, census or administrative data, social media data, etc.) Please provide details about how you controlled for confounding variables in your analyses.* |
| Population characteristics | *Describe the covariate-relevant population characteristics of the human research participants (e.g. age, genotypic information, past and current diagnosis and treatment categories). If you filled out the behavioural & social sciences study design questions and have nothing to add here, write "See above."* |
| Recruitment | *Describe how participants were recruited. Outline any potential self-selection bias or other biases that may be present and how these are likely to impact results.* |
| Ethics oversight | *Identify the organization(s) that approved the study protocol.* |

Note that full information on the approval of the study protocol must also be provided in the manuscript.

# Field-specific reporting

Please select the one below that is the best fit for your research. If you are not sure, read the appropriate sections before making your selection.

☒ Life sciences   ☐ Behavioural & social sciences   ☐ Ecological, evolutionary & environmental sciences

For a reference copy of the document with all sections, see nature.com/documents/nr-reporting-summary-flat.pdf

# Life sciences study design

All studies must disclose on these points even when the disclosure is negative.

| | |
|---|---|
| Sample size | Sample sizes were not predetermined, but are similar to those reported in previous publications (Clemens et al., 2018 Curr Biol; Sten et al., 2021 Nature). |
| Data exclusions | For behavioral experiments, recordings of male-female pairs containing copulation were excluded from analysis unless copulation statistics were subject of the analysis. |
| Replication | Each experiment presented in the paper was repeated across multiple days (often months) in at least 3 animals, and effects were consistent across animals. |
| Randomization | Experimental groups were defined based on genotype, and data acquisition was randomized with respect to different genotypes. |
| Blinding | Experiments were not done blind to genotype. However all animals that met the criteria for inclusion were analyzed, and analysis of behavioral data was based on automated approaches (pose estimation using SLEAP, song segmentation using FlySongSegmenter). |

# Reporting for specific materials, systems and methods

We require information from authors about some types of materials, experimental systems and methods used in many studies. Here, indicate whether each material, system or method listed is relevant to your study. If you are not sure if a list item applies to your research, read the appropriate section before selecting a response.

## Materials & experimental systems

| n/a | Involved in the study |
|-----|----------------------|
| ☐ | ☒ Antibodies |
| ☐ | ☐ Eukaryotic cell lines |
| ☐ | ☐ Palaeontology and archaeology |
| ☐ | ☒ Animals and other organisms |
| ☐ | ☐ Clinical data |
| ☐ | ☐ Dual use research of concern |
| ☐ | ☐ Plants |

## Methods

| n/a | Involved in the study |
|-----|----------------------|
| ☐ | ☐ ChIP-seq |
| ☐ | ☐ Flow cytometry |
| ☐ | ☐ MRI-based neuroimaging |

## Antibodies

| | |
|---|---|
| Antibodies used | Primary antibodies used were mouse anti-Bruchpilot (nc82, Developmental Studies Hybridoma Bank, AB2314866) and chicken anti-GFP (Invitrogen A10262). Secondary antibodies used were Alexa 488-conjugated goat anti-chicken (Invitrogen A11039) and Alexa 568-conjugated goat anti-mouse (Invitrogen A11004). |
| Validation | All antibodies used in this study are commercial and previously validated for immunohistochemistry in Drosophila, as described on the manufacturers' website. Primary antibodies have also been validated for application in Drosophila by the FlyLight project at Janelia Research Campus (https://www.janelia.org/project-team/flylight/protocols). |

## Eukaryotic cell lines

Policy information about cell lines and Sex and Gender in Research

| | |
|---|---|
| Cell line source(s) | *State the source of each cell line used and the sex of all primary cell lines and cells derived from human participants or vertebrate models.* |
| Authentication | *Describe the authentication procedures for each cell line used OR declare that none of the cell lines used were authenticated.* |
| Mycoplasma contamination | *Confirm that all cell lines tested negative for mycoplasma contamination OR describe the results of the testing for mycoplasma contamination OR declare that the cell lines were not tested for mycoplasma contamination.* |
| Commonly misidentified lines (See ICLAC register) | *Name any commonly misidentified cell lines used in the study and provide a rationale for their use.* |

## Palaeontology and Archaeology

| | |
|---|---|
| Specimen provenance | *Provide provenance information for specimens and describe permits that were obtained for the work (including the name of the issuing authority, the date of issue, and any identifying information). Permits should encompass collection and, where applicable, export.* |
| Specimen deposition | *Indicate where the specimens have been deposited to permit free access by other researchers.* |
| Dating methods | *If new dates are provided, describe how they were obtained (e.g. collection, storage, sample pretreatment and measurement), where they were obtained (i.e. lab name), the calibration program and the protocol for quality assurance OR state that no new dates are provided.* |

☐ Tick this box to confirm that the raw and calibrated dates are available in the paper or in Supplementary Information.

| | |
|---|---|
| Ethics oversight | *Identify the organization(s) that approved or provided guidance on the study protocol, OR state that no ethical approval or guidance was required and explain why not.* |

Note that full information on the approval of the study protocol must also be provided in the manuscript.

# Animals and other research organisms

Policy information about studies involving animals; ARRIVE guidelines recommended for reporting animal research, and Sex and Gender in Research

| | |
|---|---|
| Laboratory animals | All flies (Drosophila melanogaster) used for behavioral analysis were 3-5 days old male and female virgins (single males or a pair of one male and one female). Flies used for functional imaging experiments were 5-8 day-old virgin males. Flies used for patch-clamp electrophysiology were 1-3 day-old virgin males. Flies used for immunohistochemsitry were 3-6 days old. Images of brains are all males. Additional details are provided in the methods and tables S1 and S2. |
| Wild animals | This study did not involve wild animals. |
| Reporting on sex | Male and female flies were used in this study. Sex was determined visually. The central findings apply to male flies. |
| Field-collected samples | This study did not involve animals collected from the field. |
| Ethics oversight | No ethical approval was required for work on Drosophila melanogaster. |

Note that full information on the approval of the study protocol must also be provided in the manuscript.

# Clinical data

Policy information about clinical studies

All manuscripts should comply with the ICMJE guidelines for publication of clinical research and a completed CONSORT checklist must be included with all submissions.

| | |
|---|---|
| Clinical trial registration | *Provide the trial registration number from ClinicalTrials.gov or an equivalent agency.* |
| Study protocol | *Note where the full trial protocol can be accessed OR if not available, explain why.* |
| Data collection | *Describe the settings and locales of data collection, noting the time periods of recruitment and data collection.* |
| Outcomes | *Describe how you pre-defined primary and secondary outcome measures and how you assessed these measures.* |

# Dual use research of concern

Policy information about dual use research of concern

## Hazards

Could the accidental, deliberate or reckless misuse of agents or technologies generated in the work, or the application of information presented in the manuscript, pose a threat to:

| No | Yes | |
|---|---|---|
| ☒ | ☐ | Public health |
| ☒ | ☐ | National security |
| ☒ | ☐ | Crops and/or livestock |
| ☒ | ☐ | Ecosystems |
| ☒ | ☐ | Any other significant area |

## Experiments of concern

Does the work involve any of these experiments of concern:

| No | Yes | |
|---|---|---|
| ☒ | ☐ | Demonstrate how to render a vaccine ineffective |
| ☒ | ☐ | Confer resistance to therapeutically useful antibiotics or antiviral agents |
| ☒ | ☐ | Enhance the virulence of a pathogen or render a nonpathogen virulent |
| ☒ | ☐ | Increase transmissibility of a pathogen |
| ☒ | ☐ | Alter the host range of a pathogen |
| ☒ | ☐ | Enable evasion of diagnostic/detection modalities |
| ☒ | ☐ | Enable the weaponization of a biological agent or toxin |
| ☒ | ☐ | Any other potentially harmful combination of experiments and agents |

# Plants

| | |
|---|---|
| Seed stocks | *Report on the source of all seed stocks or other plant material used. If applicable, state the seed stock centre and catalogue number. If plant specimens were collected from the field, describe the collection location, date and sampling procedures.* |
| Novel plant genotypes | *Describe the methods by which all novel plant genotypes were produced. This includes those generated by transgenic approaches, gene editing, chemical/radiation-based mutagenesis and hybridization. For transgenic lines, describe the transformation method, the number of independent lines analyzed and the generation upon which experiments were performed. For gene-edited lines, describe the editor used, the endogenous sequence targeted for editing, the targeting guide RNA sequence (if applicable) and how the editor was applied.* |
| Authentication | *Describe any authentication procedures for each seed stock used or novel genotype generated. Describe any experiments used to assess the effect of a mutation and, where applicable, how potential secondary effects (e.g. second site T-DNA insertions, mosiacism, off-target gene editing) were examined.* |

# ChIP-seq

## Data deposition

☐ Confirm that both raw and final processed data have been deposited in a public database such as GEO.

☐ Confirm that you have deposited or provided access to graph files (e.g. BED files) for the called peaks.

| | |
|---|---|
| Data access links<br>*May remain private before publication.* | *For "Initial submission" or "Revised version" documents, provide reviewer access links. For your "Final submission" document, provide a link to the deposited data.* |
| Files in database submission | *Provide a list of all files available in the database submission.* |
| Genome browser session<br>(e.g. UCSC) | *Provide a link to an anonymized genome browser session for "Initial submission" and "Revised version" documents only, to enable peer review. Write "no longer applicable" for "Final submission" documents.* |

## Methodology

| | |
|---|---|
| Replicates | *Describe the experimental replicates, specifying number, type and replicate agreement.* |
| Sequencing depth | *Describe the sequencing depth for each experiment, providing the total number of reads, uniquely mapped reads, length of reads and whether they were paired- or single-end.* |
| Antibodies | *Describe the antibodies used for the ChIP-seq experiments; as applicable, provide supplier name, catalog number, clone name, and lot number.* |
| Peak calling parameters | *Specify the command line program and parameters used for read mapping and peak calling, including the ChIP, control and index files used.* |
| Data quality | *Describe the methods used to ensure data quality in full detail, including how many peaks are at FDR 5% and above 5-fold enrichment.* |
| Software | *Describe the software used to collect and analyze the ChIP-seq data. For custom code that has been deposited into a community repository, provide accession details.* |

# Flow Cytometry

## Plots

Confirm that:

☐ The axis labels state the marker and fluorochrome used (e.g. CD4-FITC).

☐ The axis scales are clearly visible. Include numbers along axes only for bottom left plot of group (a 'group' is an analysis of identical markers).

☐ All plots are contour plots with outliers or pseudocolor plots.

☐ A numerical value for number of cells or percentage (with statistics) is provided.

## Methodology

| | |
|---|---|
| Sample preparation | *Describe the sample preparation, detailing the biological source of the cells and any tissue processing steps used.* |
| Instrument | *Identify the instrument used for data collection, specifying make and model number.* |
| Software | *Describe the software used to collect and analyze the flow cytometry data. For custom code that has been deposited into a community repository, provide accession details.* |

| Cell population abundance | *Describe the abundance of the relevant cell populations within post-sort fractions, providing details on the purity of the samples and how it was determined.* |
|---|---|
| Gating strategy | *Describe the gating strategy used for all relevant experiments, specifying the preliminary FSC/SSC gates of the starting cell population, indicating where boundaries between "positive" and "negative" staining cell populations are defined.* |

☐ Tick this box to confirm that a figure exemplifying the gating strategy is provided in the Supplementary Information.

# Magnetic resonance imaging

## Experimental design

| Design type | *Indicate task or resting state; event-related or block design.* |
|---|---|
| Design specifications | *Specify the number of blocks, trials or experimental units per session and/or subject, and specify the length of each trial or block (if trials are blocked) and interval between trials.* |
| Behavioral performance measures | *State number and/or type of variables recorded (e.g. correct button press, response time) and what statistics were used to establish that the subjects were performing the task as expected (e.g. mean, range, and/or standard deviation across subjects).* |

## Acquisition

| Imaging type(s) | *Specify: functional, structural, diffusion, perfusion.* |
|---|---|
| Field strength | *Specify in Tesla* |
| Sequence & imaging parameters | *Specify the pulse sequence type (gradient echo, spin echo, etc.), imaging type (EPI, spiral, etc.), field of view, matrix size, slice thickness, orientation and TE/TR/flip angle.* |
| Area of acquisition | *State whether a whole brain scan was used OR define the area of acquisition, describing how the region was determined.* |

Diffusion MRI    ☐ Used    ☐ Not used

## Preprocessing

| Preprocessing software | *Provide detail on software version and revision number and on specific parameters (model/functions, brain extraction, segmentation, smoothing kernel size, etc.).* |
|---|---|
| Normalization | *If data were normalized/standardized, describe the approach(es): specify linear or non-linear and define image types used for transformation OR indicate that data were not normalized and explain rationale for lack of normalization.* |
| Normalization template | *Describe the template used for normalization/transformation, specifying subject space or group standardized space (e.g. original Talairach, MNI305, ICBM152) OR indicate that the data were not normalized.* |
| Noise and artifact removal | *Describe your procedure(s) for artifact and structured noise removal, specifying motion parameters, tissue signals and physiological signals (heart rate, respiration).* |
| Volume censoring | *Define your software and/or method and criteria for volume censoring, and state the extent of such censoring.* |

## Statistical modeling & inference

| Model type and settings | *Specify type (mass univariate, multivariate, RSA, predictive, etc.) and describe essential details of the model at the first and second levels (e.g. fixed, random or mixed effects; drift or auto-correlation).* |
|---|---|
| Effect(s) tested | *Define precise effect in terms of the task or stimulus conditions instead of psychological concepts and indicate whether ANOVA or factorial designs were used.* |

Specify type of analysis:    ☐ Whole brain    ☐ ROI-based    ☐ Both

| Statistic type for inference<br>(See Eklund et al. 2016) | *Specify voxel-wise or cluster-wise and report all relevant parameters for cluster-wise methods.* |
|---|---|
| Correction | *Describe the type of correction and how it is obtained for multiple comparisons (e.g. FWE, FDR, permutation or Monte Carlo).* |

## Models & analysis

| n/a | Involved in the study |
|---|---|
| ☐ | ☐ Functional and/or effective connectivity |
| ☐ | ☐ Graph analysis |
| ☐ | ☐ Multivariate modeling or predictive analysis |

**Functional and/or effective connectivity**

*Report the measures of dependence used and the model details (e.g. Pearson correlation, partial correlation, mutual information).*

**Graph analysis**

*Report the dependent variable and connectivity measure, specifying weighted graph or binarized graph, subject- or group-level, and the global and/or node summaries used (e.g. clustering coefficient, efficiency, etc.).*

**Multivariate modeling and predictive analysis**

*Specify independent variables, features extraction and dimension reduction, model, training and evaluation metrics.*

