## [Peer Review File · Nature]

Manuscript Title: FLEXIBLE CIRCUIT MECHANISMS FOR CONTEXT-DEPENDENT SONG SEQUENCING

Reviewer Comments & Author Rebuttals

Reviewer Reports on the Initial Version:

Referees' comments:

Referee #1 (Remarks to the Author):

Roemshied and colleagues undertake an in-depth analysis of courtship song in *Drosophila*, tying together new and old behavioral and physiological findings involving five circuit elements to develop a plausible model to describe robust and interesting context-dependent changes in song production. The mere plausibility of this model is a triumph in its potential explanatory power; it's a great foundation that will hopefully encourage more groups to test its hypotheses and add detail.

Before getting into my review, let me disclose some things about myself as a reviewer. My lab studies courtship behavior and builds circuit models, but we do not study song. Though I follow the song field I am not confident enough to assess with certainty the novelty of the findings in the paper, so I am assuming novelty unless otherwise stated. As a reviewer I am very reluctant to suggest (much less demand) experiments: I don't want to take control of the projects away from the authors. Instead, I describe what I appreciate and what I and have concerns about; which conclusions are well-supported and which (in my opinion) are not. I sign all of my reviews.

I want to state clearly that, despite my many critical or questioning comments below, I do really like this paper and would be happy to see a version of it in Nature.

If I had one wish for the paper, it would be that there was causal evidence that the rebound effect does in fact drive the switching between song modes (see point 5 below).

My overriding concern about the paper is that a casual reader would likely get the impression that the model is more complete, accurate, and detailed than it actually is. This occurs throughout the paper, with examples below (points 4 and 5 are most important for me):

1. In Figure 1B the authors lay out eight examples of song. It is immediately clear that the variety of songs produced by the male are only very superficially described by classifying them into simple or complex categories. This simplification is understandable, but I think a casual reader may get the feeling that the model explains much more song complexity than it actually does.

2. In figure 1C the authors declare an "abrupt change" in song that occurs when the male is within 4mm of the female. The data in Supplementary figure 1B, however, show the more nuanced and accurate picture: a gradual increase in complex song production from 6 to 1 mm away. In this specific case I would rather see the actual data in the main figures. Again, I understand the need to simplify a long and complex paper. I just worry that over-simplification can give a misleading picture,

especially when the text states the over-simplification without acknowledging the more accurate and complex situation.

3. The single fly song probability trace in figure 2b is clearly not representative, since its rebound probability is several times higher than average (as seen in the population average trace). I kind of understand wanting to show how strong the effect can be, but I again this again raises my over-simplification/streamlining concern.

4. My strongest concerns about over-simplifications/over-interpretations come from the genetic manipulations. A major example is pC2, which, in the model is the entry node for information to be processed to generate the various types of song. Given its position at the top of the hierarchy, one would assume that silencing pC2 would prevent song. Instead, the Murthy lab has previously found that: “pC2I-silenced males sang about twice as much as the controls” (Deutsch, Curr Bio 2019). A reader would certainly not get that impression from reading this paper. Again, I understand the need to simplify. But to me, not mentioning this huge problem with the model goes too far. I think it would be useful to state clearly the effects on song when silencing each of the circuit nodes in this paper, even when they disagree with the model. This would also give the authors a chance to explain why they think those apparent disagreements do not compromise the model.

5. And this over-simplification/streamlining concern extends to the rebound effect: an interesting and central component of the switching between song modes in the model. It is very plausible that this effect, evoked by optogenetic stimulation, does indeed drive (or at least contribute to) the natural switching between song modes. But rebound probability is never that strong (usually less than 0.3), happens very rarely in the imaging data (max 2 neurons/fly). To really make the case that this is happening in the natural context to drive song mode switching, the authors would have to do a manipulation that specifically prevented rebound firing of the neurons, showing that mode switching is reduced. That would likely require gaining molecular knowledge about the circuit, and I get that the authors would consider that to be beyond the scope of the paper. Still, in my opinion, this leaves the model in more of a plausible/provisional state. That is fine, it's just something I wish the authors would strongly and clearly acknowledge.

6. Also in the imaging data: for fly genetics reasons (I assume? I didn't see it explained) the authors cannot be sure that the neurons they are imaging in figure 3B are TN1, in fact the overwhelming majority of these neurons behave in a way that would be predicted to be the opposite of TN1 (they are activated in phase with pIP10 activation). The authors provide some reasons for this, but in the cartoon there are as many sine neurons as pulse neurons, while in the data the vast majority of the neurons would be likely to be classified as pulse. It also seems a stretch to me to designate neurons within the broad Dsx+ population as TN1 solely based on their response. And it is over-simplifying to say the least to have the y-axis on the imaging graph say “TN1 neurons” when the neurons being imaged are Dsx and so few of the neurons act in a way that would be expected of TN1 neurons.

7. I found the context dependency of the rebound phenomenon (both in the presence of the female and with stimulation of the upstream populations in isolated males) very convincing and exciting. However, I found the pulse rebound following TN1 stimulation in solitary males very unconvincing, since (as pointed out by the authors) pulse song also occurs during stimulation. This result (together

with the headless male experiment) occupies a lot of space, for pretty unconvincing effects. The rebound in the presence of a female is much more convincing; I don't understand why showing that the rebound effect is restricted to the VNC/S (I prefer ventral nervous system (VNS) for many reasons, including that the VNS looks nothing like a cord 😊) is worth so much main figure space for a relatively weak and unconvincing effect.

8. A potential issue that arises from the model and data simplifications is a lack of explanation for the occurrence of bouts that begin with the sine song component. In the initial fly data illustrating the context dependent nature of the song bout (Fig. 1c), in both cases the fly initiates song bouts with either simple or complex s. However, in the circuit model presented (Fig. 5a) excitation can only reach TN1 via pIP10, since P1a can only provide disinhibition.

I found some of the data in figures 4 and 5 to be less important to the conclusions of the paper than in figures 1-3. Figures 4a-f are clearly important and convincing, providing the neuronal basis of the context dependency (though see my concerns about pC2 loss of function above). And I thought the tap data in 4g-i was ok, I guess useful for how input gets to P1 neurons through a tap (though I know the authors are aware that olfactory and auditory input can also stimulate p1). I didn't find the estimated p1 activity and priming data that conclude figure 4 very compelling, interesting, or important for the model. Here and elsewhere I don't mean to insist that the authors should remove these data; I'm just giving my opinion.

And I'm not sure how important the modeling is in figure 5. Ultimately the circuit logic is pretty intuitive: p1 and pc2 provide information about the female and the pulse and sine neurons have rebound properties that control the switching between the two modes. Is this something that needs modeling? This is supported by the fact that the model still does a pretty good job even when you remove any of several components (In Fig. 5e the removal of any individual circuit element results in at worst a ~0.1 increase in model fit error, however with RMSE this variation is not too large). If the model were able to predict the nature of the complexity in song production (as opposed to whether the song will be simple or complex), that would be pretty impressive. But I think we are a long distance from that (at least in this paper...I wouldn't be surprised if the Murthy lab and diaspora figure it out one day).

Minor points:

I very much appreciate the efforts the authors have made to make the figures easy to understand. Still, it took me a long time to get through this paper. Some things that slowed me down:

In Figure 1d, the authors present a heat map to show what type of song the male sings depending on where he is in respect to the female. I did not understand how there can be so many dots behind the male point (are there just instances where he sings to a female completely out of his view?).

Figure 2a, I've stared at that red rectangle for a long while and still don't see how it tells me what the opto stimulation paradigms are. More useful are the (extremely thin) black lines in e.g. fig 2c. But they are never explicitly indicated as the durations of the stimuli, and are often obscured by the probability bars. The intensities of the opto stim are much more clearly presented. I still cannot

figure out what the grey lines are in figure 1b. I can see that they have blips...are they non-song sounds made by the flies (you need to zoom in many times to see them)? I also wondered if we need to see the results of every stim duration and intensity. Might it be simpler to just show the one in the main figs and put the rest in the supplement? And I think I prefer the probability traces at the bottom of Figure 1B to the layout in 1c-f and 4a-c. But that's just a thought.

In Figure 2b, I struggled to understand the relationship between trial numbers, individual flies and the population averages. For each paradigm the y-axis represents trial number, which typically denotes each horizontal line represents a single fly. However, in the single fly probability it seems that there were multiple trials per fly that were averaged to create the fly probability distribution. Does this mean that the population average is the average of individual flies, where each fly had multiple trials?

I don't like that the song type probability heat map scales change throughout the paper, for the obvious reason that it makes it harder to compare data between figures.

In Figure 2f, the authors compare the rebound sine from strong activation of the pIP10 neurons – my first concern with this is that the combination of scatter plot and box and whisker graphs makes it hard to figure out which data set goes with which condition. And there is a statistically significant difference between rebound probability in solitary vs. headless flies, but the means of the two conditions are basically identical – and the authors say in the text that the two conditions are comparable. Is this just a formatting mistake?

In figure 3b it would be nice to see example ROIs for the imaging data. Also, are the correlations between neurons the factor that is most important throughout figure 3? Seems to me that the neurons' response to the stimulation is more important (and more straightforward).

When the authors show that combined stimulation of P1a and pIP10 neurons is sufficient to elicit rebound sine song in Figure 4 a-f, it would be nice to see solitary pIP10 stimulation so that readers can compare all three conditions (only P1a, only pIP10, and then both P1a and pIP10).

In Figures 5c and 5e, using model fit error as opposed to a more explicit measurement of accuracy made things harder for me. For example, in the far case the model only predicts simple p, while in the actual data (in Figure 1c) the other three song options occur with non-zero probability, so that's not very accurate at a qualitative level. In the near case, the model predicts simple p with ~ 0.7 probability and complex s with virtually 0 probability, while in the fly data simple p only occurs with ~ 0.5 probability and complex s has clearly non-zero probabilities. I think some metric that captures these differences would be more intuitive and informative than model fit error.

Mike Crickmore

Referee #2 (Remarks to the Author):

Roemischied et al leverage the tractability of the drosophila song system to understand how social/behavioral context can influence a behavioral switch in pattern generation. The authors show that activation of a descending PIP10 neuronal system can activate pulse song (already known) but by testing a range of photoactivation intensities they discovered rebound sine song generation consistent with relief from inhibition. They examine the conditions under which this Pip10 mediated rebound sine is generated, and find that – behaviorally- the presence of a female and – neutrally – co-activation of P1a neurons (previously known to be tap-responsive) can enable rebound sine. Activation of pC2 neurons (previously known to be responsive to female visual/auditory cues) can also independently drive complex song, suggestive of pC2 mediated coactivation of pip10 and P1a. All of these neuronal systems work through at least two groups TN1 neurons, shown here to be functionally anticorrelated, in the ventral nerve cord that drive either pulse or sine song. The authors attempt to weave together these results into a coherent model in which yet-to-be identified population of neurons downstream P1a disinhibits sine-generating TN1 neurons in the VNC. This model can generate natural behavioral patterns, including key ones in which the distance to the female influences song complexity.

Overall, how switches are implemented at the circuit level is a major question in neuroscience, so this paper has the potential to be of general interest. The experiments are sound and the data are well analyzed. The major problem was the failure to identify the systems of neurons mediating the disinhibition that is at the core center of this paper. As a result, there is a lack of clarity over which conclusions are experimentally grounded and which are hypotheses (albeit sensible ones). Another issue was confusing sections due to inadequate algorithmic level description of the behavior and some difficulties squaring present results with past work.

(1) LACK OF CLARITY OVER WHAT ARE EXPERIMENTALLY SUPPORTED FACTS AND WHAT ARE INTERPRETATIONS/HYPOTHESES

The take-home model in Fig5a has its elegance, its simplicity, its plausibility, its ability to capture some key findings and explain the pulse-sine alternation in complex song and its dependence on proximity to the female. Though it is really a hypothesis and not a summary of results, the paper treats some key features of this model as ‘findings’ with only indirect support. Too many sections and sentences in this paper mention the discovery of a disinhibitory circuit to even mention – and it’s in the abstract – but the inhibition in the brain proposed to link p1a activation to rebound sine in the VNC is still a hypothesis. The same goes for the mutual inhibitory connections within the VNC that might be implementing the switch between the P and the S nodes. There are good ideas here, but they remain speculative. That these details are not really nailed down raises questions and begs any critical reader to imagine alternative scenarios that might work too.

Does it really HAVE to be a disinhibition circuit that links P1a to sine generation? I actually agree with the author’s interpretation of all of their results – the data are sound and the arguments are sound - but the fact that the necessary inhibitory systems are not identified makes me wonder about alternatives. For example to me it is not impossible that descending neuromodulatory or peptidergic systems that could be co-released alongside excitatory inotropic systems could do some heavy lifting in the service of context-dependent inhibition.

I don’t think these shortcomings sink the paper. A single paper can only do so much towards a

mechanistic description of such a complex set of nested behaviors. But I think the paper would be much stronger if it was crystal clear what is experimental result, what is hypothesis, and what future experiments could test that hypothesis.

At the absolute minimum, Fig 5, its legend, and all text needs to be extremely explicit about what is hypothesis and what is experimental result. Any mention of the discovery of a disinhibitory motif should be tempered appropriately, e.g. 'proposed disinhibitory motif' or 'conditional activation of sine consistent with a disinhibitory process.'

The slam dunk would be if the authors could actually identify the circuit linking P1a activation to song complexity. Perhaps identification of this system would support the authors' hypotheses. Perhaps not.

(2) BETTER DESCRIPTION OF THE BEHAVIOR AND THE PLACE OF THE PULSE/SINE SWITCH WITHIN IT
Courtship interactions and motor programs produced by the male are very complicated and require several 'switches' of the type investigated for the present paper (pulse vs sine). A crisp description of the full behavior and the place of the pulse/sine within it is lacking, despite the exquisite analyses that identified pulse fast vs slow (previously) or mf relative positions in relation to song complexity (e.g. fig 4). Of course no one paper is going to 'solve' all of these circuits, but an introductory figure, perhaps in supplement, that clearly lays out the full behavior will make it easier for readers, especially young graduate students, to understand exactly what is and what is not at stake in the mechanisms of pulse/sine transitions during complex song.

So I think an algorithmic description of the behavior in the spirit of Marvin Minsky (Society of Mind) would help the reader fully appreciate what exactly is at stake in the current study. Here is my first pass effort to put the pulse vs sine decision in its appropriate context:

- (1) Sing/Court vs do other things (e.g. fly, forage for food)
- (2) If singing, sine vs Pulse (addressed here)
- (3-4) If sine, left vs right wing; (4) If pulse, left vs right wing.
- (5-8) If sine with left wing, lift vs depress wing; (same goes with sine-right wing; pulse left-right wings)
- (9) If complex song ongoing (or possibly other conditions), initiate Tap or Walk
- (10) If walk pick direction and speed and, suppress tap
- (11) If tap, choose left vs right leg
- (12-13) If tapping (see * Below) left leg make contact depress leg (lift vs depress – same with right)
- (14-...) If ???, then mount – and this mount likely needs to be aimed which will additionally require directional 'choices'

This paper is therefore only about sine vs pulse (#2) – and this is just one of many, equally important in my opinion, 'decisions' that the fly makes during courtship. All of these decisions will be associated with little circuit motifs that will almost certainly involve lateral inhibition – as this is a basic principle of CPG function going back decades. Given this, the authors can better motivate why this small slice of the courtship behavior really provides the key and/or the experimental tractability to get at a context dependent switch. The authors attempt to do this by emphasizing the

dependence on complex song on sensory feedback (the presence of and distance to female, and the fact that known circuits may implement this (PC2 and P1a)).

To summarize this point, the paper in its current form immediately zeros in on the pulse-sine decision without the perspective of the whole behavior in which this decision is embedded. I think this issue could be addressed with a call to supplemental text and perhaps a supplemental figure that puts this question in broader context.

*question about the taps: When does tap happen in song - related to p or s or ps boundaries. It is shown that the tap rate is higher for complex bouts and at close distances - but there is some other CPG driving the tap? And is the tap with the left or right front leg? And does this depend on which wing? The animal cannot tap with both - so a similar winner-take all process must determine which leg taps. Are these all nested? This question about taps is not meant to motivate more experiments as it is outside the scope of the paper but simple text/intro to the tap system/behavior is necessary in advance of the section on taps.

(2.2) The results section around Fig 2, the 2p imaging in the VNC is complicated by the unstated fact that there are really two distinct all-or-none action selection 'decisions' being made. Sine vs pulse (as motivated in the text) but also left pulse vs right pulse and left sine vs right sine.) If either wing can produce either song, and if this 'decision' is under sensory influence (e.g. where the female is) then one can imagine a perfect ensemble of neurons with activity anti-correlated with the pIP20 activation even in a pulse-only context given competition between left and right wings. It's confusing in the text how this issue was address experimentally or conceptually.

For example, in the TN imaging dataset the anticorrelated populations – could they not also be related to driving left or right wings? Were the authors imaging both sides of the VNC? This section was very confusing because the authors seemed to be assuming that these neurons were pulse- or sine-driving. Perhaps there was reason for this but as I read the section I was considering an alternative that they had zeroed in on the left-vs-right decision circuits.

(2.3). This paper is filled with very intense photoactivations of premotor circuits and as it is written all the authors observed were 'normal' patterns of pulse and sine. But there is a lot to be learned about underlying circuits if there was (or if there never was) non-ethological 'garbage' that came out of photoactivations. For example, did photactivations ever cause (1) bilateral singing? (2) a weird combination of pulse and sine that was not interpretable? The reason this matters is that some circuits would make the all-or none condition a necessity (e.g. as in cricket stridulation vs flight) but others may allow co-activations (e.g. as in flexor extensor co-activation during locomotion or in dystonia).

Related to this point – the authors write in Line 117: "temporally separated pulse and rebound sine" The presence of this qualification is confusing – it makes me wonder if pulse and sine can ever NOT Be temporally separated, i.e. did photoactivation of any line ever drive indiscernable behaviors that may show that co-activation of both pathways? Co-activation of obth pathways under non-ethological conditions of genetically targeted photoactivations can reveal to what extent the two systems are necessarily all-or-none.

(2.4) Is near vs far also a switch or is it analog?

Finally, the near vs far state is also written about as a binary switch in paragraph at 347, which would also need a binary adaptive switch circuit - but I don't see this in the circuit diagram and it also does not square with the fact that in photoactivations P1a and pIP10 both seem to function in analog fashion - in which the amplitude and duration of their activations have effects on behavior. Neither of these brain neuron groups appear to function as all or none. So this section is very confusing. Please clarify.

(3) CONFUSING RESULTS (OR POSSIBLY CONFUSING PRESENTATION OF RESULTS) IN SOME SECTINOS

(3.1). In line 168 it is suggested that PC2 is primary driver of both Pip10 and P1a and therefore at the top of a hierarchy to drive complex song, especially close to a female. This 'suggestion' - predicts very clearly that pC2 photoactivation would co-activate pIP10 and P1a. Is this the case? Although I am loathe to suggest more experiments as this is already a very full paper, PC2 activation of pip10 and p1a is a strong prediction of the model – and perhaps it's already been done. But I cannot find this result in the paper or cited. And the experiment seems straightforward.

(3.2). Also I'm very confused by this assertion of PC2 at the top of the hierarchy because past work (Deutsch et al, 2019, by the authors) shows that complex song can exist, even be increased, following PC2 deletions. And it was also stated in that paper that no sine song was seen following PC2 activation, which also seems at odds with the present arguments. Lastly, this paper suggested that PC2 actually inhibited sine song – the exact opposite of what seems to be being argued here. Perhaps the authors have an understanding of how to integrate these observations together, but as it currently stands this is causing a level of confusion extremely distracting for a field outsider.

(3.3) Paragraph 492 and Fig 4: This is a very clever experiment - photoactivation of pIP10 with the female absent produces pulse only. Female presence increases bout complexity. This experimental design enables precise identification of what exactly about the female presence makes bout complex - which may be thought of as a behavioral way of testing for PC2 activation. But to test these ideas wasn't it more direct to just image the PC2 neurons and build a predictive model of what aspects of female position/body/orientation drive the neuron? And wasn't this the punchline of Deutsch et al? I cannot tell what is conceptually new about this whole section 192-212. There is a complex behavioral design, complex results, and figure panels that require head scratching. At the end of the day - wasn't it already apparent from the pc2 activation results that multimodal sensory cues control bout complexity? It was already argued that pc2 is primarily vision/sound, and p1a is primarily tap - both can modulate bout - so those results seem redundant with this whole section. Unless I'm missing an important motivation for these experiments, I think they can be moved to supplement to streamline this already large paper.

(3.4) Fig 4n-o shows that behavioral priming, presumably instantiated via p1a activation, reduces threshold of pip10 mediated pulse initiation, but without change in complexity.

Please relate this to Fig4d (coactivation of p1a and pip10 - here, it is shown that coactivation does increase complexity). Is what matters magnitude of p1a activation? Is this correct? Is the y axis of fig4d the laser power to both of the neuron types? did a mild p1a activation and

What was the pip10 irradiance in 4d? I cannot square 4d with 4n.

(4) MINOR

Abstract: Mutual inhibition and rebound excitability are not really computations (nothing is computed per se, signals are simply propagated in a certain way) - rather they are mechanisms that support a wide array of computations. I recommend replacing computations with mechanisms

Line 100: reciprocal inhibition of the inh nodes would do much more than prevent 'premature bout termination' (which is a strange and new concept in the flow of the paper, not even defined. If the authors want to motivate any points about premature bout termination they should show (or cite) bout duration distributions and make the prediction that eliminating the reciprocal inhibition between the inh nodes reduces bout duration and/or reduces the toggling between nodes. The real utility of reciprocal inhibition is the adaptive switch - robust toggling and all or none ax selection across a wide range of excitatory drives to each node.

Line 103: if pip10 is a descending neuron - so how did it stick around following decapitation? Are you photoactivating the terminals in the VNC? If so, please clarify in main text.

line 154 change control to initiation

Section at line 146 and line 159: Unclear description of results. The terms variable and complex are poorly defined. If complex is pulse and sine, what then is variable?

What are the black boxes in the rows of Fig 2c,e,h,j?

Line 162: levels is a poor word choice because it can mean amplitudes or durations and here you mean distinct durations.

line 166 change 'the' to 'a' - could future experiments conceivably identify a distinct and possibly redundant or even primary controller?

line 236. Unless I am missing something, the evidence for a disinhibitory pathway is simply that p1a activation alone could not strongly initiate song initiation but moderately affected probability of complex song. If this is correct, then the evidence for the disinhibitory pathway is really weak - at this point it's still a hypothesis.

Line 341: The authors write 'weak' but don't they mean 'brief'? It's important to distinguish amplitude versus duration of activation, as these arise from distinct behavioral events - and words like strong (weak) can mean either high (low) amplitude or long duration. Precise terminology is essential here.

Referee #3 (Remarks to the Author):

The male fruit fly unilaterally vibrates a wing as part of its courtship display. Prior studies have established that these courtship “songs” can be produced in two major modes: pulse song, which is louder; and sine song, which is quieter. It is also established that males predominantly generate pulse songs when distant from the female, then increase the proportion of sine songs with increasing proximity to the female. Consequently, as a male actively pursues a female, it produces song bouts comprising simple songs (pulse or sine) or complex songs (where they rapidly alternate between pulse and sine). Roemscheid et al use a combination of behavioral analysis, optogenetic manipulations, neural imaging and computational modeling to study how contextual cues regulate the transition from simple to complex songs. They provide new insights into the sensory cues that regulate this transition, and also provide some evidence for circuit mechanisms that regulate the context-dependent switch from simple to complex songs as well as the alternation between pulse and sine songs during complex song bouts.

An important problem addressed here is how sensory cues, especially those provided in different social contexts, alter behavior. Such context-dependent changes are features of locomotion and communication. In this latter arena, studies in humans, non-human primates and songbirds all reveal effects of social context on the quality and quantity of vocalizations. The fly provides a valuable organism in which to address explicit circuit mechanisms by which different sensory contexts unlock different motor circuit dynamics in the framework of sexual signaling and communication. In fact, the fly courtship song behavior has been mined extensively to rigorously quantify acoustic signatures of song; identify numerous sensory cues during courtship that regulate the amount and type of singing; localize neurons and circuits for pulse and sine song production; distinguish central neurons, motor neurons and flight muscle specializations for producing pulse or sine song; and the list goes on from there. Indeed, many of these advances are reported in prior publications from the senior author’s lab. More broadly, functional, genetic and comprehensive anatomical (ie, connectome) analyses in fly adults and larvae have provided detailed circuit models for sensory processing and motor control, including context-dependent changes to behavior.

It is on this background that this reviewer, who is not a fly researcher, tried to identify the novel aspects of the present study and assess their significance relative to earlier published work. As usual from the Murthy lab, many aspects of the present study are rigorously executed, including the careful characterization of social cues that regulate the switch from simple, pulse-only songs to complex songs. Another nice aspect of the study is the use of an optogenetic stimulation protocol that spans a wide range of light intensities and duty cycles to elicit sine song following prolonged optogenetic evocation of pulse song. The authors are also to be commended for further characterizing and modeling sensory cues that help to drive simple and complex song. Lastly, the integration of modeling methods to test certain compact circuit architectures that might be sufficient to explain context-dependent regulation of song is also a strength. However, the authors did not do a very good job of underscoring the major advances of the present study relative to early published research on this and related topics. Moreover, while the breadth of approaches employed

here is impressive, and I have great respect for the rigor of the behavioral analyses and the extension to circuit modeling, the circuit characterization itself is a bit thin and overly speculative, falling short of the leading efforts to analyze behaviorally-relevant central circuits in the fly. Along with a somewhat indirect style of exposition and argument, perhaps especially so for a Nature article, I was not left with the impression that I was reading about a major advance in this area of research. Perhaps some of this can be addressed by a more direct style of writing that clearly underscores what is new and why it matters. But even with such clearer writing, I believe that there are experimental gaps that lessen the potential impact of this study. Here I highlight some of the ways in which the current manuscript falls short.

1. Simple versus complex song. Both in the abstract and section 2 of the main text, the authors describe context-dependent changes from simple to complex song. From a quick scan of earlier studies, this transition from pulse only to pulse-sine song as the male engages the female courtship has already been described. While I understand that the authors need to show the behavior (again) to set up later stages of the study, it is presented in a manner that suggests novelty (this is especially so in lines 10-12 of the abstract, but is also an issue with much of section 2).

2. The supposition that complex song may be more attractive to the female is a bit off target in two ways. First, prior studies have established that sine song is important for the male to achieve copulation and, as complex song contains sine song, its attractiveness relative to pulse song alone would seem to be a given. But the notion that complex song, and particularly the alternation between pulse and sine song, is even more effective than sine song at male mating success is unknown, at least as I can discern. I am uncertain as to how the authors could separate the efficacy of sine song versus alternating pulse-sine song in this regard, but at the very least they need to test this idea. This is an example where speculation intrudes into the line of scientific argument in a disruptive way.

3. In the second paragraph of section 2, the finding that pulse song predominates at greater distances is presented as if it is a new finding. However, this has been documented previously (see Trott et al, 2012, for example). Perhaps it is not the authors intent to present this as a novel finding, and the analyses here are more sophisticated, but this is an example of how it is difficult for an outside reader to glean what exactly is novel in the present study. It would help to have the authors further distinguish the novel elements presented in figure 1 overall.

4. Same section (lines 52-54), the authors focus on longer song bouts to invoke inhibitory mechanisms, which would act in part to suppress singing when no female is present. While that is one possible explanation, it seems just as reasonable to assume that in the absence of female cues, the song generating circuit is simply offline, perhaps because the motor elements have a high threshold for activation. Excitation/arousal generated by those female cues then kicks an otherwise dormant circuit into action; no inhibition is needed.

5. Dual versus single stream control of song patterning has been bandied about in the fly song literature for some time. On the most fundamental level, the central thoracic neurons, the motor neurons and the wing muscles for pulse and sine song are distinct (e.g., von Phillipsborn, 2011; Shirangi et al, 2013, 2016), and flies never generate pulse and sine song at the same time, at least

from what I can see in this and other studies. Therefore, there must be some form of cross talk at some level of the neuromuscular control pathway to explain this “one at a time” behavioral output, regardless of whether there is single or dual channel descending control. Further, as males sing predominantly pulse songs at the first stage of the interaction with the female (at a distance), it seems that fatigue in the pulse-generating circuitry could relieve feedforward inhibition on the sine generating circuit. A winner takes all mechanism could account for the behavior regardless of whether descending control architecture is single or dual channel.

6. While the optogenetic protocol using a range of irradiance and duty cycles is a nice addition, figure 4B3-D3 from Clements et al (Current Biology, 2018) shows optogenetic stimulation of pIP10 can evoke pulse song in solitary males, which then transitions to sine song after the light pulse is turned off. It's fine to build on that earlier work in more detail, but the authors should cite their own prior work on this point, as the observation of optogenetically-evoked complex (pulse-sine) song is not novel.

7. That the optogenetic regime in solitary males only weakly recapitulates the complex song produced in the presence of a female raises the question of additional pathways operating in parallel to pIP10 or TN1. The headless fly experiment tells us that the additional drive depends on the head, which is further explored in Figure 4. But here in the experiments shown in Figure 2, is there a concern that prolonged optogenetic stimulation itself generates a rebound-off effect independent of any endogenous rebound circuitry?

8. Line 84: As per my earlier comment, I don't see how the authors can distinguish between a model where the sine generator is under tonic inhibition versus one where it simply has a high activation threshold. While citation of the work from Tuthill on VNC inhibition of proprioceptive signals is helpful, the lack of any information about such resting inhibition in sine generating circuits is a weakness of the present study.

9. Prior studies have already established that the male's interactions with the female drive song. What in those earlier studies or here supports the idea that female cues disinhibit versus excite the song pathway? Functionally these two mechanisms both result in song and there is nothing obvious to me in the behavior that would support one over the other.

10. While behavioral observations and peripheral specializations make it likely that there is some form of crosstalk between pulse and sine producing networks, evidence that it depends specifically on mutual inhibition as schematized in Figure 2m is lacking. The headless experiments in Figure 2 suggest that the putative mutual inhibition circuits reside in the VNC, but that leaves a lot of ground unexplored. The lack of an explicit circuit description to address this idea is a weakness of the study.

11. The two photon experiments are a start in the right direction towards such an explicit circuit dissection, but do not go very far, have some issues for interpretation and are not entirely consistent with a mutual inhibition model. First, while the bulk of TN1 neurons show (strongly) correlated activity, the authors focus on the smaller set that show uncorrelated activity. Is this small fraction of neurons that show anticorrelated activity larger than would be expected due to chance? Second, I have to admit being left confused by this set of results, as Shirangi et al (2016) reported that

stimulation of TN1 neurons generally drives sine song. Shouldn't most TN1 neurons be suppressed by pIP10 activation (which drives pulse song, as shown here in Figure 2c)? Third, Dsx also labels dPR1 neurons which are thought to drive sine song. Although this dissociation may suggest that the Dsx population labeled here may be excited or suppressed by pIP10, what evidence is there that mutual inhibition accounts for their anticorrelated activity? Does the proportion of correlated/anticorrelated neurons mirror the proportion of the purported pulse-driving dPR1 vs. the sine-driving TN1 neurons? Is it possible to determine the identities of these correlated/anti-correlated neurons, or to selectively record from these populations? Lastly, what is the identity of the putative inhibitory neurons that are interposed between the anticorrelated populations? This is place where a good initial observation could motivate a more explicit test of the circuit model. This characterization would likely need to extend beyond calcium imaging, which is generally not well-suited to characterize inhibitory networks. Ultimately, this more explicit characterization is lacking and it leaves the nature of the advance hard to assess.

12. In line 166, the authors conclude that pC2 neurons serve as the main determinant of song composition. How is this a distinct advance from work presented in references 8, 26 and 38?

12. Section 5. The authors do a nice job of varying the optogenetic excitation of P1a and pC2. However, again I feel that the model proposed is not sufficiently explored and tested. For instance, in line 173, the authors state that their results suggest that "P1a mediates disinhibition (via as of yet unknown circuitry)" of VNC. This is a rather unsatisfying claim and leads to the speculated existence of additional neurons in Fig. 4q – but it is unclear if these neurons exist.

13. Section 5, sensory cues. In the GLM model used to predict behavior, I am again left wondering what is the novel advance, given that the predictor variables (distance, angle, tapping, etc.) that feed into the GLM appear to be known influencers of song patterning in flies. Perhaps one novel aspect is that the authors use an estimate of p1A activity, but this estimate may largely reflect the behavior itself, since it is generated by using the tapping behavior and convolving that with previously observed calcium responses and GCAMP decay dynamics (e.g., Clowney et al., 2015). Although tapping might acutely excite p1A, how well does it explain the totality of p1A activity, particularly given prior work suggesting that p1A is modulated on longer timescales and may be sensitive to sign stimuli provided by the female (Clowney et al., 2015)? Generally, this seems to be a way to bring neural activity into this model, but it is unclear the degree to which this is distinct from just the behavioral history.

14. Relatedly, to what degree is tapping itself necessary for the generation of complex bouts? If male flies are physically prevented from tapping the females, but allowed to approach them closely, do they still generate complex song? This seems to be an important aspect of the model, given the role proposed for p1A in helping to mediate disinhibition of downstream neurons.

15. Section 6. While the authors can nicely recapitulate their results with the given circuit model, the overall lack of testing of various aspects of the model leave me uncertain that the proposed model is the best or only explanation for their results. Indeed, the authors analyze previous inhibition data in light of their model, and while they can explain these prior data, it requires that they modify their model to propose the existence of a chemical synapse between pIP10/VNC and an electrical synapse

between pIP10/inhibitory interneurons Fig. S7f. Is there evidence that either these VNC interneurons, or this electrical synapse exist? Overall, the model proposed here posits the existence of two classes of inhibitory interneurons in the VNC (one on pulse and one on sine-producing neurons), one interneuron in the brain (to allow for p1A disinhibition) and several new connections between these and other neurons. It seems to this reviewer that there ought to be some burden of proof here when proposing additional, previously unknown, circuit components.

16. A last more minor point is that I believe the authors miss the opportunity to link their research to another, older (and perhaps partly forgotten) line of research, namely changes in noctuid moth flight escape behaviors triggered by different intensities of ultrasound. While the birdsong references are interesting, there the underlying song pattern is highly similar in the two contexts, but slightly faster and more precise in the presence of a female. In the moth, like the fly, changes in sensory cues drive quite different motor patterns: Directed flight away from the ultrasound source when it is relatively faint, and a power dive into the substrate when the ultrasound intensity passes a critical threshold. This seems more similar to what the male fly does, which is to change the wing motor pattern as sensory cues change along a continuum.

Author Rebuttals to Initial Comments:

We thank the Reviewers for their detailed feedback - we have conducted new experiments and analyses and made changes to the text to address major concerns.

Point-by-Point Response to Reviewer concerns (*Reviewer comments in blue italics*, Author Responses in black):

Reviewer #1:

Roemschied and colleagues undertake an in-depth analysis of courtship song in Drosophila, tying together new and old behavioral and physiological findings involving five circuit elements to develop a plausible model to describe robust and interesting context-dependent changes in song production. The mere plausibility of this model is a triumph in its potential explanatory power; it's a great foundation that will hopefully encourage more groups to test its hypotheses and add detail.

Before getting into my review, let me disclose some things about myself as a reviewer. My lab studies courtship behavior and builds circuit models, but we do not study song. Though I follow the song field I am not confident enough to assess with certainty the novelty of the findings in the paper, so I am assuming novelty unless otherwise stated. As a reviewer I am very reluctant to suggest (much less demand) experiments: I don't want to take control of the projects away from the authors. Instead, I describe what I appreciate and what I have concerns about; which conclusions are well-supported and which (in my opinion) are not. I sign all of my reviews.

I want to state clearly that, despite my many critical or questioning comments below, I do really like this paper and would be happy to see a version of it in Nature.

If I had one wish for the paper, it would be that there was causal evidence that the rebound effect does in fact drive the switching between song modes (see point 5 below).

Our study combined optogenetics and high-resolution quantitative behavioral analysis to characterize neural contributions to song generation in *Drosophila*, and we proposed a neural circuit architecture that can account for the observed song statistics and its context-dependence (simple song sequences far from the female versus complex song sequences when near). At the core of this architecture, we proposed a circuit motif that combines mutual inhibition and post-inhibitory rebound excitability (see Fig. 2m). In support of this, we used targeted two-photon calcium imaging to identify neural correlates of mutual inhibition and post-inhibitory rebound within a population of neurons known to be involved in song production (TN1 neurons (Fig. 3a-d)). We thank the Reviewer for pointing out that while these results support our circuit hypothesis, we had not yet confirmed that sine-driving neurons within the TN1 neural cluster indeed are rebound excitable, and that sine song in freely behaving males is primarily driven via post-inhibitory rebound following release from inhibition. To address these concerns, we performed the following experiments:

- A) As hyperpolarization-activated cyclic nucleotide-gated (HCN) channels (Ih) constitute a critical building block for neuronal post-inhibitory rebound excitability across systems (Ascoli et al., 2010; Engbers et al., 2011; Gastrein et al., 2011; Ferrante et al., 2017), we designed experiments to silence these channels and examine the effects on song production. Our model predicted that

either mutating or knocking down *lh*, either system-wide or specifically in TN1 neurons, should remove or reduce both sine and complex song (this would correspond to removing post-inhibitory rebound (pir) from the sine-promoting neurons in the VNC). We found that *lh* mutant males (Hu et al., 2015; Fernandez-Chiappe et al., 2021) vigorously courted females. Nonetheless, they showed a strong reduction in complex song production (**new Fig. 3f-i**), in line with our hypothesis. In particular, *lh* mutant males sang more simple pulse-only bouts near the female (versus pulse-sine alternating bouts (complex p bouts)), consistent with our hypothesis that rebound excitability is the primary mechanism for switching between pulse and sine song. This was also true for males with a knock-down of *lh* specifically in TN1 neurons using RNAi (**new Fig. 3j-k**). When all TN1 neurons are activated, primarily sine song is produced (Fig. 2h and Shirangi et al. 2016), although TN1 neurons are a heterogeneous population (TN1A-E neurons) that contain both primarily sine-driving (TN1A) and pulse-driving (TN1C) neurons (Shirangi et al. 2016).

- B) Next, we knocked down GABA-A receptor (*Rdl*) expression in TN1 neurons via RNAi - this should have had the effect of reducing rebound excitation by decreasing the required inhibition that precedes rebound excitation. We anticipated observing a marked reduction in song complexity. We found that following knockdown of *Rdl* in TN1 neurons, the number of pulse-sine alternations was reduced compared to controls and that the probability of complex ps.. bouts was increased, supporting our hypothesis that rebound excitability of TN1 is required for complex song production (**new Fig. 3j-k**). During courtship, both *lh*- and *Rdl*- RNAi knockdown males tapped the female's abdomen as frequently as controls (data not shown), indicating the observed reduction in song complexity can be attributed to the reduction in rebound excitability and not a reduction in tap-mediated activation of P1a (see Fig. 4i).

My overriding concern about the paper is that a casual reader would likely get the impression that the model is more complete, accurate, and detailed than it actually is. This occurs throughout the paper, with examples below (points 4 and 5 are most important for me):

1. In Figure 1B the authors lay out eight examples of song. It is immediately clear that the variety of songs produced by the male are only very superficially described by classifying them into simple or complex categories. This simplification is understandable, but I think a casual reader may get the feeling that the model explains much more song complexity than it actually does.

We thank the Reviewer for this point and have now added text (**lines 39-40**) to emphasize that song sequence variability has several sources: bout patterning (the focus of this study), bout duration, amplitude modulation, modulation of the structure of pulse and sine trains, and modulation of substructure within those trains. We also now include histograms of the distributions of pulse train durations, sine train durations, and number of pulse/sine switches per bout in **new Supplemental Figure S1g**, to provide readers with a fuller appreciation for song structure.

2. In Figure 1C the authors declare an "abrupt change" in song that occurs when the male is within 4mm of the female. The data in Supplementary Figure 1B, however, show the more nuanced and accurate picture: a gradual increase in complex song production from 6 to 1 mm away. In this specific

case I would rather see the actual data in the main figures. Again, I understand the need to simplify a long and complex paper. I just worry that over-simplification can give a misleading picture, especially when the text states the over-simplification without acknowledging the more accurate and complex situation.

We have now rephrased the text to indicate the more gradual (rather than abrupt) transition between bout types as a function of male-female distance, and we have moved Supplementary Figure 1b to the main text (**new Fig. 1c**).

3. The single fly song probability trace in Figure 2b is clearly not representative, since its rebound probability is several times higher than average (as seen in the population average trace). I kind of understand wanting to show how strong the effect can be, but I again this again raises my over-simplification/streamlining concern.

In the population average in Fig. 2b, we have now replaced the errorbar with the raw data (one trace per recording), to show the heterogeneity across flies in rebound sine song production following pIP10 activation in solitary males. The population average for rebound sine is low due to 1) variability in the latency-to-peak rebound probability, and 2) heterogeneity in rebound sine song production across males.

4. My strongest concerns about over-simplifications/over-interpretations come from the genetic manipulations. A major example is pC2, which, in the model is the entry node for information to be processed to generate the various types of song. Given its position at the top of the hierarchy, one would assume that silencing pC2 would prevent song. Instead, the Murthy lab has previously found that: “pC2I-silenced males sang about twice as much as the controls” (Deutsch, Curr Bio 2019). A reader would certainly not get that impression from reading this paper. Again, I understand the need to simplify. But to me, not mentioning this huge problem with the model goes too far. I think it would be useful to state clearly the effects on song when silencing each of the circuit nodes in this paper, even when they disagree with the model. This would also give the authors a chance to explain why they think those apparent disagreements do not compromise the model.

We thank the Reviewer for pointing out the discrepancy between our model and prior results (*Deutsch et al. 2019*). The experiments by *Deutsch et al.* were performed in a different behavioral rig, so we re-ran the pC2 silencing experiments in our new setup (and re-generated the genetic strains, but used the same driver lines to target pC2 - see Fig. S5). For the new experiments, we confirmed the presence of TNT in all crosses using PCR. We found that blocking chemical synapses (via TNT) in pC2 neurons **decreased the total amount of song, including sine song** (rather than the increase reported in *Deutsch et al. 2019*). We now point out the difference between the two studies on **line 196-197**. While these new results do not alter the major findings of *Deutsch et al. 2019*, we are preparing a corrigendum to *Deutsch et al. 2019*.

Analysis of song statistics in pC2>TNT males revealed that song in silenced males was biased to simple pulse bouts both far from and near the female (less complex song both far from and near the female) (**new Fig. 4q-s**). This is **in line with our model**, because the proposed connection from pC2 to

P1a is blocked, which functionally prevents disinhibition of the rebound circuit downstream, and hence is expected to suppress the production of complex song (pulse-sine alternation song). Why is pulse song still produced in these males since our model also posits (as the Reviewer points out) that the pC2-pIP10 connection is the primary pathway for driving simple pulse song? This could either be due to additional electrical synapses from pC2 to pIP10 that remain intact, or to other inputs that drive pIP10 (e.g., male self motion, independent of vision, is known to drive pulse song (Coen et al. 2014)). Lillvis et al. 2022 have recently confirmed the existence of chemical synapses from pC2 to pIP10 (cited on pg 5), but they found that structural connectivity from pC2 to pIP10 (ie chemical synapses) is less predictive of optogenetically driven song than functional connectivity (ie activating pC2 and imaging pIP10), suggesting the presence of additional (electrical) synaptic connections.

5. And this over-simplification/streamlining concern extends to the rebound effect: an interesting and central component of the switching between song modes in the model. It is very plausible that this effect, evoked by optogenetic stimulation, does indeed drive (or at least contribute to) the natural switching between song modes. But rebound probability is never that strong (usually less than 0.3), and happens very rarely in the imaging data (max 2 neurons/fly).

We have now added text (line 141-143) to make clear that *both* in behavioral experiments with activation of pIP10 in solitary males, as well as in imaging experiments with pIP10 activation, the amount of rebound song in behavior or rebound activity in TN1 neurons (recorded via imaging) was expected to be small, due to the proposed baseline inhibition of the VNC song circuitry in the absence of female cues (as is the case for both of these experiments). In contrast, we show that activating pIP10 in a male that is actively courting a female relieves the song circuit from this baseline inhibition and ‘unlocks’ the full extent of rebound excitability, leading to rebound probability around 40% when near a female (see Fig. 2f). We subsequently show that this effect can be reproduced by co-activation of pIP10 and P1a neurons in the absence of a female (Fig. 4d).

To really make the case that this is happening in the natural context to drive song mode switching, the authors would have to do a manipulation that specifically prevented rebound firing of the neurons, showing that mode switching is reduced. That would likely require gaining molecular knowledge about the circuit, and I get that the authors would consider that to be beyond the scope of the paper. Still, in my opinion, this leaves the model in more of a plausible/provisional state. That is fine, it's just something I wish the authors would strongly and clearly acknowledge.

Please see our response above; we have now performed several experiments to either block or reduce rebound excitability, both using *lh* mutants and via TN1-specific knockdown (TN1>*lh*-RNAi and TN1>*Rdl*-RNAi), directly addressing this suggestion. We found evidence that indeed rebound excitability underlies the production of sine song and complex bouts. See **new Fig. 3f-k** and changes to the text (lines 150-161).

6. Also in the imaging data: for fly genetics reasons (I assume? I didn't see it explained) the authors cannot be sure that the neurons they are imaging in figure 3B are TN1, in fact the overwhelming majority of these neurons behave in a way that would be predicted to be the opposite of TN1 (they are activated in phase with pIP10 activation). The authors provide some reasons for this, but in the cartoon

there are as many sine neurons as pulse neurons, while in the data the vast majority of the neurons would be likely to be classified as pulse. It also seems a stretch to me to designate neurons within the broad Dsx+ population as TN1 solely based on their response. And it is over-simplifying to say the least to have the y-axis on the imaging graph say “TN1 neurons” when the neurons being imaged are Dsx and so few of the neurons act in a way that would be expected of TN1 neurons.

We have now added text (**line 552-560**) to emphasize that while we used flies that express GCaMP6s in all Dsx+ neurons, we only imaged the Prothoracic and Mesothoracic neuromeres and the Accessory Mesothoracic neuropil of the ventral nerve cord. All together these regions house the Pr1-3, Pr4, Ms1-3 and TN1 cluster of neurons (Nojima et al. 2021; Fig. S1) whose somas have distinct and identifiable locations. The data shown are from manually segmented somas from these regions that, based on their anatomical location, were unambiguously identified as TN1 neurons. TN1 can be distinguished from dPR1 (which belongs to the Pr1-3 cluster) based on the position of the somas in the anterior-posterior axis. Similarly, TN1 can be readily distinguished from its neighboring clusters (Pr4 and Ms1-3) based on its more lateral and ventral location relative to the Accessory Mesothoracic neuropil, as well as the smaller size of its somas. Our manual segmentation was based on these criteria rather than on neural responses. We have also changed the cartoon in Figure 3a to indicate that the majority of TN1 neurons show activity during pulse song production (consistent with Shiozaki et al., 2022), whereas a small subset, TN1A, is thought to be responsible for sine song (Shirangi et al. 2016), possibly due to stronger/direct synaptic connections from TN1A neurons to motor neurons (Shirangi et al., 2013).

7. I found the context dependency of the rebound phenomenon (both in the presence of the female and with stimulation of the upstream populations in isolated males) very convincing and exciting. However, I found the pulse rebound following TN1 stimulation in solitary males very unconvincing, since (as pointed out by the authors) pulse song also occurs during stimulation. This result (together with the headless male experiment) occupies a lot of space, for pretty unconvincing effects. The rebound in the presence of a female is much more convincing; I don't understand why showing that the rebound effect is restricted to the VNC/S (I prefer ventral nervous system (VNS) for many reasons, including that the VNS looks nothing like a cord 😊) is worth so much main figure space for a relatively weak and unconvincing effect.

We thank the Reviewer for this suggestion but have decided to leave the TN1 results in the main Figure - we feel they are important for understanding the generation of rebound sine song during complex bouts.

8. A potential issue that arises from the model and data simplifications is a lack of explanation for the occurrence of bouts that begin with the sine song component. In the initial fly data illustrating the context dependent nature of the song bout (Fig. 1c), in both cases the fly initiates song bouts with either simple or complex sine. However, in the circuit model presented (Fig. 5a) excitation can only reach TN1 via pIP10, since P1a can only provide disinhibition.

We have now added text to explain that bouts starting in sine mode constitute the minority of all song bouts (**line 54**). Yet, our model can still produce (rare) bouts starting in sine mode, via rebound of sine-driving neurons following disinhibition (e.g. via tap-mediated activation of P1a in the actual animal;

depending on whether sine or pulse driving neurons are closer to threshold at the time of disinhibition, this can lead to a bout starting in sine or pulse mode). **New Figure S6** now shows the unique conditions in the model that lead to bouts beginning with sine song; we have added horizontal lines in S6 as a visual aid to distinguish these conditions. We have also added text (**lines 299-304**) to explain that the model incorporates only the most basic assumptions on neural properties. For example, we used equal weights for all excitatory or all inhibitory connections in the model. Parametrizing these weights would likely lead to improved reproduction of rare bouts.

9. I found some of the data in Figures 4 and 5 to be less important to the conclusions of the paper than in Figures 1-3. Figures 4a-f are clearly important and convincing, providing the neuronal basis of the context dependency (though see my concerns about pC2 loss of function above). And I thought the tap data in 4g-i was ok, I guess useful for how input gets to P1 neurons through a tap (though I know the authors are aware that olfactory and auditory input can also stimulate P1). I didn't find the estimated P1 activity and priming data that conclude figure 4 very compelling, interesting, or important for the model. Here and elsewhere I don't mean to insist that the authors should remove these data; I'm just giving my opinion.

We thank the Reviewer for this feedback and have now moved panels m-p of the original Fig. 4 (priming data) to the supplement (**new Fig. S4**) - instead, in **new Fig. 4q-s**, we now include new panels on pC2>TNT silencing and on disinhibitory P1a-follower activity (see below).

10. And I'm not sure how important the modeling is in Figure 5. Ultimately the circuit logic is pretty intuitive: P1 and pC2 provide information about the female and the pulse and sine neurons have rebound properties that control the switching between the two modes. Is this something that needs modeling? This is supported by the fact that the model still does a pretty good job even when you remove any of several components (In Fig. 5e the removal of any individual circuit element results in at worst a ~0.1 increase in model fit error, however with RMSE this variation is not too large). If the model were able to predict the nature of the complexity in song production (as opposed to whether the song will be simple or complex), that would be pretty impressive. But I think we are a long distance from that (at least in this paper...I wouldn't be surprised if the Murthy lab and diaspora figure it out one day).

We now clarify in the figure caption (Fig. 5b) that the model fit error comprises a weighted sum of two terms, the RMSE between the experimental and model bout type distribution, weighted by a factor of 1, and the absolute difference between the number of bouts produced in experiment vs. model, weighted by a factor of 0.1. We agree this composition of the objective function, when interpreted as a pure RMSE, can lead to a false impression of the magnitude of the exp-model discrepancy.

Having the model allowed us to test specific hypotheses, including those that were highly challenging experimentally. For example, we now include new results showing disinhibition outperforms excitatory modulation of the rebound circuit in explaining naturalistic bout statistics (**new Fig. 5e**), due to the excitatory model failing to produce bouts with leading sine song (**new Fig. S7h-j**).

Minor points:

I very much appreciate the efforts the authors have made to make the figures easy to understand. Still, it took me a long time to get through this paper. Some things that slowed me down:

11. In Figure 1d, the authors present a heat map to show what type of song the male sings depending on where he is in respect to the female. I did not understand how there can be so many dots behind the male point (are there just instances where he sings to a female completely out of his view?).

We do observe occasional (simple pulse) song at larger angles relative to the female, rarely even when the female is behind the male. Our way of presenting the data (individual semi-transparent dots for each bout in all recordings) visually amplified these rare events. In the **new Fig. 1e** (formerly 1d) we therefore now show p(female location) given simple pulse or complex song, averaged across recordings, in male-centric coordinates, leading to a more representative visualization.

12. Figure 2a, I've stared at that red rectangle for a long while and still don't see how it tells me what the opto stimulation paradigms are. More useful are the (extremely thin) black lines in e.g. fig 2c. But they are never explicitly indicated as the durations of the stimuli, and are often obscured by the probability bars. The intensities of the opto stim are much more clearly presented. I still cannot figure out what the grey lines are in Figure 2b. I can see that they have blips...are they non-song sounds made by the flies (you need to zoom in many times to see them)?

We added 'silence' to the legend of **new Fig. 2b** and increased the stimulus line widths.

I also wondered if we need to see the results of every stim duration and intensity. Might it be simpler to just show the one in the main figs and put the rest in the supplement? And I think I prefer the probability traces at the bottom of Figure 1B to the layout in 1c-f and 4a-c. But that's just a thought.

We thank the Reviewer for this suggestion. We consider the breadth of optogenetic stimulation parameters a unique feature of the present study and therefore decided to leave the results for all stimuli in the main Figure.

13. In Figure 2b, I struggled to understand the relationship between trial numbers, individual flies and the population averages. For each paradigm the y-axis represents trial number, which typically denotes each horizontal line represents a single fly. However, in the single fly probability it seems that there were multiple trials per fly that were averaged to create the fly probability distribution. Does this mean that the population average is the average of individual flies, where each fly had multiple trials?

The reviewer's interpretation is correct, and we have now rephrased the caption of Fig. 2b to better reflect that the population average is the song probability averaged across males.

14. I don't like that the song type probability heat map scales change throughout the paper, for the obvious reason that it makes it harder to compare data between figures.

We agree regarding comparability and now use uniform color map scales across all song probability plots in Figures 2, 4, S2, and S4.

15. In Figure 2f, the authors compare the rebound sine from strong activation of the pIP10 neurons – my first concern with this is that the combination of scatter plot and box and whisker graphs makes it hard to figure out which data set goes with which condition. And there is a statistically significant difference between rebound probability in solitary vs. headless flies, but the means of the two conditions are basically identical – and the authors say in the text that the two conditions are comparable. Is this just a formatting mistake?

The reviewer is correct; the employed Mann-Whitney U (/ rank sum) test in this case is significant despite having nearly identical medians, due to the different tails. We now point out in the figure caption (Fig. 2f) that the distributions are different, although the medians are identical.

16. In figure 3b it would be nice to see example ROIs for the imaging data. Also, are the correlations between neurons the factor is that is most important throughout figure 3? Seems to me that the neurons' response to the stimulation is more important (and more straightforward).

We have added example ROIs (**new Fig. 3d**), and we now emphasize (**lines 137-140**) that anticorrelations between the neurons, that extend beyond the stimulus, are the crucial factor, as these are a signature of the predicted mutual inhibition and rebound excitability.

17. When the authors show that combined stimulation of P1a and pIP10 neurons is sufficient to elicit rebound sine song in Figure 4 a-f, it would be nice to see solitary pIP10 stimulation so that readers can compare all three conditions (only P1a, only pIP10, and then both P1a and pIP10).

We now include solitary pIP10 data in the **new Fig. 4e-f**.

18. In Figures 5c and 5e, using model fit error as opposed to a more explicit measurement of accuracy made things harder for me. For example, in the far case the model only predicts simple p, while in the actual data (in Figure 1c) the other three song options occur with non-zero probability, so that's not very accurate at a qualitative level. In the near case, the model predicts simple p with ~.7 probability and complex s with virtually 0 probability, while in the fly data simple p only occurs with ~0.5 probability and complex s has clearly non-zero probabilities. I think some metric that captures these differences would be more intuitive and informative than model fit error.

We thank the Reviewer for this comment. This slight discrepancy between Figures 1d and 5c is largely due to the model being fit to random samples of the song used to create Fig. 1d. We have added the bout type distributions for this subset of the experimental data to the Supplement (**new Fig S7g**).

Referee #2:

Overall, how switches are implemented at the circuit level is a major question in neuroscience, so this paper has the potential to be of general interest. The experiments are sound and the data are well analyzed. The major problem was the failure to identify the systems of neurons mediating the disinhibition that is at the core center of this paper. As a result, there is a lack of clarity over which conclusions are experimentally grounded and which are hypotheses (albeit sensible ones). Another

issue was confusing sections due to inadequate algorithmic level description of the behavior and some difficulties squaring present results with past work.

(1) LACK OF CLARITY OVER WHAT ARE EXPERIMENTALLY SUPPORTED FACTS AND WHAT ARE INTERPRETATIONS/HYPOTHESES

The take-home model in Fig. 5a has its elegance, its simplicity, its plausibility, its ability to capture some key findings and explain the pulse-sine alternation in complex song and its dependence on proximity to the female. Though it is really a hypothesis and not a summary of results, the paper treats some key features of this model as ‘findings’ with only indirect support. Too many sections and sentences in this paper mention the discovery of a disinhibitory circuit to even mention – and it’s in the abstract – but the inhibition in the brain proposed to link P1a activation to rebound sine in the VNC is still a hypothesis. The same goes for the mutual inhibitory connections within the VNC that might be implementing the switch between the P and the S nodes. There are good ideas here, but they remain speculative. That these details are not really nailed down raises questions and begs any critical reader to imagine alternative scenarios that might work too.

In our study, we proposed that the observed dependence of complex song on P1a activity was due to a disinhibitory circuit motif mediated by P1a, such that P1a activity relieves the rebound circuit from a proposed baseline inhibition that, e.g., prevents involuntary song in the male. Support for disinhibition came from our observation that very strong optogenetic activation of pIP10 neurons is required to induce complex singing in solitary males (Fig. 2c,f) and from our observation that P1a activation does not directly drive song production, but rather a persistent state that promotes song production (Fig. 4a,e). However, as pointed out by this Reviewer, while the proposed circuit architecture containing disinhibition reproduces natural song statistics in our simulations (Fig. 5), this was largely a hypothesis. To provide further evidence for disinhibition, we conducted the following experiments and analyses:

1. We exploited our circuit model to compare the predictive power of an excitatory vs. disinhibitory motif in explaining natural song statistics, by comparing fits to experimental data when using each of these motifs. We found that in line with our expectation, a disinhibitory motif provided the best fits to the experimental song data, compared to an excitatory motif (which produces context-dependent ‘song’ output but fails to produce bouts with leading sine song, in contrast to the disinhibitory model; **new Fig. S7h-j**) and a model lacking disinhibitory or excitatory modulation (**new Fig. 5f**), supporting the conclusion that disinhibition is more likely than excitation to be implemented in the actual song circuit.

2. We analyzed a new female whole-brain connectomic dataset called FlyWire (codex.flywire.ai) for disinhibitory motifs downstream of Dsx+ pC1 neurons (P1a neurons are a subset of male Dsx+ pC1 neurons; Dsx+ pC1 neurons also exist in females, although they lack P1a subset). FlyWire imports predictions about neurotransmitters used at each synapse from (Eckstein et al., 2020). We found that GABAergic disinhibition is a common motif downstream of all subtypes of pC1 neuron in females (here, disinhibition is defined as motifs comprising cholinergic excitation from a given pC1 neuron onto a first GABAergic follower neuron, termed F1, which in turn inhibits a second GABAergic follower neuron, termed F2), with the majority of output synapses

targeting AVLP, AOTU, and PVLP (with around 50k, 30k, and 20k synapses respectively; **new Supp. Fig. S4o-p**), but that these disinhibitory motifs do not necessarily contain Fru+ neurons (only 8/157 F2 followers were Fru+; not shown), making them potentially challenging to identify using genetic intersection strategies. In line with this, a recent study from Amin et al., 2023 shows reductions in sine song for VNC- but not brain-specific knockdown of Gad1 in Fru+ neurons, suggesting the inhibitory neurons mediating disinhibition are not Fru+ (and supporting our findings on the crucial role of VNC-based inhibition in driving sine and complex song). We also found a large number of glutamatergic follower neurons (data not shown; glutamate is mostly an inhibitory neurotransmitter in the *Drosophila* brain, but can occasionally be excitatory, so we did not focus on glutamatergic neurons further).

3. We then used 2-photon calcium imaging and optogenetics to search for a GABAergic disinhibitory motif downstream of P1a neurons in males (using P1a>csChrimson, Gad1>GCaMP, see Table S1-2). Specifically, we imaged activity in all GABAergic neurons in the central brain while activating P1a neurons. If P1a drives a downstream disinhibitory pathway, we expected to observe a series of two inhibitory follower neurons within the brain, with overall strong anticorrelation between the activity of these neurons. In line with this hypothesis, we found ROIs (regions of interest) corresponding to neurons with either activity immediately following P1a activation (we term these 'F1 follower neurons'), or suppressed following F1 follower activity (termed F2 follower; **new Fig. o-p**). We find that the F2 follower neurons are distributed throughout the A-P and D-V axes of the central brain, suggesting the existence of multiple F2 follower neurons for different sensorimotor pathways. In line with this, prior work (Sten et al., 2021) has already suggested the existence (though it remains to be identified) of another disinhibitory pathway downstream of P1a neurons that unlock activity within visual (LC10a) neurons.

4. To further rule out P1a-mediated *excitation* of the rebound circuit as an alternative mechanism to the proposed *disinhibitory* motif, we blocked excitation onto TN1 neurons (by downregulating nACh receptors via RNAi in TN1). If P1a-mediated excitation of sine-promoting TN1 neurons, rather than disinhibition, were responsible for the generation of sine song / complex bouts, we expected to see a reduction in complex song with this manipulation. We found that song statistics were unaffected by downregulation of nACh receptors in TN1 neurons (**included at right**), suggesting that sine and complex song are generated without direct excitation of TN1. Given that we performed several RNAi manipulations that affected complex singing (see above), we interpret this negative result as being in line with our model, in which sine and complex song arise through post-inhibitory rebound dynamics in TN1.

Redacted

Finally, we have also now rephrased the text to emphasize that the proposed disinhibitory circuit motif is a hypothesis backed by certain findings, but that the specific identity of the neurons mediating the functional disinhibition remains to be identified (see changes to **lines 224-226** and **433**). While we did attempt to identify these neurons (within the Fru+ neural population), we were not successful. Indeed, as our analysis of the female connectome suggests, there are likely to be a large number of GABAergic and glutamatergic (likely inhibitory) neurons downstream of P1a neurons. Nonetheless, our new results show that a disinhibitory motif (within the circuit model) provides the best fit to naturalistic bout statistics, and that neural correlates of the proposed circuitry are present among GABAergic neurons downstream of P1a.

Does it really HAVE to be a disinhibition circuit that links P1a to sine generation? I actually agree with the author's interpretation of all of their results – the data are sound and the arguments are sound - but the fact that the necessary inhibitory systems are not identified makes me wonder about alternatives. For example, to me it is not impossible that descending neuromodulatory or peptidergic systems that could be co-released alongside excitatory ionotropic systems could do some heavy lifting in the service of context-dependent inhibition.

We have added new modeling results in favor of disinhibition vs. excitation, showing that an excitatory motif produces context-dependent 'song' output but fails to produce bouts with leading sine song, in contrast to the disinhibitory model (**new Fig. 5f**; **new Fig. S7h-j**), and new experimental results showing brain disinhibitory activity downstream of P1a, in line with our model (**new Fig 4 o-p**). However, we now also discuss descending neuromodulation and peptidergic systems as alternative candidate mechanisms (l. **318-322**).

I don't think these shortcomings sink the paper. A single paper can only do so much towards a mechanistic description of such a complex set of nested behaviors. But I think the paper would be much stronger if it was crystal clear what is experimental result, what is hypothesis, and what future experiments could test that hypothesis.

We thank the Reviewer for these comments, and accordingly we have rephrased the text (l. 192, 204, 211, 322) to improve clarity regarding results, hypotheses, and future experiments.

At the absolute minimum, Fig 5, its legend, and all text needs to be extremely explicit about what is hypothesis and what is experimental result. Any mention of the discovery of a disinhibitory motif should be tempered appropriately, e.g. proposed disinhibitory motif' or 'conditional activation of sine consistent with a disinhibitory process.'

We have made several changes to the main text and the caption of Fig. 5, indicating what is a result, and what is a hypothesis (l. 192, 204, 211, 322).

The slam dunk would be if the authors could actually identify the circuit linking P1a activation to song complexity. Perhaps identification of this system would support the authors' hypotheses. Perhaps not.

We agree that identifying the predicted disinhibitory motif downstream of P1a is a highly desirable next step, but we believe that the number of experiments required to identify these neurons (including identifying genetic driver lines that target the relevant inhibitory neurons, new imaging experiments, and silencing/activation behavioral experiments, and likely male connectomic information) is beyond the scope of the present study. Yet, we have now performed new imaging experiments and modeling to support our prediction. Specifically, in lieu of a male connectome, we screened publicly available *female* connectome data for disinhibitory motifs postsynaptic to the pC1 neural cluster (that P1a is a part of in the male), and confirmed the presence of a large number of disinhibitory motifs (comprised of GABA first and second follower neurons) in the female (**new Fig. S4o-p**). This does not include the large number of GABA-Glu, Glu-GABA, and Glu-Glu follower neurons (glutamate is primarily an inhibitory neurotransmitter in the *Drosophila* brain, but because it can sometimes be excitatory, we did not include it in our analysis). Encouraged by this result, we combined optogenetic activation of P1a with GCaMP imaging from GABAergic neurons in *male* flies, to show that pairs of inhibitory P1a-follower neurons exist with response characteristics expected from the predicted disinhibitory circuit motif (**New Fig. 4m-p**). Interestingly the F2 (second) follower ROIs were distributed throughout the A-P and D-V axes of the central brain (**new Fig. 4p**), suggesting the existence of multiple disinhibitory circuits (as in the female brain, **new Fig. S4o-p**). Prior work on LC10 neurons (Sten et al., 2021) posited, but did not identify, a disinhibitory motif downstream of P1a neurons that should project to the AOTU to disinhibit the LC10 visual neurons. Notably, we find F2 follower ROIs in regions outside the AOTU (**new Fig. 4o-p**). Lastly, we used our model to explicitly show that a circuit comprising a disinhibitory motif shows a superior fit to our behavioral song data than a circuit with a quasi-equivalent excitatory motif (**new Fig. 5f**), and we find that this is due to the excitatory model failing to produce bouts with leading sine song (in contrast to the disinhibitory model and wild-type song (**new Fig. S7h-j**)). These results are in line with our proposed model, although the causal link of the identified neurons with disinhibition of the song circuit remains to be tested in future work.

(2) BETTER DESCRIPTION OF THE BEHAVIOR AND THE PLACE OF THE PULSE/SINE SWITCH WITHIN IT

Courtship interactions and motor programs produced by the male are very complicated and require several ‘switches’ of the type investigated for the present paper (pulse vs sine). A crisp description of the full behavior and the place of the pulse/sine within it is lacking, despite the exquisite analyses that identified pulse fast vs slow (previously) or mf relative positions in relation to song complexity (e.g. fig 4). Of course no one paper is going to ‘solve’ all of these circuits, but an introductory figure, perhaps in supplement, that clearly lays out the full behavior will make it easier for readers, especially young graduate students, to understand exactly what is and what is not at stake in the mechanisms of pulse/sine transitions during complex song.

So I think an algorithmic description of the behavior in the spirit of Marvin Minsky (Society of Mind) would help the reader fully appreciate what exactly is at stake in the current study. Here is my first pass effort to put the pulse vs sine decision in its appropriate context:

- (1) Sing/Court vs do other things (e.g. fly, forage for food)*
- (2) If singing, sine vs Pulse (addressed here)*
- (3-4) If sine, left vs right wing; (4) If pulse, left vs right wing.*
- (5-8) If sine with left wing, lift vs depress wing; (same goes with sine-right wing; pulse left-right wings)*
- (9) If complex song ongoing (or possibly other conditions), initiate Tap or Walk*

(10) If walk pick direction and speed and, suppress tap

(11) If tap, choose left vs right leg

*(12-13) If tapping (see * Below) left leg make contact depress leg (lift vs depress – same with right)*

(14-...) If ???, then mount – and this mount likely needs to aimed which will additionally require directional 'choices'

We have added a schematic modified from Fan et al. 2013 to the supplement (**new Figure S1f**; also see below “to summarize this point”), to orient the reader to the subset of courtship behaviors the present manuscript covers and refer to it in the main text (**I. 25**). We have not included within this schematic all possible behaviors the fly could engage in (foraging, feeding, flying, etc.), but rather behaviors that are part of his courtship activities.

This paper is therefore only about sine vs pulse (#2) – and this is just one of many, equally important in my opinion, 'decisions' that the fly makes during courtship. All of these decisions will be associated with little circuit motifs that will almost certainly involve lateral inhibition – as this is a basic principle of CPG function going back decades. Given this, the authors can better motivate why this small slice of the courtship behavior really provides the key and/or the experimental tractability to get at a context dependent switch. The authors attempt to do this by emphasizing the dependence of complex song on sensory feedback (the presence of and distance to female, and the fact that known circuits may implement this (PC2 and P1a)). To summarize this point, the paper in its current form immediately zeros in on the pulse-sine decision without the perspective of the whole behavior in which this decision is embedded. I think this issue could be addressed with a call to supplemental text and perhaps a supplemental figure that puts this question in broader context.

We now include an introductory schematic (**new Fig. S1f**) to put the pulse-sine decision into a broader context, and we point out (**I. 25**) that at any moment, flies face a large number of decisions to make (whether to feed, forage, escape, court, fight, sleep, etc.), and these decisions are influenced by internal states and sensory feedback.

question about the taps: When does tap happen in song - related to p or s or ps boundaries. It is shown that the tap rate is higher for complex bouts and at close distances - but there is some other CPG driving the tap? And is the tap with the left or right front leg? And does this depend on which wing? The animal cannot tap with both - so a similar winner-take all process must determine which leg taps. Are these all nested? This question about taps is not meant to motivate more experiments as it is outside the scope of the paper but **simple text/intro to the tap system/behavior is necessary in advance of the section on taps.*

We have expanded the introduction on tapping and the likely circuitry mediating it (line 169ff). Specifically, males tap the female abdomen with one or two foreleg tarsi to initiate and maintain courtship (Rendel 1945; Spieth 1952). Further, tapping activates P1a neurons in the brain that drive a persistent arousal state via downstream recurrent circuitry (Clowney et al., 2015; Jung et al., 2020).

(2.2) The results section around Fig 2, the 2p imaging in the VNC is complicated by the unstated fact that there are really two distinct all-or-none action selection 'decisions' being made. Sine vs pulse (as

motivated in the text) but also left pulse vs right pulse and left sine vs right sine.) If either wing can produce either song, and if this 'decision' is under sensory influence (e.g. where the female is) then one can imagine a perfect ensemble of neurons with activity anti-correlated with the pIP10 activation even in a pulse-only context given competition between left and right wings. It's confusing in the text how this issue was addressed experimentally or conceptually.

For example, in the TN1 imaging datasets the anticorrelated populations – could they not also be related to driving left or right wings? Were the authors imaging both sides of the VNC? This section was very confusing because the authors seemed to be assuming that these neurons were pulse- or sine-driving. Perhaps there was reason for this but as I read the section I was considering an alternative that they had zeroed in on the left-vs-right decision circuits.

We have added text (l.74-75 and the **new Fig. S1f** in response to comment 2.1) to clarify that wing choice is not part of our model. Data from the lab reveals that wing choice circuits are distinct from those involved in song patterning, and that song patterning activity is present in both hemispheres of the VNC, although the male typically sings with one wing at a time (unpublished). The anticorrelated pair of TN1 neurons shown in Figure 3 (and all pairs in the summary panel) were recorded in the same hemisphere, and we have added text (main and figure caption) to clarify this. Therefore we believe the observed signatures of mutual inhibition are not involved in wing choice but the production of song.

*(2.3). This paper is filled with very intense photoactivations of premotor circuits and as it is written all the authors observed were 'normal' patterns of pulse and sine. But there is a lot to be learned about underlying circuits if there was (or if there never was) non-ethological 'garbage' that came out of photoactivations. For example, did photactivations ever cause (1) bilateral singing? (2) a weird combination of pulse and sine that was not interpretable? The reason this matters is that some circuits would make the all-or none condition a necessity (e.g. as in **cricket stridulation vs flight**) but others may allow co-activations (e.g. as in flexor extensor co-activation during locomotion or in dystonia). Related to this point – the authors write in Line 117: "temporally separated pulse and rebound sine" The presence of this qualification is confusing – it makes me wonder if pulse and sine can ever NOT Be temporally separated, i.e. did photoactivation of any line ever drive indiscernable behaviors that may show that co-activation of both pathways? Co-activation of both pathways under non-ethological conditions of genetically targeted photoactivations can reveal to what extent the two systems are necessarily all-or-none.*

We now emphasize that photoactivation was always bilateral (activating neurons in both hemispheres), and we did not consider wing choice here (l. **72-75**). Yet, to address the Reviewer's questions: 1) we find that bilateral photoactivation in solitary males mostly, but not exclusively, causes more bilateral song, whereas a large proportion of female-directed wild-type song is produced with one wing (**Figure below**) - this does not disrupt the song itself (the temporal and frequency characteristics are unperturbed whether the song is generated by bilateral versus unilateral wing extension). 2) Throughout our photoactivation experiments, we never observed superpositions of pulse and sine song.

We have further rephrased the text to clarify that here, 'temporally separated' means pulse and sine song occur (in behavioral experiments) at well-defined times relative to the stimulus, such that the measured calcium fluorescence can be mapped to these time bins.

(2.4) Is near vs far also a switch or is it analog?

Finally, the near vs far state is also written about as a binary switch in paragraph at 347, which would also need a binary adaptive switch circuit - but I don't see this in the circuit diagram and it also does not square with the fact that in photoactivations P1a and pIP10 both seem to function in analog fashion - in which the amplitude and duration of their activations have effects on behavior. Neither of these brain neuron groups appear to function as all or none. So this section is very confusing. Please clarify.

Related to Reviewer 1's comment, we have moved panel b from Supplementary Figure 1 to Figure 1c and added text (I. 51-52) that although bout composition is a smooth function of distance, for simplicity we consider near and far two distinct social contexts throughout the manuscript.

(3) CONFUSING RESULTS (OR POSSIBLY CONFUSING PRESENTATION OF RESULTS) IN SOME SECTIONS

(3.1). In line 168 it is suggested that PC2 is primary driver of both Pip10 and P1a and therefore at the top of a hierarchy to drive complex song, especially close to a female. This 'suggestion' - predicts very clearly that pC2 photoactivation would co-activate pIP10 and P1a. Is this the case? Although I am loathe to suggest more experiments as this is already a very full paper, PC2 activation of pip10 and p1a is a strong prediction of the model – and perhaps it's already been done. But I cannot find this result in the paper or cited. And the experiment seems straightforward.

We agree that imaging from P1a following activation of pC2 would be an appropriate, albeit challenging (with the existing genetic reagents), experiment to test our model. Due to these challenges, we have instead recorded song of males with genetically inactivated pC2 neurons. These males showed strongly reduced amounts of song compared to controls, and the majority of the remaining song corresponded to simple pulse song (**new Fig 4q-r**). These results strongly support our model, in which pC2 constitutes the top of the circuit hierarchy. Recent work (Lillvis et al. 2022) has further used expansion microscopy to confirm our proposed connection between pC2 and pIP10.

(3.2). Also I'm very confused by this assertion of PC2 at the top of the hierarchy because past work (Deutsch et al, 2019, by the authors) shows that complex song can exist, even be increased, following

PC2 deletions. And it was also stated in that paper that no sine song was seen following PC2 activation (this is not the case; Deutsch et al. observed pulse song followed by sine song after activation of pC2 (e.g. their Fig. 5A), but do not discuss this result), which also seems at odds with the present arguments. Lastly, this paper suggested that PC2 actually inhibited sine song – the exact opposite of what seems to be being argued here. Perhaps the authors have an understanding of how to integrate these observations together, but as it currently stands this is causing a level of confusion extremely distracting for a field outsider.

We thank the Reviewer for pointing out the discrepancy between our model and prior results (*Deutsch et al. 2019*). The experiments by *Deutsch et al.* were performed in a different behavioral rig, so we re-ran the pC2 silencing experiments in our new setup (and re-generated the genetic strains, but used the same driver lines to target pC2 - see Fig. S5). For the new experiments, we confirmed the presence of TNT in all crosses using PCR. We found that blocking chemical synapses (via TNT) in pC2 neurons **decreased the total amount of song, including sine song** (rather than the increase reported in *Deutsch et al. 2019*). We now point out the difference between the two studies on **line 196-197**. While these new results do not alter the major findings of *Deutsch et al. 2019*, we are preparing a corrigendum to *Deutsch et al. 2019*.

Analysis of song statistics in pC2>TNT males revealed that song in silenced males was biased to simple pulse bouts both far from and near the female (less complex song both far from and near the female) (**new Fig. 4q-s**). This is **in line with our model**, because the proposed connection from pC2 to P1a is blocked, which functionally prevents disinhibition of the rebound circuit downstream, and hence is expected to suppress the production of complex song (pulse-sine alternation song). Why is pulse song still produced in these males since our model also posits that the pC2-pIP10 connection is the primary pathway for driving simple pulse song? This could either be due to additional electrical synapses from pC2 to pIP10 that remain intact, or to other inputs that drive pIP10 (e.g., male self motion, independent of vision, is known to drive pulse song (Coen et al. 2014)). Lilvis et al. 2022 have recently confirmed the existence of chemical synapses from pC2 to pIP10 (cited on pg 5), but they found that structural connectivity from pC2 to pIP10 (ie chemical synapses) is less predictive of optogenetically driven song than functional connectivity (ie activating pC2 and imaging pIP10), suggesting the presence of additional (electrical) synaptic connections.

(3.3) Paragraph 492 and Fig 4: This is a very clever experiment - photoactivation of pIP10 with the female absent produces pulse only. Female presence increases bout complexity. This experimental design enables precise identification of what exactly about the female presence makes bout complex - which may be thought of as a behavioral way of testing for PC2 activation.

But to test these ideas, wasn't it more direct to just image the PC2 neurons and build a predictive model of what aspects of female position/body/orientation drive the neuron? And wasn't this the punchline of Deutsch et al? I cannot tell what is conceptually new about this whole section 192-212. There is a complex behavioral design, complex results, and figure panels that require head scratching. At the end of the day - wasn't it already apparent from the pc2 activation results that multimodal sensory cues control bout complexity? It was already argued that pc2 is primarily vision/sound, and p1a is primarily tap - both can modulate bout - so those results seem redundant with this whole section. Unless I'm missing an important motivation for these experiments, I think they can be moved to supplement to streamline this already large paper.

We thank the Reviewer for their comments. We agree that imaging from pC2 during free courtship would be an excellent experiment to test our model, but we emphasize that this experiment is highly challenging: it requires calcium imaging from pC2 neurons in males that are in a courtship-like state and perceiving naturalistic visual cues. We have begun such experiments in the lab, and are working on setups that will allow head-fixed males to *consistently* engage in fictive courtship. In one such experiment, we imaged from pC2 and P1a neurons in head-fixed males walking on a spherical treadmill and presented with a live female stimulus and a fictive moving visual cue. Preliminary analysis show that primarily the activity of P1a follows that of pC2, in line with the conclusion that pC2 is upstream of P1a (**see Figure below**).

Redacted

We also want to stress that while the Reviewer is correct in pointing out that it was known before that pC2 is both responsive to visual and auditory stimuli, its parallel functional connection to both pIP10 and P1a, that our experiments uncovered, was previously unknown. Further, it was previously assumed that the primary role of P1a was to mediate a minute-long, ‘persistent’, courtship state. Our results demonstrate that P1a has another role in modulating song output at much faster timescales, since the coincident activity of P1a (through tapping or optogenetic activation) can transform pIP10-driven simple song to complex song. These results were key in forming our circuit model and, while building on prior work, moved forward our understanding of context-dependent song patterning.

(3.4) Fig 4n-o shows that behavioral priming, presumably instantiated via p1a activation, reduces threshold of pip10 mediated pulse initiation, but without change in complexity. Please relate this to Fig4d (coactivation of p1a and pip10 - here, it is shown that coactivation does increase complexity). Is what matters magnitude of p1a activation? Is this correct? Is the y axis of fig4d the laser power to both of the neuron types?

We have added data from weak co-activation of P1a and pIP10 to the comparison in the new **Fig. S4j**, to further clarify that the acute activity of P1a, not the persistent effect of its activation, transforms simple song to complex song. To specifically address the Reviewer’s first question, it is not the magnitude but the (concurrent) timing of P1a activation that matters for the switch from simple to complex song. Regarding the Reviewer’s second question: for optogenetics, we used LED illumination of the entire fly at the specified irradiance levels, so the amount of irradiance used for activation of P1a or pIP10 alone matches that used for activation of P1a or pIP10 during coactivation.

What was the pip10 irradiance in 4d? I cannot square 4d with 4n.

We have added irradiance labels to the y-axes of all matrix plots in Figure 4 (irradiance levels are consistent across plots). We now emphasize that it is the *_acute_* activity of P1a that needs to coincide with pIP10 activity (Fig. 4d), rather than the persistent, minute-long, courtship state that is initiated by P1a (Clowney et al., 2015; Hoopfer et al., 2015; **new Fig. S4j**), that matters for bout complexity.

(4) MINOR

Abstract: Mutual inhibition and rebound excitability are not really computations (nothing is computed per se, signals are simply propagated in a certain way) - rather they are mechanisms that support a wide array of computations. I recommend replacing computations with mechanisms

We agree and replaced “computations” with “mechanisms” (or “computational features” where appropriate) throughout the manuscript.

Line 100: reciprocal inhibition of the inh nodes would do much more than prevent 'premature bout termination' (which is a strange and new concept in the flow of the paper, not even defined. If the authors want to motivate any points about premature bout termination they should show (or cite) bout duration distributions and make the prediction that eliminating the reciprocal inhibition between the inh nodes reduces bout duration and/or reduces the toggling between nodes. the real utility of reciprocal inhibition is the adaptive switch - robust toggling and all or none ax selection across a wide range of excitatory drives to each node.

We have removed the sentence about premature bout termination and instead emphasize that the proposed circuit architecture implements an adaptive switch (Mysore & Knudsen, 2012; Sharpee 2012), which allows for robust toggling between song modes and all-or-none action selection across a wide range of excitatory drives to each node.

Line 103: if pip10 is a descending neuron - so how did it stick around following decapitation? Are you photoactivating the terminals in the VNC? If so, please clarify in main text.

We now emphasize that we're activating the terminals of pIP10 in decapitated males.

line 154 change control to initiation

done

Section at line 146 and line 159: Unclear description of results. The terms variable and complex are poorly defined. If complex is pulse and sine, what then is variable?

We rephrased this sentence.

What are the black boxes in the rows of Fig 2c,e,h,j?

We believe these boxes were artifacts of our illustration software and hope these are fixed in the updated manuscript.

Line 162: levels is a poor word choice because it can mean amplitudes or durations and here you mean distinct durations.

We replaced 'levels' with 'durations'.

line 166 change 'the' to 'a' - could future experiments conceivably identify a distinct and possibly redundant or even primary controller?

We replaced 'the' with 'a'.

line 236. Unless I am missing something, the evidence for a disinhibitory pathway is simply that p1a activation alone could not strongly initiate song initiation but moderately affected probability of complex song. If this is correct, then the evidence for the disinhibitory pathway is really weak - at this point it's still a hypothesis.

We have performed new imaging experiments and modeling in support of a disinhibitory circuit motif. Specifically, in lieu of a male connectome, we screened publicly available female connectome data for disinhibitory motifs postsynaptic to the pC1 neural cluster (that P1a is a part of in the male), and confirmed the presence of disinhibition in the female (**new Fig. S4o-p**). Encouraged by this result, we combined optogenetic activation of P1a with GCaMP imaging from GABAergic neurons in male flies, to show that pairs of inhibitory P1a-follower neurons exist for which the activity is anticorrelated, as expected from the predicted disinhibitory circuit motif (**new Fig. 4o-p**). Lastly, we used our model to explicitly show that a circuit comprising a disinhibitory motif shows a superior fit to our behavioral song data than a circuit with a quasi-equivalent excitatory motif (**new Fig. 5f**), and we find that this is due to the excitatory model failing to produce bouts with leading sine song (in contrast to the disinhibitory model and wild-type song (**new Fig. S7h-j**)). These results are in line with our proposed model, although the causal link of the identified neurons with disinhibition of the song circuit remains to be tested in future work.

Line 341: The authors write 'weak' but don't they mean 'brief'? It's important to distinguish amplitude versus duration of activation, as these arise from distinct behavioral events - and words like strong (weak) can mean either high (low) amplitude or long duration. Precise terminology is essential here.

We have replaced all instances of 'weak' with 'brief' where we refer to short-duration activity.

Referee #3:

The male fruit fly unilaterally vibrates a wing as part of its courtship display. Prior studies have established that these courtship "songs" can be produced in two major modes: pulse song, which is louder; and sine song, which is quieter. It is also established that males predominantly generate pulse songs when distant from the female, then increase the proportion of sine songs with increasing

proximity to the female. Consequently, as a male actively pursues a female, it produces song bouts comprising simple songs (pulse or sine) or complex songs (where they rapidly alternate between pulse and sine).

We thank the Reviewer for their feedback and now emphasize in the main text (**I. 373-376**) that the previously established view was that song consists of two major modes, 'pulse' and 'sine', with pulse being the dominant mode far from the female, and sine probability increasing closer to the female. However, the classification of song into 'simple' and 'complex' bouts, rather than 'pulse' and 'sine', was not established by prior studies but crucial for our interpretation and identification of the circuit mechanism underlying song patterning. Specifically, while the established 'pulse'/sine' view would favor circuit models with dedicated pathways for each song mode (as discussed in Clyne & Miesenboeck 2008), our study revealed that sine song rarely occurs without accompanying pulse song. This observation led us to the necessary experiments to conclude that a dedicated pathway for pulse song exists, but there is no corresponding pathway for sine song. Instead, the large majority of sine song is driven indirectly, via post-inhibitory rebound following release from inhibition by the pulse pathway.

Roemschied et al. use a combination of behavioral analysis, optogenetic manipulations, neural imaging and computational modeling to study how contextual cues regulate the transition from simple to complex songs. They provide new insights into the sensory cues that regulate this transition, and also provide some evidence for circuit mechanisms that regulate the context-dependent switch from simple to complex songs as well as the alternation between pulse and sine songs during complex song bouts. An important problem addressed here is how sensory cues, especially those provided in different social contexts, alter behavior. Such context-dependent changes are features of locomotion and communication. In this latter arena, studies in humans, non-human primates and songbirds all reveal effects of social context on the quality and quantity of vocalizations. The fly provides a valuable organism in which to address explicit circuit mechanisms by which different sensory contexts unlock different motor circuit dynamics in the framework of sexual signaling and communication. In fact, the fly courtship song behavior has been mined extensively to rigorously quantify acoustic signatures of song; identify numerous sensory cues during courtship that regulate the amount and type of singing; localize neurons and circuits for pulse and sine song production; distinguish central neurons, motor neurons and flight muscle specializations for producing pulse or sine song; and the list goes on from there. Indeed, many of these advances are reported in prior publications from the senior author's lab. More broadly, functional, genetic and comprehensive anatomical (ie, connectome) analyses in fly adults and larvae have provided detailed circuit models for sensory processing and motor control, including context-dependent changes to behavior.

We agree that the prior knowledge of neurons involved in song production facilitated our circuit study. While we knew from these prior studies which neurons were involved in driving song, we did not know how the circuit was organized to generate context-dependent song. As the Reviewer points out, it is the basis for the context-dependence in varying song patterns that is under study here, and which is of broad relevance to the field. We emphasize that while Coen et al., Nature 2014 was pioneering in uncovering the sensory modulation of song statistics, in a pulse/sine-centric interpretation, we here provide a re-interpretation of song generation in a simple/complex framework, and we provide a mechanistic explanation of how context-dependent song production is achieved at the neural circuit

level. We further agree that the advent of connectomic information has begun to boost our knowledge about circuits underlying sensory processing and motor control, including context-dependent changes to behavior in *Drosophila* larvae (e.g. Jovanic et al., 2016). Yet, for adult *Drosophila* we still lack a whole-brain connectome (papers from FlyWire are not yet published) and we do not yet have any connectomes (partial or full) for the adult male brain or VNC. Since many of the neurons involved in song behavior are male-specific, the crucial information is not available from public **female** connectomic resources (hemibrain or FlyWire). **We further emphasize that even the availability of a male connectome (for the brain or VNC) would have been insufficient to uncover our proposed circuit mechanism**, as, for example, the central involvement of rebound excitability (via I_h) would have been invisible in connectome data. In particular, without our functional dissection of the male song circuit, knowledge of the wiring diagram alone would not have provided knowledge of how individual nodes contribute to complex vs. simple song.

It is on this background that this reviewer, who is not a fly researcher, tried to identify the novel aspects of the present study and assess their significance relative to earlier published work. As usual from the Murthy lab, many aspects of the present study are rigorously executed, including the careful characterization of social cues that regulate the switch from simple, pulse-only songs to complex songs. Another nice aspect of the study is the use of an optogenetic stimulation protocol that spans a wide range of light intensities and duty cycles to elicit sine song following prolonged optogenetic evocation of pulse song. The authors are also to be commended for further characterizing and modeling sensory cues that help to drive simple and complex song. Lastly, the integration of modeling methods to test certain compact circuit architectures that might be sufficient to explain context-dependent regulation of song is also a strength.

However, the authors did not do a very good job of underscoring the major advances of the present study relative to early published research on this and related topics.

We thank the Reviewer for the positive comments and agree that we did not do a sufficient job of highlighting what was novel in this study, and we have now made a number of changes to the text (highlighted below) to make this clearer.

Moreover, while the breadth of approaches employed here is impressive, and I have great respect for the rigor of the behavioral analyses and the extension to circuit modeling, the circuit characterization itself is a bit thin and overly speculative, falling short of the leading efforts to analyze behaviorally-relevant central circuits in the fly. Along with a somewhat indirect style of exposition and argument, perhaps especially so for a Nature article, I was not left with the impression that I was reading about a major advance in this area of research. Perhaps some of this can be addressed by a more direct style of writing that clearly underscores what is new and why it matters. But even with such clearer writing, I believe that there are experimental gaps that lessen the potential impact of this study. Here I highlight some of the ways in which the current manuscript falls short.

We thank the Reviewer for their feedback - we agree (and also with the other two Reviewers who made similar comments) that we lacked sufficient experimental evidence to fully support our model. We have therefore addressed this concern with the following new analyses and experiments:

Regarding the proposed role of disinhibition in facilitating complex song production:

Support for disinhibition originally came from our observation that very strong optogenetic activation of pLP10 neurons is required to induce complex singing in solitary males (Fig. 2c,f) and from our observation that P1a activation does not directly drive song production, but rather a persistent state that promotes song production (Fig. 4a,e). However, as pointed out by this Reviewer, while the proposed circuit architecture containing disinhibition reproduces natural song statistics in our simulations (Fig. 5), this was largely a hypothesis (which we now clarify, l. 204, 316). To provide further evidence for disinhibition, we conducted the following experiments and analyses:

1. We have now exploited our circuit model to compare the predictive power of an excitatory vs. disinhibitory motif in explaining natural song statistics, by comparing fits to experimental data when using each of these motifs. We found that in line with our expectation, a disinhibitory motif provided the best fits to the experimental song data, compared to an excitatory motif (which produces context-dependent 'song' output but fails to produce bouts with leading sine song, in contrast to the disinhibitory model; **new Fig. S7h-j**) and a model lacking disinhibitory or excitatory modulation (**new Fig. 5f**), supporting the conclusion that disinhibition is more likely than excitation to be implemented in the actual song circuit.

2. We analyzed a new female whole-brain connectomic dataset called FlyWire (codex.flywire.ai) for disinhibitory motifs downstream of Dsx+ pC1 neurons (P1a neurons are a subset of male Dsx+ pC1 neurons; Dsx+ pC1 neurons also exist in females, although they lack P1a subset). FlyWire imports predictions about neurotransmitters used at each synapse from (Eckstein et al., 2020). We found that GABAergic disinhibition is a very common motif downstream of all subtypes of pC1 neuron in females (here, disinhibition is defined as motifs comprising cholinergic excitation from a given pC1 neuron onto a first GABAergic follower neuron, termed F1, which in turn inhibits a second GABAergic follower neuron, termed F2), with the majority of output synapses targeting AVLP, AOTU, and PVLP (with around 50k, 30k, and 20k synapses respectively; **new Supp. Fig. S4o-p**), but that these disinhibitory motifs do not necessarily contain Fru+ neurons (only 8/157 F2 followers were Fru+; not shown), making them potentially challenging to identify using genetic intersection strategies. We also found a large number of glutamatergic follower neurons (data not shown; glutamate is mostly an inhibitory neurotransmitter in the *Drosophila* brain, but can occasionally be excitatory, so we did not focus on glutamatergic neurons further).

3. We then used 2-photon calcium imaging and optogenetics to search for a GABAergic disinhibitory motif downstream of P1a neurons in males (using P1a>csChrimson, Gad1>GCaMP, see Table S1-2). Specifically, we imaged activity in all GABAergic neurons in the central brain while activating P1a neurons. If P1a drives a downstream disinhibitory pathway, we expected to observe a series of two inhibitory follower neurons within the brain, with overall strong anticorrelation between the activity of these neurons. In line with this hypothesis, we found ROIs (regions of interest) corresponding to neurons with either activity immediately following P1a activation (we term these 'F1 follower neurons'), or suppressed following F1 follower activity (termed F2 follower; **new Fig. o-p**). We found a number of F2 followers neurons

which are distributed throughout the A-P and D-V axes of the central brain, suggesting the existence of multiple F2 follower neurons for different sensorimotor pathways. These experiments confirm that there is brain disinhibition circuitry downstream of P1a activation (which in behaving males drives complex song production, in concert with pIP10 activation).

4. To further rule out P1a-mediated *excitation* of the rebound circuit as an alternative mechanism to the proposed *disinhibitory* motif, we blocked excitation onto TN1 neurons (by downregulating nACh receptors via RNAi in TN1). If P1a-mediated excitation of sine-promoting TN1 neurons, rather than disinhibition, were responsible for the generation of sine song / complex bouts, we expected to see a reduction in complex song with this manipulation. We found that song statistics were unaffected by downregulation of nACh receptors in TN1 neurons **(included at right)**, suggesting that sine and complex song are generated without direct excitation of TN1. Given that we performed several RNAi manipulations that did affect complex singing (see above), we interpret this negative result as being in line with our model, in which sine and complex song arise through post-inhibitory rebound dynamics in TN1.

Redacted

Finally, we have also now rephrased the text to emphasize that the proposed disinhibitory circuit motif is a hypothesis backed by certain findings, but that the specific identity of the neurons mediating the functional disinhibition of the song pattern generating circuits remains to be identified (see changes to **lines 224-226** and **433-434**). While we did attempt to identify these neurons (within the Fru+ neural population), we were not successful. Indeed, as our analysis of the female connectome suggests, there are likely to be a large number of GABAergic and glutamatergic (likely inhibitory) neurons downstream of P1a neurons. Nonetheless, our new results show that a disinhibitory motif (within the circuit model) provides the best fit to naturalistic bout statistics, and that neural correlates of the proposed circuitry are present among GABAergic neurons downstream of P1a.

Regarding the proposed role of rebound excitability in driving sine and complex song:

See our comments above - We now show that mutant males systemically lacking *Ih* (a major ingredient for rebound excitability) exclusively sing simple pulse song, and further that TN1-specific knockdown of *Ih* and *Rdl* (GABA-A receptors) both reduce song complexity (**new Fig. 3f-k**). These results strongly support our model in which complex song arises via endogenous rebound circuitry and GABA-A mediated mutual inhibition.

In addition, we have made a number of changes to the text to emphasize the novelty of our results. While in the lab, we already perform *in vivo* imaging from sensorimotor circuits during behavior, we have not yet recapitulated context-dependent changes in song sequencing (simple song far, complex song near) in head-fixed flies. Nonetheless, we have recorded whole-brain activity from males in a

courtship state, and found that primarily the activity of P1a follows that of pC2, in line with the conclusion that pC2 is upstream of P1a.

Redacted

1. Simple versus complex song. Both in the abstract and section 2 of the main text, the authors describe context-dependent changes from simple to complex song. From a quick scan of earlier studies, this transition from pulse only to pulse-sine song as the male engages the female courtship has already been described. While I understand that the authors need to show the behavior (again) to set up later stages of the study, it is presented in a manner that suggests novelty (this is especially so in lines 10-12 of the abstract, but is also an issue with much of section 2).

[See response to Reviewer 3's first comment above] Indeed previous work (e.g. Coen et al. 2014) had reported that the probability to sing sine song increases at short male-female distances. We now emphasize that prior studies speculated that the basic units of song are pulse and sine, and therefore require dedicated pathways for its neural control (as discussed in Clyne & Miesenboeck 2008). Our findings instead support the conclusion that the basic types of song are 'simple pulse' and 'complex'. While this distinction may seem minor at the surface, it was key in pointing us towards a model in which sine song is driven indirectly (through rebound excitation) and almost exclusively produced in combination with pulse song, and hence does not require a dedicated command neuron such as pIP10 for pulse song. To our knowledge, this is a novel circuit mechanism previously unknown for social / communication circuits.

2. The supposition that complex song may be more attractive to the female is a bit off target in two ways. First, prior studies have established that sine song is important for the male to achieve copulation and, as complex song contains sine song, its attractiveness relative to pulse song alone would seem to be a given. But the notion that complex song, and particularly the alternation between pulse and sine song, is even more effective than sine song at male mating success is unknown, at least as I can discern. I am uncertain as to how the authors could separate the efficacy of sine song versus alternating pulse-sine song in this regard, but at the very least they need to test this idea. This is an example where speculation intrudes into the line of scientific argument in a disruptive way.

We agree that this result is confusing at this point in the manuscript. We wish to clarify that our intent was to speculate about a potential function of complex song, since to date the precise role of sine song (mainly present in complex bouts) remains unknown, apart from one study suggesting that it increases female receptivity (von Schilcher 1976). Recent circuits work has focused on neurons tuned to the male

pulse song that drive female receptivity (Wang et al. Nature 2020), female changes in locomotion (Deutsch et al. CB 2019), and female rejection (Wang et al. Curr Biol 2020). Wang et al. Nature 2020 showed that pulse song, but not (isolated) sine song, promotes female vaginal plate opening (which facilitates copulation). Yet, we find that the majority of males sing bouts containing sine song followed by pulse song (hence, complex song) immediately preceding copulation (Fig. S1b). A recent study identified for the first time auditory neurons tuned to sine song (Baker et al. 2022) and future work should be able to determine how these neurons, together with neurons tuned to pulse song, drive changes in female behavior to song sequences. Whether complex song plays a role in promoting / enhancing vaginal plate opening and copulation success remains to be tested in future work.

3. In the second paragraph of section 2, the finding that pulse song predominates at greater distances is presented as if it is a new finding. However, this has been documented previously (see Trott et al. 2012, for example). Perhaps it is not the authors intent to present this as a novel finding, and the analyses here are more sophisticated, but this is an example of how it is difficult for an outside reader to glean what exactly is novel in the present study. It would help to have the authors further distinguish the novel elements presented in figure 1 overall.

We thank the Reviewer for pointing this out. We now cite Trott et al. (with Coen 2014; l. 42). Trott et al quantified the relative proportion of pulse/sine song within 30s windows, thus missing the fine structure of bouts. It was previously shown that the proportion of pulse song increases, and the proportion of sine decreases, with distance (Trott et al.; Coen et al. 2016; Clemens et al. 2018), but in none of these studies was it possible to draw conclusions about the circuitry, eg. labeled lines for pulse and sine vs. other solutions. In our case, several aspects of the detailed bout statistics were important to build the foundation of the circuit model: 1) the majority of bouts started in pulse mode, and 2) at short distance the proportion of rapid pulse-sine alternations increased, suggesting the mutual inhibition / rebound motif (this would not have been the case if e.g. simple sine bouts occurred more often at short distance, in addition to simple pulse bouts).

4. Same section (lines 52-54), the authors focus on longer song bouts to invoke inhibitory mechanisms, which would act in part to suppress singing when no female is present. While that is one possible explanation, it seems just as reasonable to assume that in the absence of female cues, the song generating circuit is simply offline, perhaps because the motor elements have a high threshold for activation. Excitation/arousal generated by those female cues then kicks an otherwise dormant circuit into action; no inhibition is needed.

We have collected further evidence for the proposed inhibitory motifs. First, we have used our model to show that a disinhibitory motif outperforms quasi-equivalent excitation in reproducing naturalistic bout statistics (**new Fig. 5f**). Second, we have performed new imaging experiments to show the existence of disinhibitory activity in the brain and postsynaptic to P1a, as proposed in our model (**new Fig. 4o-p**). Third, we have performed new experiments showing the requirement of I_h for complex song production, and specifically rebound excitability of TN1 neurons for sine production (**new Fig. 3f-k**). Yet, identifying the precise neurons involved in disinhibition of the song circuit (which will require large-scale behavioral screens of sparse driver lines) still awaits further study, for which our work provides the essential foundation.

5. Dual versus single stream control of song patterning has been bandied about in the fly song literature for some time. On the most fundamental level, the central thoracic neurons, the motor neurons and the wing muscles for pulse and sine song are distinct (e.g., von Phillipsborn, 2011; Shirangi et al, 2013, 2016), and flies never generate pulse and sine song at the same time, at least from what I can see in this and other studies. Therefore, there must be some form of cross talk at some level of the neuromuscular control pathway to explain this “one at a time” behavioral output, regardless of whether there is single or dual channel descending control. Further, as males sing predominantly pulse songs at the first stage of the interaction with the female (at a distance), it seems that fatigue in the pulse-generating circuitry could relieve feedforward inhibition on the sine generating circuit. A winner takes all mechanism could account for the behavior regardless of whether descending control architecture is single or dual channel.

We thank the Reviewer for this comment. We agree that a winner-take-all mechanism could explain pulse-sine switching in a dual channel model, but we believe it would fail to explain e.g. rebound sine song following activation of pIP10 (the ‘pulse channel’) when no female is around (Fig. 2b,c,e), because there is no direct drive of the sine circuitry, and hence there is no latent sine activity that then is released following fatigue of the pulse pathway. Our initial results instead supported a model in which pulse-sine alternations arise from rebound excitability, which we further confirmed with new experiments (targeting rebound excitability system-wide, in I_h mutants, and TN1-specific, via knockdown of I_h and Rdl , **new Fig. 3f-k**).

We had not explicitly addressed fatigue before, but we believe this will be an important aspect in understanding variability of bout *duration*, which we do not cover in the present manuscript. In preliminary experiments with 10s-long optogenetic activation of pIP10 in solitary males, we observe that after an initial stereotyped pulse bout, a transition to sine song occurs already during ongoing stimulation (**at right**). This supports the conclusion that pIP10 or downstream neurons along the pulse pathway exhibit fatigue (via spike-frequency adaptation, for example), which allows for brief rebound activity in the sine-driving neurons due to the fatigue-induced release from inhibition. Yet, the ongoing stimulus immediately allows pIP10 to take over again to drive the next pulse train. Also, the inter-pulse interval (IPI) clearly increases during the initial pulse train before the transition to sine-pulse switching, another signature of spike-frequency adaptation.

6. While the optogenetic protocol using a range of irradiance and duty cycles is a nice addition, figure 4B3-D3 from Clemens et al. (Current Biology, 2018) shows optogenetic stimulation of pIP10 can evoke pulse song in solitary males, which then transitions to sine song after the light pulse is turned off. It's

fine to build on that earlier work in more detail, but the authors should cite their own prior work on this point, as the observation of optogenetically-evoked complex (pulse-sine) song is not novel.

We have added a reference to Clemens et al., 2018 to the main text (I. 83) and state that pulse-sine sequences following pIP10 activation were observed but not discussed in that paper.

7. That the optogenetic regime in solitary males only weakly recapitulates the complex song produced in the presence of a female raises the question of additional pathways operating in parallel to pIP10 or TN1. The headless fly experiment tells us that the additional drive depends on the head, which is further explored in Figure 4. But here in the experiments shown in Figure 2, is there a concern that prolonged optogenetic stimulation itself generates a rebound-off effect independent of any endogenous rebound circuitry?

We now include additional evidence for endogenous rebound circuitry that directly addresses this point (**new Figure 3f-k**). Specifically, we show that mutant males lacking *Ih* (a major ingredient for rebound excitability) exclusively sing simple pulse song, and further that TN1-specific knockdown of *Ih* and *Rdl* (GABA-A receptor) both reduce song complexity. These results strongly support our model in which complex song arises via endogenous rebound circuitry.

8. Line 84: As per my earlier comment, I don't see how the authors can distinguish between a model where the sine generator is under tonic inhibition versus one where it simply has a high activation threshold. While citation of the work from Tuthill on VNC inhibition of proprioceptive signals is helpful, the lack of any information about such resting inhibition in sine generating circuits is a weakness of the present study.

We now used our model to show that disinhibition outperforms excitation in explaining naturalistic bout statistics (**new Fig. 5f**). We assume that in a model with high activation threshold of the sine circuitry, *excitation* of the sine circuit would be crucially involved in sine production, in contrast to our model in which sine song arises via post-*inhibitory* rebound. To test this hypothesis, we knocked down (excitatory) acetylcholine receptors within TN1 of males that courted a wildtype female. This did not alter the proportion of complex song compared to controls (**see right**), suggesting that excitation of TN1 is not a key mechanism in driving sine and complex song.

Redacted

9. Prior studies have already established that the male's interactions with the female drive song. What in those earlier studies or here supports the idea that female cues disinhibit versus excite the song pathway? Functionally these two mechanisms both result in song and there is nothing obvious to me in the behavior that would support one over the other.

We believe several pieces of evidence support the idea that female cues disinhibit versus excite the song pathway (this is discussed in more detail above):

First, rebound sine for pIP10 activation in solitary males is very brief compared to that observed following activation in presence of a female (Fig 2c-d), as expected for a baseline inhibition of the song circuit, which is relieved by female cues.

Second, acute P1a activity (as induced via tapping of the female's abdomen) does not drive acute song (Fig. 4a), whereas concurrent activation of P1a and pIP10 drives complex song (Fig. 4d; which in turn is rarely present for activation of only pIP10, Fig. 2c). If P1a drove excitation of the song circuit, rather than disinhibition, we'd expect an acute song response during activation of P1a. These results strongly support the idea that female cues disinhibit rather than excite the song pathway.

Third, we used our model to show that disinhibition outperforms excitation in explaining naturalistic bout statistics (**new Fig. 5f**).

Fourth, we performed new imaging experiments, combining optogenetic activation of P1a with GCaMP imaging from GABAergic inhibitory neurons (expressing *Gad1*), to show that **pairs of inhibitory P1a follower neurons exist with the predicted disinhibitory dynamics (new Fig. 4o-p)**. While these results address the Reviewer's concern, they leave the ultimate test of causal involvement of the identified disinhibitory neurons in complex song production for future work. We feel that work is outside the scope of this study - identifying the specific disinhibitory motif likely will require analysis of a male connectome (that does not yet exist) and behavioral screens of sparse genetic driver lines.

10. While behavioral observations and peripheral specializations make it likely that there is some form of crosstalk between pulse and sine producing networks, evidence that it depends specifically on mutual inhibition as schematized in Figure 2m is lacking. The headless experiments in Figure 2 suggest that the putative mutual inhibition circuits reside in the VNC, but that leaves a lot of ground unexplored. The lack of an explicit circuit description to address this idea is a weakness of the study.

See response to comment 7 above. Our new evidence for endogenous rebound circuitry also addresses this point. Specifically, we show that mutant males systemically lacking *Ih* (a major ingredient for rebound excitability) exclusively sing simple pulse song, and further that TN1-specific knockdown of *Ih* and *Rdl* (GABA-A receptors) both reduce song complexity (**new Fig. 3f-k**). These results strongly support our model in which complex song arises via endogenous rebound circuitry and GABA-A mediated mutual inhibition.

11. The two photon experiments are a start in the right direction towards such an explicit circuit dissection, but do not go very far, have some issues for interpretation and are not entirely consistent with a mutual inhibition model. First, while the bulk of TN1 neurons show (strongly) correlated activity, the authors focus on the smaller set that show uncorrelated activity. Is this small fraction of neurons that show anticorrelated activity larger than would be expected due to chance?

The TN1 cluster comprises 22 neurons per hemisphere, out of which a minority of around four cells constitutes the sine-driving TN1A subtype (Shirangi et al., 2016; Shiozaki et al., 2023). Consistent with our results, optogenetic activation of the entire TN1 cluster in solitary male flies has previously been shown to drive sine song, although this cluster includes the TN1C subtype which modulates properties of pulse song (Shirangi et al., 2016). We assume that the TN1 neurons active during activation of pIP10 in our imaging experiment correspond to the TN1C subtype, whereas the neurons showing anticorrelation to pIP10 activation correspond to the TN1A subtype. **We emphasize that the 'sine**

node' that is central to the rebound circuit in our model corresponds to the TN1A subtype (l. 75-76, l. 144). The comparatively low number of neurons anticorrelated to pIP10 activation is also in line with our model, which posits the imaged neurons are under baseline inhibition, which is overridden for neurons involved in pulse song production (due to optogenetic activation of pIP10), but remains intact for neurons driving sine song.

We have performed new experiments targeting cellular properties controlling rebound excitability, to dissect the proposed rebound circuitry. Specifically, we now show that mutant males systemically lacking I_h , a key ingredient for neuronal rebound excitability, lack complex song bouts (**new Fig. 3f-i**). Further, we show that downregulating I_h or GABA-A receptors (*Rdl*) within TN1 neurons reduces song complexity (**new Fig. 3 j-k**). **These results strongly support our model and provide a mechanistic understanding of complex song production.**

Second, I have to admit being left confused by this set of results, as Shirangi et al (2016) reported that stimulation of TN1 neurons generally drives sine song. Shouldn't most TN1 neurons be suppressed by pIP10 activation (which drives pulse song, as shown here in Figure 2c)?

Please see above. The TN1A subpopulation of TN1 that is thought to drive sine song comprises around 4/22 neurons per hemisphere (Shirangi et al., 2016; Shiozaki et al., 2023), and the remaining ~18 neurons include cells involved in pulse song production (TN1C). Nonetheless, activation of all TN1 neurons (as we perform in our study, and as is done in the Shirangi et al. study) does drive sine song (could be due to stronger/direct synaptic connections from TN1A neurons to motor neurons (Shirangi et al., 2013)). Together with our behavioral results (highly reliable pulse song and rare rebound sine for pIP10 activation in solitary males; Fig. 2c), we expected to see only low amounts of rebound activity in the TN1 population during calcium imaging. In a recent preprint, Shiozaki et al. 2023 replicate our finding for pIP10>chrimson, TN1>GCaMP, showing that a large fraction of TN1 neurons is responsive during pulse song production, while others are active during sine song production.

Third, Dsx also labels dPR1 neurons which are thought to drive pulse song.

We have now added text (**lines 566-568**) to emphasize that while we used flies that express GCaMP6s in all Dsx+ neurons, we only imaged the Prothoracic and Mesothoracic neuromeres and the Accessory Mesothoracic neuropil of the ventral nerve cord. All together these regions house the Pr1-3, Pr4, Ms1-3 and TN1 cluster of neurons (Nojima et al. 2021 Fig. S1) whose somas have distinct and identifiable locations. The data shown are from manually segmented somas from these regions that, based on their anatomical location, were unambiguously identified as TN1 neurons. TN1 can be distinguished from dPR1 (which belongs to the Pr1-3 cluster) based on the position of the somas in the anterior-posterior axis. Similarly, TN1 can be readily distinguished from its neighboring clusters (Pr4 and Ms1-3) based on its more lateral and ventral location relative to the Accessory Mesothoracic neuropil, as well as the smaller size of its somas. Our manual segmentation was based on these criteria rather than on neural responses, so we are confident about our designation of neurons analyzed in Fig. 3 as TN1.

Although this dissociation may suggest that the Dsx population labeled here may be excited or suppressed by pIP10, what evidence is there that mutual inhibition accounts for their anticorrelated

activity? Does the proportion of correlated/anticorrelated neurons mirror the proportion of the purported pulse-driving dPR1 vs. the sine-driving TN1 neurons? Is it possible to determine the identities of these correlated/anti-correlated neurons, or to selectively record from these populations? Lastly, what is the identity of the putative inhibitory neurons that are interposed between the anticorrelated populations? This is a place where a good initial observation could motivate a more explicit test of the circuit model. This characterization would likely need to extend beyond calcium imaging, which is generally not well-suited to characterize inhibitory networks. Ultimately, this more explicit characterization is lacking and it leaves the nature of the advance hard to assess.

We have performed new experiments that provide further evidence for the proposed mutual inhibition / rebound circuit. Specifically, we recorded song of I_n mutant males to show that post-inhibitory rebound excitability is required for complex song production, and we show that rebound excitability and GABA receptors within TN1 neurons are required for complex song production (**new Fig. 3f-k**).

Our model is further supported by a recent preprint (Amin et al., 2023) that broadly links inhibition to sine song production, via GABAergic Fru+ neurons of the VNC that express *Gad1* and *Rdl*.

12. In line 166, the authors conclude that pC2 neurons serve as the main determinant of song composition. How is this a distinct advance from work presented in references 8, 26 and 38?

- Reference 8: Deutsch et al. 2019. Figure 5 panels A-C in Deutsch et al. 2019 show that optogenetic activation of pC2 neurons drives pulse and sine song. However, Deutsch et al. present no evidence that different levels of pC2 activity can drive simple and complex song and hence can explain the context dependence of song patterning we describe. In particular, within the narrow range of optogenetic stimulus parameters employed by Deutsch et al., sine probability following pC2 activation was approximately constant, unlike the linear relationship between sine probability and stimulus duration discovered in our work (emphasizing the relevance of exploring a broad range of optogenetic stimulus parameters). Further, no study prior to ours showed evidence that different levels of pC2 activity drive finer-scale song patterning (Pfast, Pslow, sine).
- Reference 26: Zhou et al. 2014. Zhou et al. 2014 focus on Dsx+ neurons in the female, and to our knowledge do not characterize the role of pC2 in male song composition.
- Reference 38: Kohatsu and Yamamoto 2015. Kohatsu and Yamamoto used a more coarse-grained analysis of male courtship behaviors in relation to pC2 activity. In particular, they did not explicitly record song, but relied on male wing extension as a proxy for song. This limitation did not allow the authors to distinguish pulse and sine song, nor did they describe context dependent song patterning and the involvement of pC2 therein. Hence, their study did not suggest that pC2 is the main determinant of song composition.

12. Section 5. The authors do a nice job of varying the optogenetic excitation of P1a and pC2. However, again I feel that the model proposed is not sufficiently explored and tested. For instance, in line 173, the authors state that their results suggest that “P1a mediates disinhibition (via as of yet unknown circuitry)” of VNC. This is a rather unsatisfying claim and leads to the speculated existence of additional neurons in Fig. 4q – but it is unclear if these neurons exist.

To add further support for our proposal that P1a mediates disinhibition, we have performed new modeling, analyses and experiments. First, we used our model to show that a disinhibitory circuit motif provides superior fits to behavioral data compared to a quasi-equivalent excitatory motif (**new Fig. 5f**), and we find that this is due to the excitatory model failing to produce bouts with leading sine song (in contrast to the disinhibitory model and wild-type song; **new Fig. S7h-j**). Second, in publicly available female connectome data, we found that disinhibitory motifs are common for pC1 follower neurons (and P1a, albeit not present in the female, constitutes a subset of pC1 in the male; **new Fig. S4o-p**). Third, we performed new imaging experiments, combining optogenetic activation of P1a with GCaMP imaging from GABAergic inhibitory neurons (expressing *Gad1*), to show that **pairs of inhibitory P1a follower neurons exist with the predicted disinhibitory dynamics (new Fig. 4o-p)**. While these results address the Reviewer's concern, they leave the ultimate test of causal involvement of the identified disinhibitory neurons in complex song production for future work.

13. Section 5, sensory cues. In the GLM model used to predict behavior, I am again left wondering what is the novel advance, given that the predictor variables (distance, angle, tapping, etc.) that feed into the GLM appear to be known influencers of song patterning in flies. Perhaps one novel aspect is that the authors use an estimate of p1A activity, but this estimate may largely reflect the behavior itself, since it is generated by using the tapping behavior and convolving that with previously observed calcium responses and GCaMP decay dynamics (e.g., Clowney et al., 2015). Although tapping might acutely excite p1A, how well does it explain the totality of p1A activity, particularly given prior work suggesting that p1A is modulated on longer timescales and may be sensitive to sign stimuli provided by the female (Clowney et al., 2015)? Generally, this seems to be a way to bring neural activity into this model, but it is unclear the degree to which this is distinct from just the behavioral history.

While the role of acute P1a activity in mediating chasing during courtship is a recent finding (Sten et al., Nature 2021) **no prior study has investigated the role of tapping or acute P1a activity at the timescale of milliseconds to seconds in modulating song patterning.**

Importantly, Sten et al. 2021 considered timescales of 10s of seconds and used experiments with tethered flies; we note that the complex song dynamics observed in our free behavior experiments may not be easy to recapitulate in tethered studies, and therefore **our leveraging of natural behavior to uncover circuit dynamics is an important advantage of our study.**

We agree with the Reviewer that it was previously known that individual features influence song output (and we cite these studies). Building on these previous results, the GLM approach allowed us to rank the influence of individual features, and this ranking suggests that tapping/P1a activity contribute less to song patterning than primarily visual features (male-female angle and distance; Fig. 4k), which (together with our new results on pC2 inactivation, **new Fig. 4q-s**) supports our conclusion that pC2 (integrating visual information) is upstream in the hierarchy. We further agree with the Reviewer that our tap-based approach likely underestimates P1a activity (which we now emphasize in the text), and we emphasize that our model posits pC2 as an additional input to P1a that can drive these neurons in lieu of taps. We also point out that our deep learning-based detection of taps during natural behavior (via SLEAP) is new, and never explored as a predictor for song patterning in prior studies.

14. Relatedly, to what degree is tapping itself necessary for the generation of complex bouts? If male flies are physically prevented from tapping the females, but allowed to approach them closely, do they

still generate complex song? This seems to be an important aspect of the model, given the role proposed for p1A in helping to mediate disinhibition of downstream neurons.

Our modeling results (GLM, Fig. 4j-l) imply that visual features (mfAngle and mfDist) outperform purely tap-inferred P1 activity in predicting simple vs. complex bout production, suggesting taps alone do not suffice for complex bout generation.

15. Section 6. While the authors can nicely recapitulate their results with the given circuit model, the overall lack of testing of various aspects of the model leave me uncertain that the proposed model is the best or only explanation for their results. Indeed, the authors analyze previous inhibition data in light of their model, and while they can explain these prior data, it requires that they modify their model to propose the existence of a chemical synapse between pIP10/VNC and an electrical synapse between pIP10/inhibitory interneurons Fig. S7f. Is there evidence that either these VNC interneurons, or this electrical synapse exist? Overall, the model proposed here posits the existence of two classes of inhibitory interneurons in the VNC (one on pulse and one on sine-producing neurons), one interneuron in the brain (to allow for p1A disinhibition) and several new connections between these and other neurons. It seems to this reviewer that there ought to be some burden of proof here when proposing additional, previously unknown, circuit components.

To address this (and other) Reviewer's comment, we performed additional experiments to test several aspects/predictions of our model. First, we performed new behavioral experiments to test and confirm our model prediction that post-inhibitory rebound dynamics (via I_h) are required for complex song, and rebound excitability specifically of TN1 neurons mediates rebound sine song (**new Fig. 3f-k**). We further performed new analyses to show that a circuit model comprising disinhibition exhibits superior fit to wild-type song statistics compared to a model comprising a quasi-equivalent excitatory motif (**new Fig. 5f**), and we find that this is due to the excitatory model failing to produce bouts with leading sine song (in contrast to the disinhibitory model and wild-type song; **new Fig. S7h-j**). In lieu of a male connectome, we searched and identified disinhibitory circuit motifs postsynaptic to neurons of the pC1 neural cluster in the public female connectome (P1a is a subset of pC1 in the male; **new Fig. S4o-p**), and we collected new imaging data to confirm our model hypothesis that disinhibitory circuit motifs exist postsynaptic to P1a in the male (**new Fig. 4o-p**). While these new analyses provide mechanistic evidence for the proposed rebound mechanism of sine/complex song generation, we do not yet have causal evidence that the identified P1a-follower neurons mediate disinhibition of the song circuit. The ultimate test of causal involvement of the identified disinhibitory neurons in complex song production is hence left for future work, and we now explicitly mention this in the main text (**I. 224-226**).

16. A last more minor point is that I believe the authors miss the opportunity to link their research to another, older (and perhaps partly forgotten) line of research, namely changes in noctuid moth flight escape behaviors triggered by different intensities of ultrasound. While the birdsong references are interesting, there the underlying song pattern is highly similar in the two contexts, but slightly faster and more precise in the presence of a female. In the moth, like the fly, changes in sensory cues drive quite different motor patterns: Directed flight away from the ultrasound source when it is relatively faint, and a power dive into the substrate when the ultrasound intensity passes a critical threshold. This seems

more similar to what the male fly does, which is to change the wing motor pattern as sensory cues change along a continuum.

We thank the Reviewer for pointing us to this relevant line of research. We have added text to discuss the context-dependence of escape behavior in noctuid moths, crickets, and flies (**I. 370-376**).

Reviewer Reports on the First Revision:

Referees' comments:

Referee #1 (Remarks to the Author):

The authors have responded to my concerns with language or figure design changes, or by providing reasonable explanations as to why they'd rather not make a change. I think many of these changes will give the reader a more accurate appreciation for the nuances of the data and how strongly the conclusions are supported. This was a major concern of mine, as well as of Reviewer 2.

I am supportive of this paper being published in Nature. I have only 2 critical comments to make about the current manuscript that I would like to see addressed. I do not need to see another version.

1. Loss of rebound manipulations

This, for me, was the key missing piece from the previous manuscript. I hate to have to say that I find the new data unconvincing and I believe the paper was better without them. There is, for example, a roughly 5% reduction in median complex song production between *lh*-RNAi and controls. And outliers are driving some of this already small difference.

Though the full animal *lh* mutant phenotypes are strong, those effects could be coming from anywhere, and I can think of other explanations than rebound excitation. There are no physiological experiments, so (unless I am missing something) the evidence for the following strong-sounding statements is extremely weak:

"We found that compared to controls, song complexity was reduced in males with downregulated rebound excitability (Fig. 3j-k)."

"These results provide mechanistic evidence for our model, confirming that post-inhibitory rebound excitability is a key driver of complex song, and that VNC TN1 neurons exhibit hallmarks of a rebound circuit that underlie complex song dynamics".

In our lab we routinely perform screens of hundreds or thousands of RNAi lines and know very well that a 10% change in many sexually dimorphic behaviors can easily arise from slight differences in genetic background—even when the UAS-RNAi lines are supposed to have the same genetic background. All stocks went through at least one genetic bottleneck years ago and have been inbreeding in isolation for hundreds of generations.

With phenotypes this small and suspect, controls become more important. The authors do not control for the effects of the RNAi background, only for the driver line. Given the extreme variability across individual animals (and, presumably, trials), I think it is very possible that RNAi-specific controls (e.g., UAS-*lh*-RNAi/+) would make the phenotypes even smaller, or even disappear.

A smaller point: is not clear to me what exactly UAS-control-RNAi means. Is it a GFP-RNAi? An empty UAS? A scrambled hairpin?

As I said in my earlier review and above, I support this paper being published in Nature, but I have to insist that either this new data is removed (preferably) or described with the extreme skepticism that, in my opinion, it deserves.

To be specific: I do not think it is okay to say or imply that full animal lh mutants (a huge disruption) having less song complexity, on its own, provides real mechanistic support for the rebound model, especially working within the defined circuit elements. And, because of the small effect sizes and large animal-to-animal/trial-to-trial variability, I don't believe the RNAi results at all. Even if the change were certainly due to the RNAi and not genetic background or sampling, how meaningful is a 5% change in median when the spread in control animals ranges from ~35-55% (not including outliers)?

2. Opposite effects of pC2 silencing between Murthy lab papers

The inclusion of the phrase "contrasting with a previous study" is an improvement from not commenting on this issue at all, but this statement (inside a parenthetical) only points out that the authors are aware of the contrasting results between papers. It does not help the reader understand what happened or how seriously to take the new result, or indicate that the contrasting results came from within the same lab, or why they believe these new results and not the old ones, or that it is not an example of phenotype vs. no phenotype but no song vs. twice as much song. So there is a lot wrapped up in that word "contrasting" and I wish it were unfurled a little more. Maybe the current authors shouldn't have to explain the (apparently mistaken) results of earlier lab members, but it is a missed opportunity because knowing what happened could be useful for the field. I have to say that I don't understand how a different behavioral rig could go any distance toward explaining the difference between essentially no courtship (here) and twice as much courtship (previous paper).

The no courtship pC2 silencing is more in line with what we see when we manipulate these neurons. So I think it's great that the previous paper will be corrected and I commend the authors for that.

Mike Crickmore

Referee #2 (Remarks to the Author):

This is a strong revision that addresses concerns raised in the first round by myself and other reviewers. The strength of this paper, and the appeal to a general audience, is the identification of circuit components that enable an all-or-none 'decision' (pulse vs sine song) to be made in a flexible way based on sensory feedback. The specificity of the architecture - aspects of which echo previous all-or-none selection architectures identified in the retina, in zebrafish hindbrain, and in owl tectum - is expanded here because of the powerful tools leveraged in drosophila to interesting courtship behaviors.

Two important issues that were satisfactorily addressed include:

1. Fortifying evidence of the disinhibition motif

Pitting the suitability of excitation versus rebound via disinhibition in the model, and showing the latter fits the data better, is an expected but nice result that strengthens the main claim of the paper. The new finding that Ih knockout reduces song complexity, exactly as predicted by the model, is an important addition because alternative results would have ruled out the model. Overall, I think the aggregate evidence for the proposed motif is sufficient to support the main claims of the paper, which is worded much more carefully than the first draft.

2. Putting the pulse-sine decision in context.

Though it was a small overall point in the content of this large and complex paper, I appreciate the addition of Sup Fig 1f. If presented at a journal club, I think this panel would be included in the necessary setup of the paper - to both communicate how complex the larger behavior in which the choice studied here is nested, and also to clarify what is and what is not on the table for examination. Along those lines, I appreciate that it is explicitly stated that wing choice is not being studied.

I have only a minor point and request.

The 'instead' at line 5 of the abstract unnecessarily pits the present findings against previous ones in different systems, but diverse architectures for solving common problems is likely the norm in biology. I recommend a minor re-wording, e.g. 'We demonstrate here a different principle where...'

It is good that the authors acknowledge the incompatibility with previously published work from the same lab (Deutch et al, 2019). It would be re-assuring to see the promised corrigendum as promptly as possible and ideally before the publication of this paper.

Author Rebuttals to First Revision:

To The Editors and Reviewers:

Thank you for the comments on our revised manuscript. Below we provide point-by-point responses to the Reviewers' additional comments. Reviewer comments are in blue italics and our responses in black regular font. We have now also addressed the Editorial instructions with regard to formatting of the manuscript (source data tables and a code repository are currently finalized) - please let us know what additional changes we need to make.

Thank you,
Mala (on behalf of all authors)

Referees' comments:

Referee #1 (Remarks to the Author):

*The authors have responded to my concerns with language or figure design changes, or by providing reasonable explanations as to why they'd rather not make a change. I think many of these changes will give the reader a more accurate appreciation for the nuances of the data and how strongly the conclusions are supported. This was a major concern of mine, as well as of Reviewer 2. **I am supportive of this paper being published in Nature. I have only 2 critical comments to make about the current manuscript that I would like to see addressed. I do not need to see another version.***

1. Loss of rebound manipulations: This, for me, was the key missing piece from the previous manuscript. I hate to have to say that I find the new data unconvincing and I believe the paper was better without them. There is, for example, a roughly 5% reduction in median complex song production between lh-RNAi and controls. And outliers are driving some of this already small difference.

We attribute the relatively small, but statistically significant, effect size (we find an approximately **13% reduction** in median complex song production between lh-RNAi and genotype-matched controls) to the nature of the manipulation (downregulation of gene expression via RNAi instead of complete block of expression). In combination with the stronger phenotype observed in the lh mutant males, our data indicate that lh (HCN (hyperpolarization-activated cyclic nucleotide-gated) channel) expression in TN1 neurons contributes to complex song production. Yet, we have now added text to discuss the possibility that:

- 1) neurons outside the TN1 population (in which the lh mutant is expressed) might also contribute to sine song production, and rely on lh channels for their

rebound properties - this is why we observe stronger reduction of sine song in the full animal mutant versus in the TN1 knockdown (**line 149-150**),

- 2) rebound excitation of TN1 may rely on the concerted action of several types of ion channels, including Ih, facilitating robust rebound excitability over a broad range of Ih expression levels, such that a *reduction* in Ih channel expression (via RNAi knockdown) leads to a comparatively modest effect, whereas a complete lack of Ih (as in Ih mutants) fully removes rebound excitability (**lines 150-151**),
- 3) reducing the amount of Ih channels, rather than removing all channels, could lead to stronger net inhibition, which in turn increases the Ih conductance to compensate for the reduced amount of channels. Specifically, fewer Ih channels take longer to counteract a given amount of incoming inhibition, which ultimately results in stronger and faster activation of Ih because the neurons are more hyperpolarized (e.g., see Figs 2-3 in McCormick & Pape, 1990). (**lines 151-152**)

We now added all of these explanations to the text and interpret the RNAi findings given these caveats.

Though the full animal Ih mutant phenotypes are strong, those effects could be coming from anywhere, and I can think of other explanations than rebound excitation. There are no physiological experiments, so (unless I am missing something) the evidence for the following strong-sounding statements is extremely weak:

“We found that compared to controls, song complexity was reduced in males with downregulated rebound excitability (Fig. 3j-k).”

“These results provide mechanistic evidence for our model, confirming that post-inhibitory rebound excitability is a key driver of complex song, and that VNC TN1 neurons exhibit hallmarks of a rebound circuit that underlie complex song dynamics”.

We have changed the referenced sentence to “Overall, these results suggest that VNC TN1 neurons exhibit hallmarks of a rebound circuit that underlie complex song dynamics.” (**line 152-153**), and we removed the section heading.

In our lab we routinely perform screens of hundreds or thousands of RNAi lines and know very well that a 10% change in many sexually dimorphic behaviors can easily arise from slight differences in genetic background—even when the UAS-RNAi lines are supposed to have the same genetic background. All stocks went through at least one genetic bottleneck years ago and have been inbreeding in isolation for hundreds of generations. With phenotypes this small and suspect, controls become more important. **The authors do not control for the effects of the RNAi background, only**

for the driver line. Given the extreme variability across individual animals (and, presumably, trials), I think it is very possible that RNAi-specific controls (e.g., UAS-lh-RNAi/+) would make the phenotypes even smaller, or even disappear. A smaller point: is not clear to me what exactly UAS-control-RNAi means. Is it a GFP-RNAi? An empty UAS? A scrambled hairpin?

We have now changed the wording in **Table S2** to specify that we used stocks of the genetic background that the UAS-RNAi constructs were injected into (stock #36303 from BDSC) and crossed them to the driver lines for use as controls.

As I said in my earlier review and above, **I support this paper being published in Nature, but I have to insist that either this new data is removed (preferably) or described with the extreme skepticism that, in my opinion, it deserves.**

Based on additional feedback from the 2nd Reviewer (see below), we have now added text to discuss the limitations of the mutant and knockdown experiments (**lines 149-152**), rather than removing the experiment.

To be specific: I do not think it is okay to say or imply that full animal lh mutants (a huge disruption) having less song complexity, on its own, provides real mechanistic support for the rebound model, especially working within the defined circuit elements. And, **because of the small effect sizes and large animal-to-animal/trial-to-trial variability, I don't believe the RNAi results at all.** Even if the change were certainly due to the RNAi and not genetic background or sampling, how meaningful is a 5% change in median when the spread in control animals ranges from ~35-55% (not including outliers)?

See our comments above.

2. Opposite effects of pC2 silencing between Murthy lab papers

The inclusion of the phrase “contrasting with a previous study” is an improvement from not commenting on this issue at all, but this statement (inside a parenthetical) only points out that the authors are aware of the contrasting results between papers. It does not help the reader understand what happened or how seriously to take the new result, or indicate that the contrasting results came from within the same lab, or why they believe these new results and not the old ones, or that it is not an example of phenotype vs. no phenotype but no song vs. twice as much song. So there is a lot wrapped up in that word “contrasting” and I wish it were unfurled a little more. Maybe the current authors shouldn't have to explain the (apparently mistaken) results of

earlier lab members, but it is a missed opportunity because knowing what happened could be useful for the field. I have to say that I don't understand how a different behavioral rig could go any distance toward explaining the difference between essentially no courtship (here) and twice as much courtship (previous paper).

We have now added text (**line 194-196**) to emphasize that due to the discrepancy between the previous finding in Deutsch et al. 2019 and our model, we re-created the pC2>TNT flies from scratch, used PCR to confirm the presence of TNT, and re-ran the silencing experiment (and controls) in the new setup. We have now further submitted a corrigendum for Deutsch et al. 2019 to Current Biology.

The no courtship pC2 silencing is more in line with what we see when we manipulate these neurons. So I think it's great that the previous paper will be corrected and I commend the authors for that.

Mike Crickmore

Referee #2 (Remarks to the Author):

This is a strong revision that addresses concerns raised in the first round by myself and other reviewers. The strength of this paper, and the appeal to a general audience, is the identification of circuit components that enable an all-or-none 'decision' (pulse vs sine song) to be made in a flexible way based on sensory feedback. The specificity of the architecture - aspects of which echo previous all-or-none selection architectures identified in the retina, in zebrafish hindbrain, and in owl tectum - is expanded here because of the powerful tools leveraged in drosophila to interesting courtship behaviors.

Two important issues that were satisfactorily addressed include:

1. Fortifying evidence of the disinhibition motif

Pitting the suitability of excitation versus rebound via disinhibition in the model, and showing the latter fits the data better, is an expected but nice result that strengthens the main claim of the paper. The new finding that lh knockout reduces song complexity, exactly as predicted by the model, is an important addition because alternative results would have ruled out the model. Overall, I think the aggregate evidence for the proposed motif is sufficient to support the main claims of the paper, which is worded much more carefully than the first draft.

2. Putting the pulse-sine decision in context.

Though it was a small overall point in the content of this large and complex paper, I appreciate the addition of Sup Fig 1f. If presented at a journal club, I think this panel would be included in the necessary setup of the paper - to both communicate how complex the larger behavior in which the choice studied here is nested, and also to clarify what is and what is not on the table for examination. Along those lines, I appreciate that it is explicitly stated that wing choice is not being studied.

I have only a minor point and request.

The 'instead' at line 5 of the abstract unnecessarily pits the present findings against previous ones in different systems, but diverse architectures for solving common problems is likely the norm in biology. I recommend a minor re-wording, e.g. 'We demonstrate here a different principle where...'

We have now changed the wording of the abstract (**lines 5-6**) accordingly.

It is good that the authors acknowledge the incompatibility with previously published work from the same lab (Deutsch et al, 2019). It would be reassuring to see the promised corrigendum as promptly as possible and ideally before the publication of this paper.

The corrigendum to Deutsch et al., 2019 has now been submitted to Current Biology, and we will update the Editors at Nature on its publication.

"I read reviewer 1's report with great interest and revisited the key issues - centering around point #1 and the *Ih* knockdown (in the whole fly) and specifically in TN1. **I agree with reviewer 1 that the language the authors use to report the data are overstated with respect to what the data actually show. There are also some oddities in the presentation of the data in Fig. 3. I'm not sure I think they should eliminate the data altogether - as they DID see a significant reduction in song complexity with the TN1 specific knockdown - but - something is amiss. I'd like to hear what the authors have to say about the finding that the *Ih* mutant can still do pulses and bouts.** This is a non-trivial result.

Our model posits that while there is a direct pathway for driving pulse song, sine song is driven primarily via post-inhibitory rebound. **The finding that the *Ih* mutants nearly exclusively produced pulse song is very well in line with our model.** Specifically, while we expected the system-wide lack of *Ih* to remove rebound excitability in both the

pulse and the sine driving neurons, and therefore to strongly reduce the amount of complex bouts, we did not expect a change in the ability to sing pulse song alone (no post-inhibitory rebound is needed for pulse-only simple song). Our simulations show that a model lacking rebound excitability in both the pulse and the sine node exclusively produces pulse song (via the direct excitatory $pc2 \rightarrow p$ connection), which matches the experimental observations. Yet, in most of the tested lh mutants, overall song probability was strongly reduced compared to controls, and not all males sang. Those that were able to sing produced pulse-only song. In part, the substantial reduction in the amount of song might be attributable to the strong / system-wide perturbation.

1. How do the authors explain this - if their model requires rebound mechanisms to transition from pulse to sine? Also surely there are other CPGs in the system (that may be operating within pulse or within sine) - the fact that these mutants can sing at all is interesting...but it makes me wonder to what extent lh was really knocked down - or if other channels (e.g. LVA Ca channels could, in some compensatory way, be homeostatically overexpressed to enable the cells to discharge). Any way you slice it, the existence of song and even complex song in these mutants needs more forthright explanation.

Regarding the (expected) finding of pulse song in the lh mutants, please refer to our response to the previous comment. Regarding the finding of sine and complex song in the lh mutants, we re-examined the raw song data for the lh mutant outliers (**Fig. 3i**; one each for mutant A and B, with merely ~40% simple pulse bouts near the female). We confirmed the detection of true sine song produced by these two males - we are not at present sure what the explanation is (homeostatic compensatory mechanisms?), and we will look into it in future experiments. We address this issue in **lines 150-152**.

2. I'm confused why Fig3 presents the whole lh mutant and the TN1 specific lh knockdown with different metrics. Panels h and i show the probability of simple bouts as the x axis. Then Panels j and k switch the x axes to different measures of the bout complexity - the mean number of ps alternations (j) and the probability of a complex bout (k). Are the control data in k the same as the control data in i? Why not just show the TN1 specific knockdown song analytics in a way that's comparable to the whole-fly lh knockdown? I agree with the other reviewer that the effect of lh knockdown in the TN1 neurons alone is significant but subtle - especially if rebound is the KEY mechanism revealed by this paper for song complexity. I'm guessing that either this is an incomplete knockdown of lh or that other mechanisms exist? I am not a drosophila neurobiologist familiar with these methods - so I defer to Reviewer 1

(who seems to be) in what to expect in an RNAi-mediated knockdown. I hate to ask for more expts at this point, but Fig3k is definitely not the slam dunk that the authors were probably expecting and I agree that the data are out of alignment with the text which presents the lh knockdown as more robust than Fig3k shows.

Our choice of different metrics for mutant and knockdown experiments in **Fig. 3** was motivated by the different targets of the genetic manipulations: Since the system-wide lh mutant experiments targeted rebound excitability in *both* the pulse and the sine driving neurons, we expected to see a reduction/removal in the amount of complex bouts both starting in pulse mode (the majority of complex bouts in wild-type flies) *and* starting in sine mode, and therefore chose the proportion of simple pulse bouts as a metric that captures reduction in both types of complex song (1 - proportion complex pulse - proportion complex sine). The TN1-specific knockdown of lh targeted rebound excitability of primarily sine-driving TN1A neurons, with the strongest expected effect on the proportion of complex pulse-leading bouts. In the event of (e.g. spontaneous) activity of the sine-driving neurons, subsequent rebound activity in the pulse driving neurons is possible (because lh is not targeted in the majority of pulse-driving neurons), which could lead to underestimation of the effect size when using the identical metric as for the lh mutants.

So I agree with Reviewer 1 in that the results are less dramatic than they 'should' be if the model is right. In the initial examination of this I trusted the language of their text, observed the significance in Fig3 and was satisfied. I'm glad Reviewer 1 looked more closely and noticed that there is still a lot of complex singing in these mutants! **If rebound is what enables the ps transition, then how is it the case that TN1 lh knockdowns still have 40% of their bouts with ps transitions? And the authors either need to eliminate the data (less preferred) or address head on why/how this is happening."**

For the TN1-specific knockdown of lh, we expected a weaker effect than in the mutants, due to the nature of the manipulation. Specifically, we expected to see a reduction, not a removal, of complex bouts. In addition to the reduction that we see, we find that the complex bouts that are produced contain fewer pulse-sine alternations, which is also in line with our expectation. Lastly, we independently show that TN1-specific knockdown of GABA-A receptors (Rdl) – which are also required for post-inhibitory rebound – leads to similar results, providing further evidence for our model. Yet, we have now added text to discuss the possibility that additional ion channel types could be involved in rebound excitation, and that homeostatic mechanisms could compensate for the knockdown (**lines 150-152**). Specifically, reducing rather than removing lh might allow for stronger inhibition of TN1, which in

turn leads to stronger and faster activation of these available channels due to the facilitated stronger hyperpolarization (Figs. 2-3 in McCormick & Pape, 1990).

So, there is general agreement that the data should be kept in while acknowledging and discussing the limitations.

We have now added text to further acknowledge and discuss the limitations of our Ih mutant and knockdown experiments (**lines 149-152**).

Likewise, we would ask that the wording and descriptions around the differences between this work and the previous work from your lab be stated clearly, as the reviewers suggest.

We have added text to expand on the differences between our work and Deutsch et al. 2019 (**lines 194-296**).

Reviewer Reports on the Second Revision:

Referees' comments:

Referee #1 (Remarks to the Author):

I thank the authors for responding to my concerns and support publication.

Mike Crickmore

Referee #2 (Remarks to the Author):

I am generally satisfied with the reviewere responses and think the paper should be published with minimal delay.